# Intelligent in-cell electrophysiology: Reconstructing intracellular action potentials using a physics-informed deep learning model trained on nanoelectrode array recordings

Keivan Rahmani[1], Yang Yang[2,3], Ethan Paul Foster [2,4], Ching-Ting Tsai[2,3], Dhivya Pushpa Meganathan[1], Diego D. Alvarez [5], Aayush Gupta[6], Bianxiao Cui [2,3,7], Francesca Santoro [8,9,10], Brenda L. Bloodgood[5], Rose Yu[6], Csaba Forro [2,3,8,9,10,11] ✉ & Zeinab Jahed [1,12] ✉

Intracellular electrophysiology is essential in neuroscience, cardiology, and pharmacology for studying cells' electrical properties. Traditional methods like patch-clamp are precise but low-throughput and invasive. Nanoelectrode Arrays (NEAs) offer a promising alternative by enabling simultaneous intracellular and extracellular action potential (iAP and eAP) recordings with high throughput. However, accessing intracellular potentials with NEAs remains challenging. This study presents an AI-supported technique that leverages thousands of synchronous eAP and iAP pairs from stem-cell-derived cardiomyocytes on NEAs. Our analysis revealed strong correlations between specific eAP and iAP features, such as amplitude and spiking velocity, indicating that extracellular signals could be reliable indicators of intracellular activity. We developed a physics-informed deep learning model to reconstruct iAP waveforms from extracellular recordings recorded from NEAs and Microelectrode arrays (MEAs), demonstrating its potential for non-invasive, long-term, high-throughput drug cardiotoxicity assessments. This AI-based model paves the way for future electrophysiology research across various cell types and drug interactions.

The drug development process is costly and inefficient, taking 10–15 years and approximately $5 billion per drug[1,2], with 63% of costs in preclinical development and 32% in clinical studies[3]. A key challenge in drug development is the limited predictive power of preclinical screening, which relies on animal models and cell lines[4] that may not accurately represent human physiology due to interspecies differences. Only 12% of drugs entering clinical trials are approved[5], with

cardiotoxicity and hepatotoxicity as major reasons for drug failure[6,7], primarily due to drug-induced cardiotoxicity stemming from adverse effects on ion channels critical for action potential generation, which alter cardiomyocyte electrophysiology and elevate arrhythmic risks. Electrophysiology, which investigates the electrical characteristics of biological cells and tissues, is crucial for understanding drug mechanisms, developing cardiac and neurological therapies, and

evaluating cardiotoxicity across various drugs[8]. Preclinical proarrhythmic assessments currently rely on in vitro hERG potassium channel assays or in vivo electrocardiogram measurements in large animals[9]. However, these methods are costly and labor-intensive, so they are usually employed late in development, limiting the feasibility of making chemical modifications. Furthermore, evaluating a drug's effect solely on the hERG channel can be insufficient, as drugs can affect multiple ion channels. For example, Verapamil, a strong hERG blocker, is clinically safe due to its calcium channel blocking effect, which offsets its impact on hERG channels[10]. Intracellular action potentials (iAPs) reflect the activity of multiple cardiac ion channels, and subtle changes in iAPs can indicate cardiotoxicity or serve as biomarkers[8]. Advances in cardiomyocyte culture and the use of human induced pluripotent stem cell-derived cardiomyocytes (hiPSC-CMs) have greatly enhanced the availability of human cardiomyocytes for in vitro studies. Consequently, the CiPA initiative, launched by the FDA and other agencies, proposes in vitro screening of cardiac iAPs[11,12]. This approach may offer a more accurate assessment of cardiac toxicity than the hERG assay and accelerate toxicity evaluation of early drug candidates.

The gold standard for intracellular electrophysiology technique currently used in all areas of medical sciences is the patch-clamp technique. This method is very powerful and can measure intracellular potentials with very high precision. However, this technique is low in throughput, manual, and remains invasive to the recorded cell (Fig. 1b). Automated patch-clamp systems, while improving throughput, require enzymatic dissociation of cells, which can alter cardiomyocyte electrophysiological properties[13,14]. Optical recording of iAPs using voltage-sensitive dyes or proteins is versatile and scalable but limited by low sampling rates, reduced sensitivity, and potential cytotoxicity from photobleaching, which may affect cellular behavior[15,16].

Extracellular electrophysiology techniques such as microelectrode arrays (MEAs) overcome the invasiveness and throughput limitations of patch clamp[17,18], however, as the recording electrode remains outside the cell, this technique provides limited information on the shape of the electrical signals and cannot resolve the subtle changes in the intracellular potentials required for cardiotoxicity assessment[17–19].

Nanoelectrode arrays (NEAs) (Fig. 1a), consisting of free-standing nano-scale electrodes, up to 200 times smaller than the size of the cell, have emerged as a promising method that combines the advantages of intra and extracellular electrophysiology techniques and have the ability to record high throughput extracellular signals, and on-demand intracellular signals, in parallel, from single cells in various cell types such as neurons and cardiomyocytes. A variety of NEAs have recently been developed[20–24], each differentiated by their shape, throughput, and access mechanisms to intracellular potential. A key challenge with NEAs lies in accessing the intracellular potential of cells. Certain NEAs can spontaneously access the intracellular space, but this generally suffers from limited probability and control. An alternative is the use of transient and controlled electroporation. This method, though more complex, uses a short electric pulse to temporarily create localized pores in the cell membrane, allowing the electrode to gain intracellular access and record iAPs at specific intervals. While these pores are highly localized, they can still disrupt cellular physiology at the nanoscale, and they eventually reseal, limiting the duration of intracellular recordings. A recording method that has the advantages of being high-throughput, long-term, and non-invasive, like extracellular measurements, yet as accurate as patch clamp or intracellular recordings, would be optimal. Here, we aim to build a model to reconstruct iAP waveforms using eAP recordings. Existing models use circuit elements relating the physical parameters such as gap size, double layer capacitance, and electrode properties to model the relationship between intra and extracellular membrane potentials[25–30].

Models such as the Bidomain[31], Extracellular-Membrane-Intracellular[32], and Kirchhoff Network Model[33] have attempted to describe physical relations between extracellular field potentials to intracellular waveforms in cardiac cells. However, A fundamental challenge common to these models is their heavy reliance on empirical data for parameter estimation and validation. The data, crucial for the accuracy of these models, is often sparse and not readily available and might vary from cell to cell[34], restricting their broad applicability. Our study introduces the Physics-Informed Attention-UNET (PIA-UNET), a deep-learning approach for reconstructing iAP waveforms from eAP waveforms. Unlike traditional models that rely on extensive parameter estimation, PIA-UNET intuitively translates the relationship between eAPs and iAPs by focusing on intrinsic patterns, thus bypassing the complex parameter estimation step. We leveraged recent advancements in our NEA technology[35] to simultaneously record thousands of eAP/iAP pairs. Our NEAs or "nanocrowns" feature a nanoscale crown structure at their tips with a thickness of ~100 nm. This design enables the acquisition of high-quality extra and intra-cellular signals from cells.

Through quantitative analysis of correlations between eAP and iAP pairs, our study posits that there is enough information in eAPs to reconstruct iAPs. We then show that the iAP can fully and accurately be reconstructed from the eAP recordings using our physics informed deep learning model.

## Results and discussion

### Recording of time-synchronized pairs of eAP and iAP

Our goal for this work is to reconstruct iAP waveforms from eAP recordings by using time-synchronized pairs of eAP and iAP recordings with nanoelectrodes to train a deep-learning model. We seek to answer the question of whether enough information is contained in the eAP for a deep learning model to be able to reconstruct the iAP. Crucial for the effectiveness of any deep learning model is the quality of the training dataset. Therefore, while previous studies have demonstrated the similarity between normalized iAP waveforms recorded by NEAs and the gold standard patch clamp technique, we conducted a more comprehensive analysis (as detailed in Supplementary Note I). We established a signal-to-noise ratio (S/N) cutoff of 90 dB (the eAP magnitude relative to the recording noise) to ensure the fidelity of the NEA recordings relative to patch clamp recordings. This S/N cutoff was enforced when selecting data in all subsequent analyses. To obtain precise time- synchronized pairs of eAP and iAP, we used recordings from neighboring nanoelectrode channels in a confluent monolayer of human stem-cell-derived cardiomyocytes that are in close physical proximity, where one nanoelectrode channel measured intracellularly while the other extracellularly. In a separate experiment, we demonstrated that two neighboring nanoelectrode channels exhibit highly similar iAP waveforms when comparing them with various metrics including differences in their cycle times (dCT) and action potential durations (APD), as well as their correlation (r) and Mean Absolute Error (MAE) over an extended period of time, and under various drug conditions (Fig. 1c–f) (detailed in Supplementary Note II). Therefore, we assume that the iAP collected from a nanoelectrode channel can act as an accurate training target to reconstruct from the eAP measured in a neighboring nanoelectrode channel, as the iAP waveforms are extremely similar over nanoelectrode channels in close proximity. To collect a diverse spectrum of synchronized eAP and iAP pairs for our training dataset, we introduced various ion-channel blockers in a dose-dependent manner to hiPSC-derived cardiomyocytes (hiPSC-CMs) to achieve different action potential shapes. The drugs used included dofetilide (primarily blocks hERG (IKr) potassium channels)[36], quinidine (blocks Na + channels (INa), K + channels (IKr and IKs), and Ca2 + channels (ICa))[37–39], nifedipine (selectively blocks L-type Ca2+ channels (ICa-L))[40], flecainide (blocks Na + channels (INa) and, to a lesser extent, K + channels (IKr))[41,42], lidocaine (primarily blocks Na + channels (INa))[43] and propranolol, a beta-blocker that also blocks Na + channels

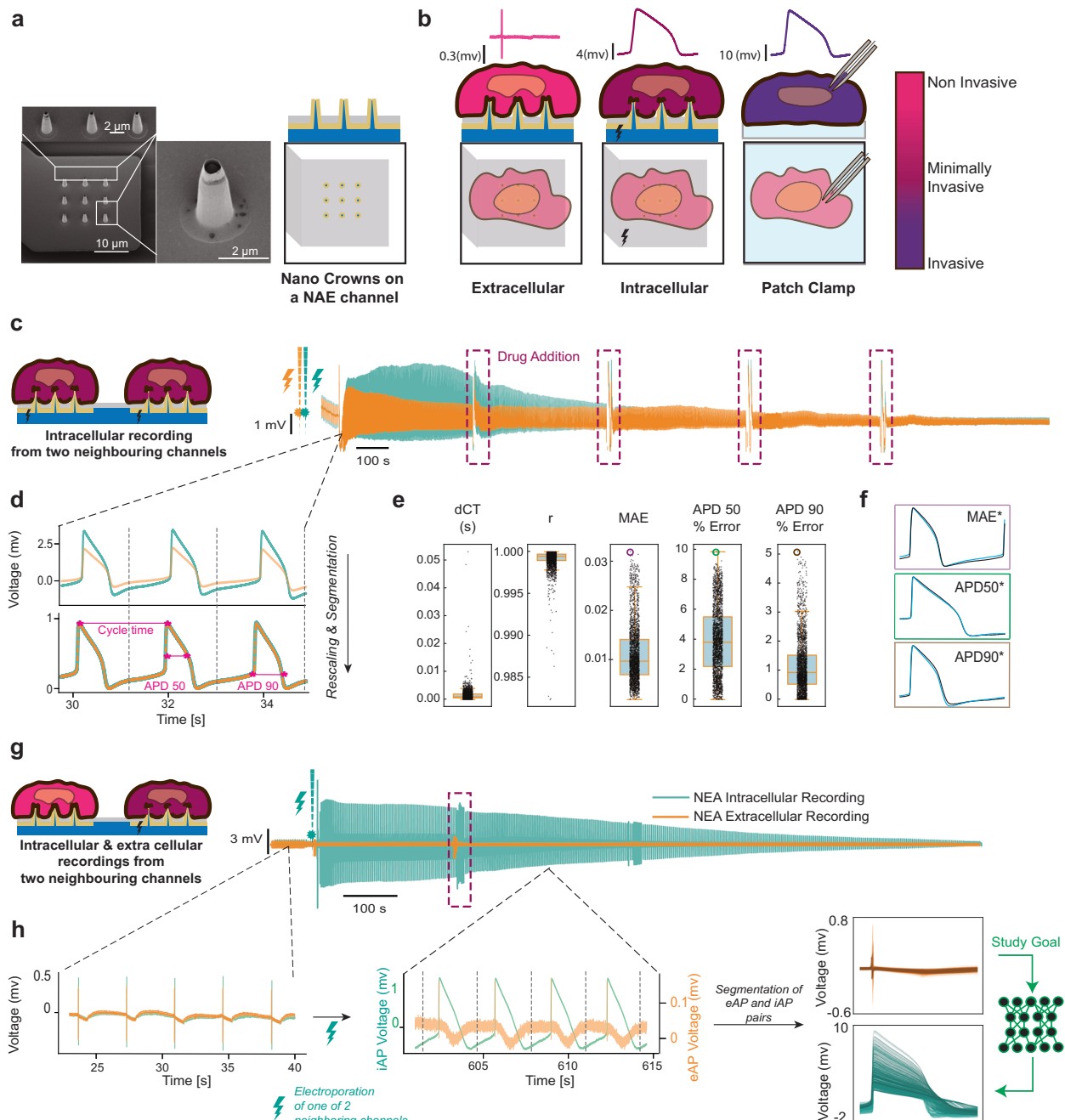

**Fig. 1 | Nanoelectrode eAP and iAP data collection and pre-processing.**
**a** Scanning Electron Microscope (SEM) image of an NEA channel with nine nano-crown electrodes, alongside a schematic of the NEA setup. **b** A methodological comparison for capturing cardiac action potentials, presenting NEA alongside the reference patch clamp (PC) technique, ordered by their degree of invasiveness. The NEA's functionality to convert eAPs into iAPs through precise biphasic electric pulses (electroporation) is demonstrated. **c** Simultaneous iAP recording from neighboring channels with a multi-step addition of Dofetilide, administered in concentrations of 0.3, 1, 3, and 10 nM. The process was conducted in three stages at approximately 400, 800, and 1200 s during the recording session to collect a diverse range of iAP shapes. **d** Comparison of iAP recording from neighboring channels by normalizing and segmenting them into arrays of length 8000 indices or 1.6 s. **e** Box plot distributions of difference in cycle time (dCT), correlation coefficient (r), mean absolute error (MAE), and APD50% and APD90% errors

between neighboring and normalized iAP pairs with S/N > S/N* (= 90) for *n* = 2661 samples, comparing 22 pairs of neighboring iAP channels from two independent cell cultures. The box plot shows the median (center line), interquartile range (IQR; box bounds), whiskers (1.5×IQR), and outliers (points beyond whiskers). **f** Examples of iAP pairs from neighboring channels with the highest MAE, APD50, and APD90% errors, as indicated in the box plots. **g** collecting diverse iAPs and corresponding eAPs waveforms from neighboring channels on NEA from hiPSC-CMs, through the addition of drugs. **h** Collection of iAP/eAP pairs from neighboring channels by applying electroporation to one channel. Both iAP and eAP waveforms are normalized according to the Methods section. The signals are then segmented into windows of 800 indices or 1.6 s. The figure presents a wide range of collected eAPs and iAPs, including an overlay of the non-normalized eAP and iAP pairs. The study goal is to reconstruct iAP from eAP as illustrated.

(INa)[44]. We collected a total of 2364 synchronized eAP/iAP pairs from independent recording sets (see "Methods"). Data were obtained from ten independent experiments using different cell cultures on different NEAs from different batches and/or wafers.

## Correlations between eAP and iAP waveform features

The distinctive capability of our platform to record time-synchronized eAP and iAP pairs enabled the direct comparison of these two waveforms and computed correlations between their shape features. This is important to quantify whether there is any plausibility to the claim that the eAP holds enough information to reconstruct the iAP. For instance, eAP and iAP pairs recorded from arrhythmic cells revealed a strong correlation ($r = 0.903$) between the eAP amplitude and the iAP spiking velocity that had not been reported before in the literature (Fig. 2a). To quantitatively describe the correlations between iAP and eAP pairs, we characterized the waveforms by defining several features on each waveform (Fig. 2b, described in detail in Methods) and calculating their cross-correlations (Fig. 2c). The distribution of these eAP and iAP features for our dataset is depicted in Supplementary Fig. S10. Note that we acknowledge that the amplitude-related characteristics, but not the shape[35], of iAPs are significantly influenced by the NEA recording technique (i.e., the quality of electroporation and size of membrane pores that electrically connect NEAs to the intracellular domain). Conversely, the amplitude of eAPs can be impacted by variations in iAPs waveforms[32,45]. This is further evidenced by the gradual decrease in iAP amplitude over time due to the gradual resealing of induced membrane pores, an effect not observed with eAPs (Supplementary Fig. S11d, S11g). Hence, while detailing the iAP waveform traits, we focus on attributes less sensitive to the recording technique for iAPs. Our APD metrics, specifically APD 10 to APD 100, measure the timeframe for an action potential to decrease to 10% and recover to 100% of its peak value, providing insight into the waveform's recovery phase independent of amplitude variations. These temporal aspects offer a more stable basis for comparing iAP and eAP dynamics.

Our analysis of several eAP features, (features described in Fig. 2b **and** Supplementary Note IV), showed strong correlations with iAP durations (APDs), with correlation coefficients greater than 0.60 (Fig. 2c). To identify which specific features of the eAP signal can precisely predict iAP features and to quantitatively explore the relationships between these eAP and iAP features, we first used an efficient, fast and scalable tree boosting machine learning method, XGBoost[46–48]. Next, we split the data to assess its accuracy and generalizability.

1. **The training dataset**: included eAP/iAP recordings from NEA using hiPSC-CMs exposed to dofetilide, quinidine, nifedipine, flecainide, and lidocaine. To validate the generalizability of our study, three different test sets were used:
2. **Test 1**: hiPSC-CMs exposed to dofetilide, a drug included in the training set, were tested on a different NEA device that was not used for training recordings, to obtain eAP/iAP pairs.
3. **Test 2**: hiPSC-CMs exposed to Propranolol (a drug not included in the training set) were tested on a different NEA device to obtain eAP/iAP pairs.
4. **Test 3**: eAP/iAP pairs were recorded from a different laboratory using commercial hiPSC-CMs on a commercial MEA[49]. For this test, simultaneous patch clamp recordings of the same cells were used to obtain iAPs.

Figure 2d shows the distributions of the training and test sets for some of the eAP features and iAP APD values, while Supplementary Fig. S10 shows the distributions of all features. Our results indicate that XGBoost can accurately predict iAP features from eAP features across a wide range of waveforms, as shown in Fig. 2e, which presents reconstructed APD lines comparing predicted values with actual ones for all test set scenarios. The XGBoost model's prediction accuracy for APD

benchmarks (APD 30, APD 50, APD 70, and APD 90) was assessed across training and test sets (Fig. 2f and Supplementary Note XII). The mean absolute APD error across all benchmarks was $0.002 \pm 0.002$ s for the training set, $0.027 \pm 0.009$ s for Test Set 1, $0.046 \pm 0.012$ s for Test Set 2, and $0.046 \pm 0.036$ s for Test Set 3 (Fig. 2g). When expressed as percentages, the mean errors were $0.48 \pm 0.37\%$ for the training set, $4.45 \pm 1.32\%$ for Test Set 1, $10.13 \pm 2.70\%$ for Test Set 2, and $11.00 \pm 6.39\%$ for Test Set 3.

To understand which eAP features are most important in predicting iAP features, we utilized SHapley Additive exPlanations (SHAP) summary plots (Fig. 2h, i). These plots rank features by their influence on APD predictions, quantified as the average absolute shift in predicted APDs caused by variations in each feature. Our analysis indicates that $\Delta T_d$, $\Delta V_1/\Delta V_2$, $\Delta T_s$, and increase rate (IR) (see Fig. 2b) are the most important predictors of APD values (Fig. 2h). These results agree with previous qualitative relationships shown between eAP features and iAP features[18]. Furthermore, as shown in Fig. 2i, we identified the top three eAP features locally affecting APD 30, 50, 70, and 90 predictions (reflecting 30, 50, 70, and 90% cell repolarization levels) (Fig. 2i). For APD 70 and 90, $\Delta T_d$, increase rate (IR), and $\Delta T_s$ were found to be crucial predictors. Higher $\Delta T_d$ and, to a lesser extent, $\Delta T_s$ predict increased APD values, whereas a lower IR suggests lower APD values. $\Delta T_d$ plays the same role concerning APD 30 and 50. Furthermore, for APD 30 and 50, $\Delta T_s$ and the ratio $\Delta V1/\Delta V2$ are the other most significant influencers. Interestingly, for both APD 30 and APD 50, higher $\Delta T_s$, as opposed to APD 70 and APD 90, are associated with lower APD values. These new relationships revealed between eAP and iAP features were identifiable only because of the availability of our unique dataset, consisting of several eAP and iAP pairs. Having identified eAP features as accurate predictors of iAP features, we asked if the entire iAP waveform could be reconstructed from eAP using deep learning.

## Deep learning for reconstructing iAPs from eAP

Deep learning algorithms excel at processing and analyzing high-dimensional data (i.e., data with many features or variables), as is the case with our eAP and iAP signals. In our efforts to reconstruct the entire iAP waveform from the corresponding eAP, we developed a modified Attention-Residual-Block UNET model, enhanced with a pseudo-physics loss function. Our Attention Physics informed UNET (PIA-UNET) model (Fig. 3a) consists of an encoder and decoder[50] where the encoder compresses eAP features and the decoder reconstructs the iAP. Furthermore, our model incorporates a modified version of the Aliev-Panfilov[51–55] model as the foundational physics component (see "Methods") (Fig. 3a). The modified Aliev-Panfilov model, both simple and efficient, describes cardiac electrophysiology by depicting it as a traveling excitation wave followed by a non-excitable (refractory) region. We consider Aliev-Panfilov as a "pseudo physics" function in our study, owing to its phenomenological approach to mimic action potentials. Our goal with the integration of physics-informed loss functions was to ensure predictions that are not only data-consistent but also physics-plausible. This approach helps prevent unrealistic predictions when working with new eAP recordings from different devices, maintaining robustness across varying data sources. The model captures iAP waveforms and prevents super-repolarization (a brief undershoot at the end of repolarization) in the system.

The successful reconstruction of normalized iAP waveforms by our PIA-UNET is demonstrated through representative overlays between reconstructed and actual iAPs for four distinct waveforms in our test sets (Fig. 3b). This comparative visualization highlights the model's proficiency in accurately capturing the nuanced shapes of iAPs across a diverse spectrum of eAP shapes. Incorporating a pseudo-physics function into the model's loss function corrected the slight undershoot observed in NEA iAPs (Fig. 3b for Test 2 data) and enhanced generalizability, particularly in Test 3 (MEA data) (Fig. 3b for Test 1 and SI-Table 1). The model's predicted values for membrane

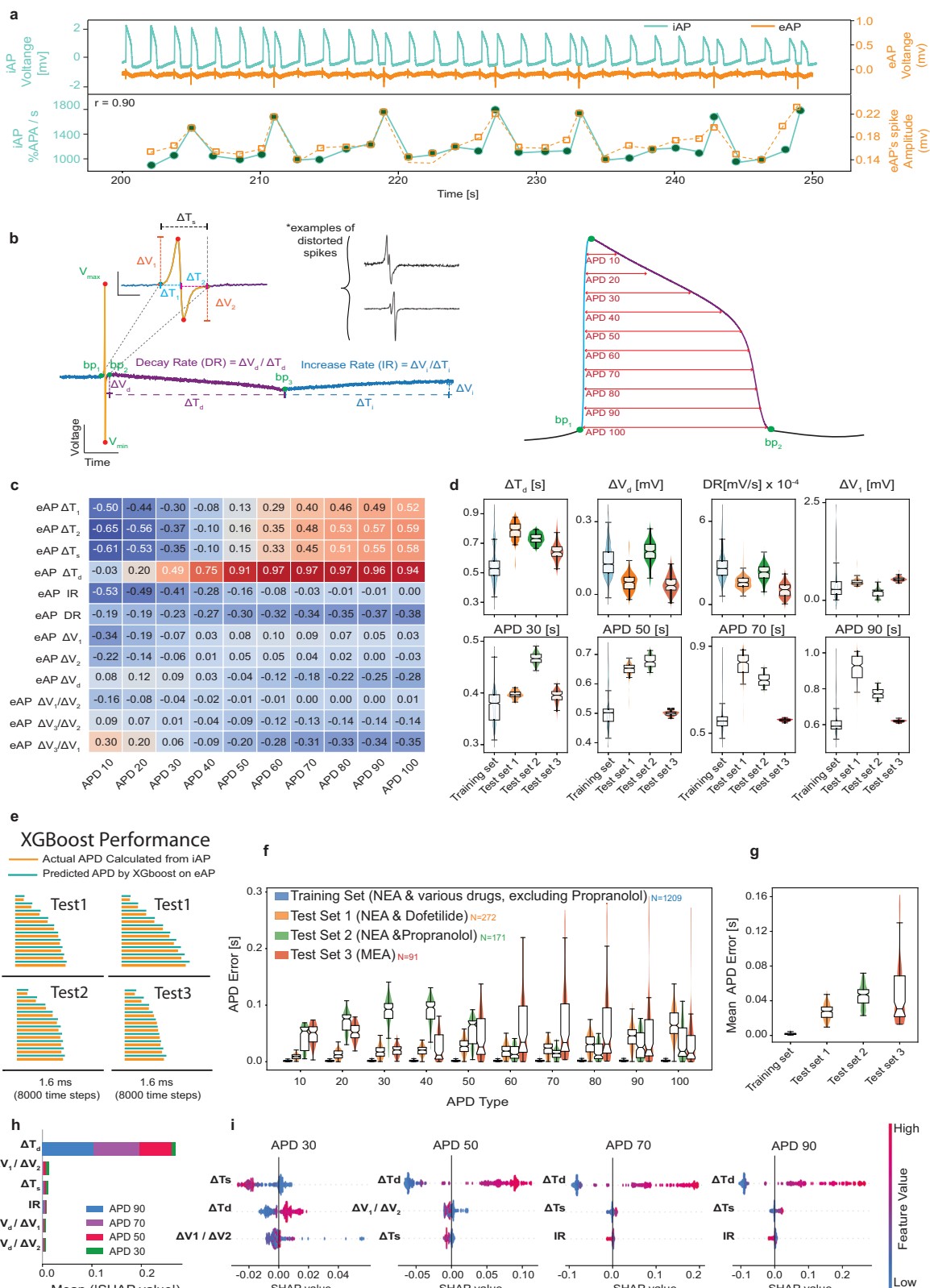

potentials across the three test sets align closely with the actual values, demonstrating a high correlation (*r* = 0.99) (Fig. 3c). In addition, the model exhibits an MAE of 0.041 ± 0.008 on Test 1, 0.049 ± 0.017 on Test 2, and 0.039 ± 0.009 on Test 3, while the MAE on the training set remains lower at 0.023 ± 0.014 (Fig. 3d). Furthermore, the average APD errors were less than 6.35% for APD70 and APD90, and less than 5.75% for APD30 and APD50 (Fig. 3e). The mean absolute APD error from

APD10 to APD100 was low at 0.017 ± 0.016 s (3.99 ± 3.41%) for the training set, and 0.037 ± 0.008 s (6.16 ± 3.41%) for Test 1, 0.021 ± 0.022 s (3.52 ± 3.23%) for Test 2, and 0.025 ± 0.011 s for Test 3 (5.21 ± 2.22%) (Fig. 3f). Altogether, these various error metrics demonstrate our model's ability to accurately reconstruct iAP waveforms from eAP recordings. While we have shown that eAP recordings are good predictors of iAP, we posit that a certain level of fidelity in the

**Fig. 2 | Quantitative relationships between eAPs and iAP waveform features. a** A representative simultaneous recording of eAP and iAP from arrhythmic cells (top) demonstrates a strong correlation ($r = 0.90$) between eAP amplitude (maximum voltage during the spiking phase [mV]) and iAP spike velocity (percentage change in iAP voltage over time [% change in iAP/s]). This association is further evident as oscillations in the extracellular recordings reflect the action potential's repolarization phase. **b** Key descriptors to characterize eAP and iAP waveforms and examples of distorted eAPs. The details are described in the Methods section. **c** A correlation analysis between eAP features and iAP APD values on undistorted eAP/iAP pairs ($n = 1049$). **d** The distributions of normalized eAP features alongside iAP features ($n_{training} = 1512$, $n_{Test1} = 272$, $n_{Test2} = 171$ and, $n_{Test3} = 91$). **e** Examples of XGBoost predicted and actual APD lines for the three test sets. **f** The distribution of prediction errors for APD values across the training set and test sets ($n_{training-val} = 1209$, $n_{Test1} = 272$, $n_{Test2} = 171$ and $n_{Test3} = 91$). **g** Distribution of mean APD errors for Test1 ($0.020 \pm .007$ s), Test2 ($0.040 \pm .006$ s), Test3 ($0.047 \pm .038$ s), and the training set ($0.002 \pm .001$ s) for $n_{training-val} = 1209$, n

$_{Test1} = 272$, $n_{Test2} = 171$ and, $n_{Test3} = 91$. **h** SHAP values identify $\Delta T_d$, $\Delta V_1 / \Delta V_2$, $\Delta T_s$, $\Delta V_1$, and IR as the most significant eAP features for predicting iAP features. SHAP values illustrate how varying these features affects the predicted values relative to the model's average output, with feature importance defined as the average of absolute changes imposed on predictions by varying feature values. **i** The ranked significance of eAP signal features in predicting APD values according to partial dependency plots. Each dot represents a single predicted APD value, with its color indicating the feature's value and its position on the $X$-axis showing its SHAP value, reflecting the expected deviation in APD prediction. For example, in the APD90 plot, increasing $\Delta T_s$ (transitioning from blue to red) results in higher predicted APD90 values, whereas $\Delta T_s$ and APD30 exhibit the opposite trend. Note that SHAP values may be influenced by local minima, potentially limiting their representation of the global relationship between features. The box plots show the median (center line), interquartile range (IQR; box bounds), whiskers ($1.5 \times$ IQR), and outliers (points beyond whiskers).

eAP recording is essential for accurately encoding the information needed to reconstruct the iAP waveform. For instance, if the eAP signal lacks the detailed features depicted in Fig. 2b, the precision of our model in predicting the corresponding iAPs is expected to decrease due to the absence of necessary information within the eAP signal. Consistent with this, our PIA-UNET model demonstrated reduced accuracy for eAP signals with low amplitudes of the positive eAP spike ($\Delta V_1$) and high noise, as illustrated in Fig. 3g. As previously indicated (Fig. 2a), lower eAP spike amplitude may result from a biological phenomenon related to reduced spiking velocity of the iAP or poor cell/electrode coupling. In any scenario, an eAP signal deficient in essential information compromises the model's ability to accurately reconstruct the iAP signal. The increased MAEs associated with these findings, along with the rigorous model training design, strengthen our confidence that the model is effectively learning the relationship between eAP and iAP. PIA-UNET outperforms the feature-based XGBoost in action potential analysis. While XGBoost is limited to predicting APD values and requires extensive feature engineering, PIA-UNET reconstructs entire iAP shapes directly from raw eAP data, providing a more comprehensive understanding of cardiac electrophysiology. PIA-UNET also handles noisy or distorted data more effectively and demonstrates better generalization across diverse test sets, minimizing the risk of overfitting.

**Application of our proposed system for high-throughput multi-channel assessment of drug induced cardiotoxicity**

The distinctive ability of our model to accurately reconstruct iAP from non-invasive eAP recordings can hold significant potential for applications in electrophysiology. One such application is within the context of the CIPA initiative, aimed at developing improved in vitro models for more accurate evaluation of cardiotoxicity using stem-cell-derived cardiomyocytes. Our proposed non-invasive electrophysiology approach can detect subtle changes in the iAP waveforms, such as the repolarization prolongation of stem-cell-derived cardiomyocytes in the presence of a cardiotoxic drug (dofetilide) (Fig. 4a, b). As shown in Fig. 4a, the drug dosage can be added during a long-term non-invasive eAP recording, and the iAP can be reconstructed at any given time point before or after drug administration (Fig. 4b). Furthermore, our model can report on changes in any desired iAP feature including APD values throughout the recording as shown in Fig. 4c. Acknowledging the importance of addressing uncertainties in our iAP predictions, which is critical for drug screening applications, we also incorporated a confidence interval that accompanies our iAP predictions (Fig. 4b, c). The 90% confidence interval was evaluated by reconfiguring the final layer of our model to yield three outputs: $V_{0.05}$, $V_{0.5}$, and $V_{0.9}5$ which represent the 0.05, 0.5, and 0.95 quantiles (see "Methods"). Building on our system's ability to detect drug-induced variations in iAP waveforms at the single-channel level, we extended

our work to include long-term, multi-channel parallel recordings of eAPs. This was followed by the reconstruction of iAPs across multiple cells in a network of stem-cell-derived cardiomyocytes (Fig. 4e). Using this approach, the drug-induced effects as indicated by changes in APD values, can be non-invasively monitored over extended periods for several cells simultaneously (Fig. 4f). Using this high-throughput approach, we conducted population-level analyses to identify APD changes (Fig. 4g), a key parameter for cardiotoxicity assessment[8,56] and drug screening[8,56]. Our method enables detailed observation of variations within the cardiomyocyte population, facilitating in-depth cardiac electrophysiology studies at both individual cell and broader population levels.

In this work, we presented a non-invasive, intelligent electrophysiology technique that combines two recent advancements: (1) nanoelectrode arrays, which can simultaneously record intracellular and extracellular signals from thousands of interconnected cells[20,22–24,35], and (2) PIA-UNET, which enables fast and precise reconstruction of iAP signals. Using state-of-the-art nanoelectrode arrays, we gathered a unique dataset of thousands of diverse iAP and eAP pairs from monolayers of human stem-cell-derived cardiomyocytes. Through this dataset, we uncovered new relationships between eAP and iAP features and developed a physics-informed deep learning model to accurately reconstruct iAP waveforms from eAP signals. We evaluated the performance and generalizability of our model, trained on NEA iAP/eAP pairs, using eAPs recorded from NEAs exposed to a drug from the training set, a different unseen drug, and eAPs from commercial MEA recordings. In addition, we demonstrated the technique's utility for high-throughput, long-term monitoring of proarrhythmic drug effects at both single-cell and population levels. While our study is limited by the size and diversity of our dataset, the potential for future research in intracellular electrophysiology is vast. Expanding the dataset to encompass a wider range of electrogenic cell types and drug interactions will not only refine the accuracy of our findings but also broaden their applicability. This expansion is a key step toward developing more robust and comprehensive models, which we aim to share with the wider scientific community.

## Methods
### Error definition

$$MAE = \frac{1}{n} \sum \left| Potential_{actual} - Potential_{predicted} \right| \tag{1}$$

$$MSE = \frac{1}{n} \sum \left( Potential_{actual} - Potential_{predicted} \right)^2 \tag{2}$$

$$APD\,i\,Error\,(s) = \left| APD\,i_{actual} - APD\,i_{predicted} \right| \tag{3}$$

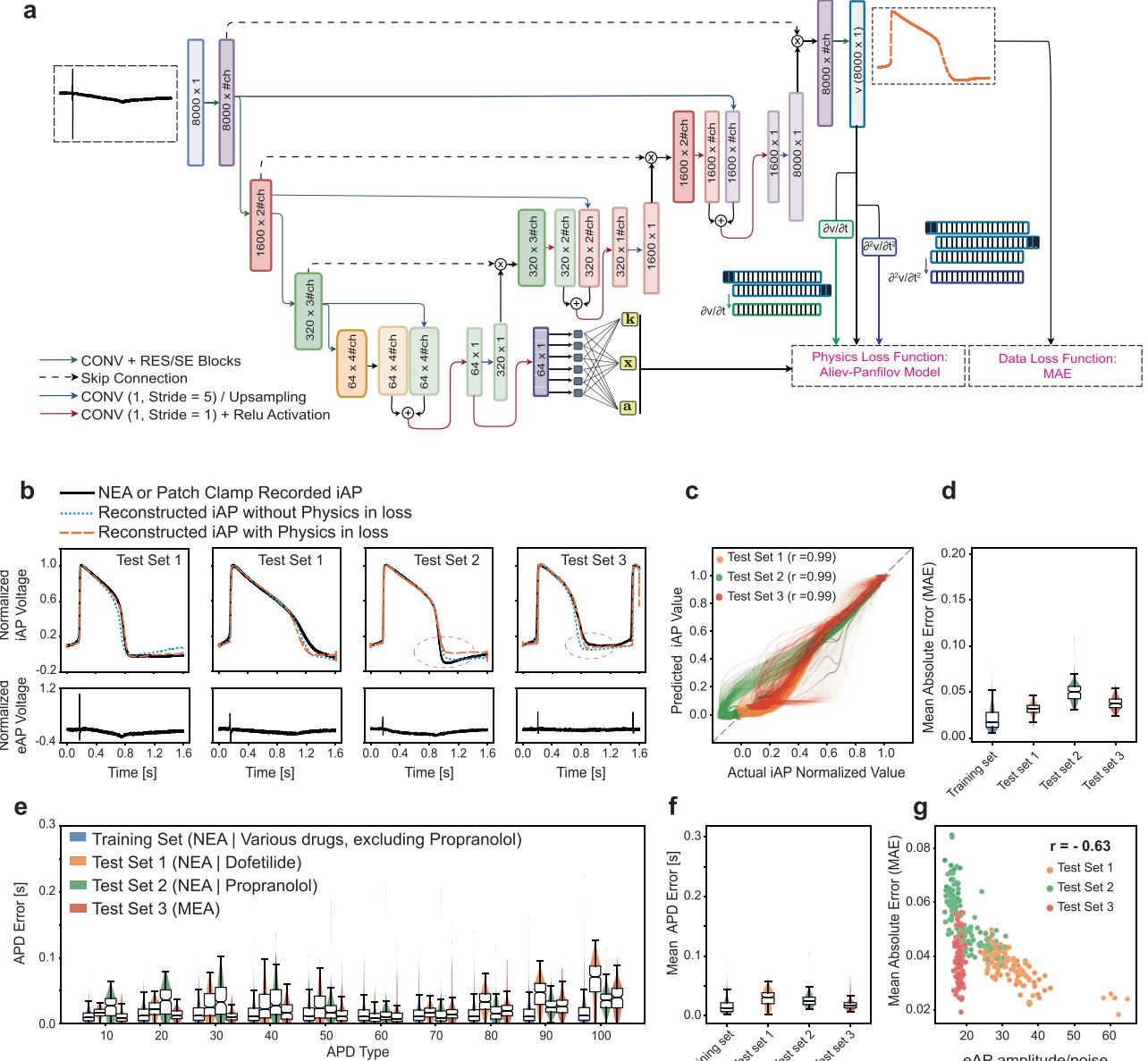

**Fig. 3 | Physics-Informed Attention Unet (PIA-UNET) to reconstruct the entire iAP waveform from the eAP signal. a** PIA-UNET Model Schematic: A visual representation of the PIA-UNET model used in the study. **b** Comparison of reconstructed iAPs using PIA-UNET with and without physics-informed loss functions, alongside the actual iAPs on the three test sets. Incorporating physics corrected the brief undershoot at the end of repolarization in some NEA iAP recordings (Test 2) and resulted in more accurate predictions on eAPs recorded with the MEA (Test 3). **c** This panel shows the model's predicted potential values closely aligning with the actual values ($r = 0.99$) calculated across all 8000 points from $n = 421$, 187, and 149 samples for the three respective test sets. **d** The model exhibits a MAE of $0.041 \pm 0.008$ on Test 1, $0.049 \pm 0.017$ on Test 2, and $0.039 \pm 0.008$ on Test 3. The MAE on the training set is $0.022 \pm 0.014$. The higher MAE on Test 2 is attributed to the correction of NEA iAP through the incorporation of physics into the loss

function. **e** The distribution of APD errors for predicted iAP values across the training set and test sets ($n_{training} = 1842$, $n_{Test1} = 421$, $n_{Test2} = 187$ and $n_{Test3} = 149$). **f** The mean APD error from **APD10 to APD100** was **$0.037 \pm 0.008$ s ($6.17 \pm 3.41\%$)** for **Test 1**, $0.021 \pm 0.022$ s ($3.53 \pm 3.23\%$) for **Test 2**, and **$0.025 \pm 0.011$ s ($5.21 \pm 2.21\%$)** for **Test 3**, with the training set showing a lower error of **$0.017 \pm 0.016$ s ($3.99 \pm 3.41\%$)** for $n_{training} = 1842$, $n_{Test1} = 421$, $n_{Test2} = 187$ and, $n_{Test3} = 149$. **g** The MAE comparison on the test set versus eAP shows that the eAP waveform amplitude/noise ratio is the primary factor influencing prediction error. Signals with lower noise levels are expected to be reconstructed more accurately. This ratio is calculated by dividing the eAP maximum value by the noise level. The box plots show the median (center line), interquartile range (IQR; box bounds), whiskers ($1.5 \times$ IQR), and outliers (points beyond whiskers).

$$APD i \text{ Percentage Error}(\%) = \frac{|APDi_{actual} - APDi_{predicted}|}{APDi_{actual}} \times 100 \quad (4)$$

$$\text{Total APD Error} = \sum_{i \in \{10, 20, \ldots, 100\}} |APDi_{actual} - APDi_{predicted}| \quad (5)$$

## Fabrication of nano electrode arrays

The fabrication process is fully explained in the paper by Jahed et al.[35]. Briefly, we used maskless photolithography followed by deep reactive ion etching to develop vertical SiO2 nanopillars, which were then coated with Pt metal to achieve conductivity. The metal was etched from the tip of the pillars using a directional dry etch to achieve the nano crown shape.

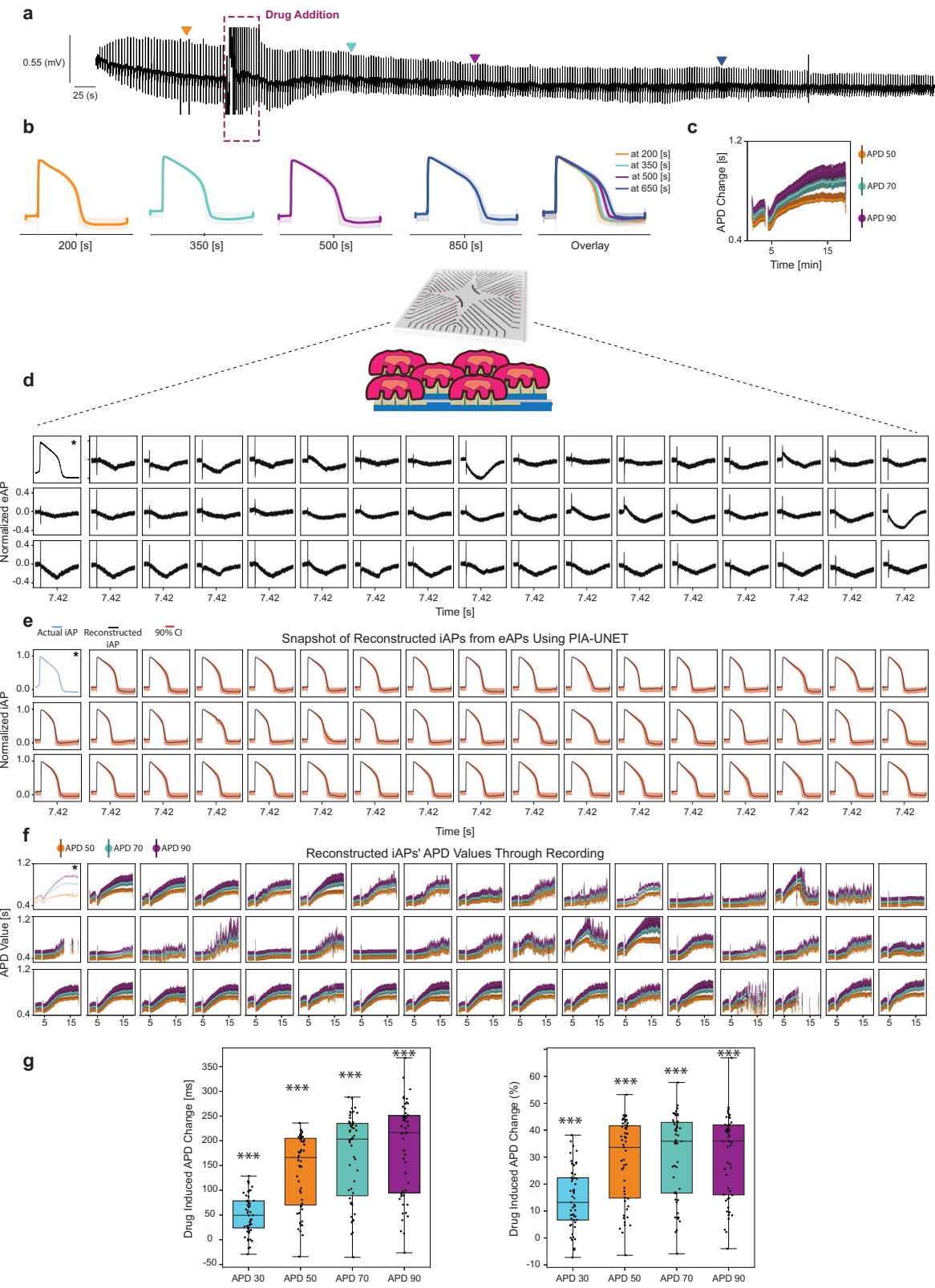

## hiPSC-CM differentiation and characterization.

The non-commercial hiPSC-CMs used in this study were differentiated in the laboratory of Prof. Joseph C. Wu, at the Stanford Cardiovascular Institute using highly standardized protocols described previously in detail by Burridge et al. [35,57,58]. Briefly, hiPSCs (line SCVI-273) were treated with 6 μM CHIR99021 (Selleck Chemical) in RPMI supplemented with B27 without insulin for 2 days, followed by recovery in RPMI supplemented with

B27 without insulin for 1 day, and then by treatment with 5 μM IWR-1 (Selleck Chemical) for 2 days. After recovery in fresh RPMI plus B27 without insulin medium for 2 days, cells were switched to RPMI plus B27 with insulin for 2 days. The hiPSC-CMs were purified with glucose-free RPMI plus B27 with insulin medium for 2–4 days and maintained in RPMI plus B27 with insulin medium for subsequent experiments. Using patch clamp techniques, these cells were demonstrated to be

**Fig. 4 | Demonstration of high-throughput pharmacology by reconstructing iAPs from multi-channel eAP recordings and predicting the drug dose response of cardiac myocytes. a** An example of a one-channel recording for an extended duration, with 10 nM of Dofetilide added to the dish around $t \approx 300$ s. **b** Reconstructed iAPs using QPIA-UNET from eAP recording samples at 200, 350, 500, and 850 s. The solid line represents the $q = .5$ prediction; the shaded band between the $q = .05$ and $q = .95$ quantiles indicates the model's prediction uncertainty. **c** Changes in APD50, APD70, and APD90 values over time, calculated from the reconstructed iAP signals at corresponding sample times. Central points represent APD values from the $q = .5$ predictor; vertical error bars indicate uncertainty spanning the range between the $q = .05$ and $q = .95$ quantiles. **d** Snapshot of simultaneous eAP recordings from 50 channels on NEA at $t \approx 445$ s (7.42 mins), along with iAP recordings from the neighboring channel next to the second channel on NEA using electroporation (*). **e** Reconstructed iAPs from simultaneous eAP recordings across 49 channels on NEAs, captured at $t = 445$ s, with the actual iAP recording (*) from the neighboring channel next to the second channel. The solid line shows the iAP predicted by QPIA-UNET with $q = .5$; the shaded band between the $q = .05$ and $q = .95$ quantiles indicates the model's prediction uncertainty. **f** Changes in APD values (50, 70, and 90) calculated from reconstructed iAPs, showing variation over the recording period (~ 19 min), with 10 nM of Dofetilide added to the dish around $t \approx 300$ s. Central points represent APD values from the $q = .5$ predictor; vertical error bars indicate uncertainty spanning the range between the $q = .05$ and $q = .95$ quantiles. **g** Box plot distribution illustrating maximum drug-induced changes in APD at 40, 50, 70, and 90% repolarization levels after exposure to 10 nM Dofetilide, expressed both in milliseconds and as a percentage ($n = 50$). A one-sided $t$ test was conducted to assess if the changes in APD are greater than zero.*** indicates $p$-value < 0.001 compared to no change. Observed $p$-values: $2.02 \times 10^{-13}$, $1.04 \times 10^{-17}$, $4.09 \times 10^{-18}$, and $2.81 \times 10^{-18}$. The box plots show the median (center line), interquartile range (IQR; box bounds), whiskers ($1.5 \times$ IQR), and outliers (points beyond whiskers).

heterogeneous with ventricular-like cells being the predominant (57%) along with atrial-like and nodal-like cells[57]. The chemically defined differentiation method used in the Wu lab has repeatedly shown to provide a reproducible and scalable method for deriving cardiomyocytes from hiPSCs[58]. Furthermore, results from independent labs suggest that hiPSC-CMs derived from iPSC lines using this protocol can be consistently recovered after cryopreservation, and demonstrate comparable and functional sarcoplasmic reticulum calcium handling[59]. The commercial hiPSC cardiomyocytes were purchased from Celogics (Celo.Cardiomyocytes, Celogics, CAT # C50) and used as an additional test set due to their low variability and high Sensitivity to electrophysiological changes from drug treatment as shown by the vendors). Previous characterization of these cells by patch-clamp and fluorescent microscopy demonstrated their Human cardiomyocyte-like electrophysiology, as well as ventricular cardiomyocyte markers and structural characteristics.

**hiPSC-CM culture on NEA and MEA devices.** Prior to plating cells on NEA devices, the device was coated with 1 mg/ml poly-L-Lysine at room temperature for 15 min, then treated with 0.5% Glutaraldehyde in PBS at room temperature for 10 min, followed by 1:200 Matrigel in DMEM/F12 at 37 °C for 3 h before seeding cells. The cultured hiPSC-CM were disassociated from the plate with TryPLE select 10X at 37 °C for 5 min after 25–60 days of differentiation. Cells were resuspended in a culture medium supplemented with 10% KnockOut Serum Replacement (KSR), then seeded at ~ $1.2 \times 10^5$ cells/device. Measurements were taken for over a month from 5 days post cell attachment. For MEA recordings, commercial hiPSC-ventricular cardiomyocytes (Celo.Cardiomyocytes, Celogics, CAT # C50) were used. The cells were thawed and prepared according to the manufacturer's protocol. These Celo.-cardiomyocytes were derived from a proprietary hiPSC line (fibroblast, Caucasian male donor) and exhibited spontaneous beating starting on day 2 post-thaw, with stabilization at 45–60 beats per minute (bpm) by day 7. Studies by the manufacturer show that Celo.Cardiomyocytes formed synchronous monolayers and expressed ventricular cardiomyocyte-specific markers, such as connexin 43, indicating high interconnectivity. These cells also demonstrated physiologically relevant electrophysiological properties, as evidenced by their response to various ion channel modulators (Celo.Cardiomyocytes. Celogics https://www.celogics.com/celocardiomyocytes).

**NEA and MEA recordings.** For both NEA and MEA recordings, measurements were taken at 32 °C in RPMI with B27 and 10 mM HEPES. This temperature was chosen because it keeps the membrane pores, which form during electroporation, open for longer iAP recordings compared to 37 °C. HEPES was used to stabilize the pH in the room atmosphere. Recordings were taken using a 60-channel voltage amplifier (MEA1060-Inv-BC, Multi-Channel Systems, Reutlingen, Germany), with a sampling rate of 5 kHz for NEA recordings and 10 kHz for

MEA recordings. For NEA recordings, electroporation was achieved by delivering a ± 4 V biphasic square wave with a 200 µs duration at each phase at each electrode. For MEA recordings, commercial MEAs (60MEA100/10iR-Ti) from Multichannel Systems, powered by Harvard Bioscience, Inc., were sterilized and prepared for cell culture. The MEAs first underwent UVO treatment for 10 min using a UVO-CLEANER, Model 42 (Jelight Company, Inc.). Following this, the MEAs were washed three times with phosphate-buffered saline (PBS) and then washed twice with 70% ethanol inside a biosafety cabinet (BSC). The devices were air-dried for 30 min to 1 h and subsequently sterilized under ultraviolet (UV) light in the BSC for 1 h. After sterilization, the MEAs were coated with fibronectin (Millipore Sigma, F1141-1MG) at a concentration of 50 µg/mL (1:20 dilution). A droplet of 8 µL of the diluted fibronectin was applied to the center of the MEA, covering the electrodes, and incubated for at least 1 h at 37 °C. Cells were seeded onto the fibronectin-coated MEAs at a density of 50,000 cells in 8 µL. The devices were then incubated at 37 °C for 60–90 min, followed by the slow addition of 400 µL of plating medium (Celogics, CAT # C50-PM, supplemented with C50-PS) while the MEAs were positioned at a 30° angle. The MEAs were then returned to a flat position and incubated for 24 h before replacing 100% of the plating medium with advanced medium (Celogics, CAT # CM200, supplemented with C50-MS). Media changes of 50% were performed every 48 hours thereafter. By day 5, the hiPSC-derived ventricular cardiomyocytes on the MEAs were ready for electrophysiology experiments.

**Drug experiment.** *Nifedipine (Sigma), Quinidine (Sigma, Q3625), Propranolol (Sigma, P0884), Lidocaine (Sigma, L7757), and Flecainide (Sigma, F6777) were dissolved in DMSO to make 100 mM stock solution; Dofetilide (Sigma) was dissolved in DMSO to make 10 mM stock solution.* For each dose, the stock drug was diluted in measurement medium into 2x of the targeted dose and warmed up to 32 °C. Upon drug administration, 500 µl of the 2x dose drug was added to the 500 µl existing measurement medium to help homogeneous diffusion. The cells were electroporated 1 min after the start of measurements; the drug was administered 400 s after electroporation; and the recording lasted for ~30 min for each repeat. The drug's effect on action potential and the corresponding iAP waveforms are provided in (Supplementary Note III).

**Obtaining a diverse dataset of eAP and iAP pairs.** We tried two approaches for data collection: incremental drug addition and single high-dose drug application. In the first approach, we added the drugs in multiple steps to capture a spectrum of iAP shapes and their corresponding eAPs. However, we also applied a relatively high concentration of the drugs in a single step during each recording session (Fig. 1g), which enabled us to achieve a wider range of iAP durations (Supplementary Fig. 2a). The single high-dose approach minimized the noise typically associated with incremental drug addition. Given that

both amplitude and S/N ratio tend to decrease over time—resulting in significantly lower S/N ratios and amplitudes in the final traces compared to the initial ones (Supplementary Fig. 2b–2e, with paired $t$ test $p$-values of $8.6 \times 10-15$ and $1.1 \times 10-8$, respectively)—this strategy allowed us to collect high S/N ratio data at higher drug dosages. Moreover, the APD values changed gradually until stabilization (Supplementary Fig. 2h), providing a spectrum of iAPs with APD values ranging from normal to fully impacted by the drug. Consequently, this approach, similar to multi-step drug addition, resulted in a diverse range of eAP and iAP pairs, as shown in Supplementary Fig. 2h, 2i, it facilitated the acquisition of longer APDs in a shorter time with improved S/N ratios while allowing the APD values to change incrementally.

**Bandpass filtering.** The data underwent bandpass filtering to selectively isolate the frequencies of interest, utilizing an acausal third-order Butterworth filter renowned for its flat frequency response within the passband. With a sampling frequency set at 5000 Hz, the filter coefficients were designed for a low-cut frequency at 0.1 Hz and a high-cut frequency at 2499 Hz. Furthermore, to extract the signal's noise, the signal was subjected to a filter between 2499 Hz and 4000 Hz, and the standard deviation of this filtered signal was taken. However, for both training and validation purposes, raw eAP and iAP signals are used directly.

**Data filtering and segmentation.** The initial step in data processing involved identifying high-quality pairs of neighboring cell recordings for inclusion in our dataset. To determine the action potential peaks in intracellular recordings, we employed the peak-finding algorithm from the *scipy.signal* library. This algorithm was configured to identify peaks with a minimum height of twelve times the noise level and a separation distance of at least 60% of the average action potential period. Once the peaks were identified, we segmented the data around these points. 8000 points (equivalent to 1.6 s), starting 1000 points (or 0.2 s) before the identified peak. Following segmentation, we applied stringent selection criteria for inclusion in our analysis. Only segments exhibiting a high signal-to-noise for intracellular channels (> 90 dB) were considered. In addition, segments were required to have an action potential amplitude of at least 0.5 mV in the intracellular channel.

**Characterizing the eAP and iAP waveforms.** To characterize eAP waveforms, five critical points where the waveform gradient changes significantly were identified: just before the spike ($bp_1$), maximum point ($V_{max}$), minimum point ($V_{min}$), right after the minimum point ($bp_2$), and where it starts to rise again ($bp_3$). The vertical span from $bp_1$ to $V_{max}$ is labeled $\Delta V_1$, and from $V_{min}$ to $bp_2$ is $\Delta V_2$. Similarly, $\Delta V_d$ is the vertical distance from $bp_2$ to $bp_3$. To describe the horizontal dimensions within the spike, $\Delta T_1$ and $\Delta T_2$ describe the span of the positive and negative spike phases, respectively, and $\Delta T_d$ denotes the distance between $bp_2$ and $bp_3$. Decay Rate (DR, mV/s) and Increase Rate (IR, mV/s), represent the slope from $bp_2$ to $bp_3$ and just after $bp_3$, respectively. Our action potential duration (APD) metrics, specifically APD 10 to APD 100, measure the timeframe for an action potential to decrease to 10% and recover to 100% of its peak value, providing insight into the waveform's recovery phase independent of amplitude variations. Further details are provided in (Supplementary Note IV).

**Signal to noise ratio (S/N) calculation.** To calculate the S/N for iAPs, we initially employ filtering techniques on the raw recording. A low-pass filter is used to isolate the signal component, while a high-pass filter helps in extracting the noise component. After filtering, we segment these filtered components into windows of length $d$. This results in arrays for the signal, $\mathbf{V} = [v_1, v_2 ..., v_d]]$, and for the noise, $\mathbf{N} = [n_1, n_2 ..., n_d]$. Following the segmentation, we calculate the Root Mean Square of the Signal $RMS_{Signal}$ and the Root Mean Square of the

Noise $RMS_{noise}$ as follow:

$$\text{RMS}_{signal} = \sqrt{\frac{1}{d} \sum_{i=1}^{d} v_i^2} \qquad (6)$$

Similarly, the $RMS_{noise}$ is calculated as:

$$\text{RMS}_{noise} = \sqrt{\frac{1}{d} \sum_{i=1}^{d} n_i^2} \qquad (7)$$

Finally, the Signal to Noise Ratio in decibels (S/N [dB]) is determined using the formula:

$$\text{S/N}_{iAP}[dB] = 20 \log\left(\frac{\text{RMS}_{signal}}{\text{RMS}_{noise}}\right) \qquad (8)$$

For eAP signals, we calculated $S/N$ as the ratio between the maximum amplitude in the signal array ($\mathbf{V}$) and the standard deviation of the noise array ($\mathbf{N}$).

$$\text{S/N}_{eAP} = \left(\frac{\max(\mathbf{V})}{\text{std}(\mathbf{N})}\right) \qquad (9)$$

**eAP and iAP normalization.** The normalization is crucial to bring all data to a comparable scale and to emphasize relative changes in signal features over absolute values. The windows of iAPs are normalized to a range between 0 and 1 starting from 0.1. For eAPs, a distinct approach is used for normalization. A specific segment of the eAP signal, particularly the values between indices 1150 and 1350, is selected. This segment is critical as it characterizes the noise within the eAP signal based on the assumption that it accurately represents the noise characteristics of the entire signal. The peak index in each window is typically around index 1000, just before this segment. To quantify the noise, the function calculates the standard deviation (σ) of this selected segment, which serves as a measure of variation or dispersion within the values. Next, the eAP signal undergoes normalization by subtracting the mean of the entire eAP signal from each value. To ensure the normalization starts from 0, the initial value of the eAP signal is subtracted from all subsequent values. The final step scales the signal relative to its noise level, achieved by dividing the mean-adjusted values by 60 times the calculated standard deviation. The normalization methods applied to eAP and iAP arrays given that raw eAP and iAP are arrays of values $[x_1, x_2 ..., x_{8000}]$ are as follows:

$$\text{iAP}_{Normalized}(x_i) = 0.1 + 0.9\left(\frac{\text{iAP}_{raw}(x_i) - \text{iAP}_{raw}(x_1)}{\max(\text{iAP}_{raw}) - \text{iAP}_{raw}(x_1)}\right) \qquad (10)$$

$$\text{eAP}_{Normalized}(x_i) = \frac{\text{eAP}_{raw}(x_i) - \text{mean}(\text{eAP}_{raw})}{60\,\sigma(\mathbf{N})} - \text{eAP}_{raw}(x_1) \qquad (11)$$

## PIA-UNET architecture
UNET is an autoencoder architecture often used in biomedical applications for image segmentation and data reconstruction; here, we provide the detailed structure of our attention to UNET[60].

**Input.** A vector with dimensions (batch size × 8000 × 1)

**Convolution-batchnormalization-ReLU (CBR) block.** The CBR block forms the basic building block of our architecture. It incorporates a 1D convolution layer followed by a batch normalization layer and a ReLU activation function. The "*He*" normal initialization method is employed for the convolution layers.

**Squeeze-and-excitation (SE) block.** Within residual blocks (Res-blocks), SE blocks are to recalibrate channel-wise feature responses adaptively. The SE mechanism acts to recalibrate the preliminary

features by adaptively reweighting the channel-wise feature responses. It accomplishes this by performing global average pooling, dimensionality reduction (by a factor of 8), and subsequent scaling using a two-layer fully connected network with ReLU and sigmoid activations, respectively on Resblock input. These scaling factors are then used to reweight the original Resblock output.

**Residual block (Resblock).** Nested within the architecture, each resblock serves as a mechanism for efficient feature extraction and transformation. Each Resblock starts with two consecutive Convolution-BatchNormalization-ReLU (CBR) blocks, first applying a ReLU activation, then a Sigmoid activation. An SE block then refines this output by providing adaptive weights, helping the model focus on important features while ignoring the less relevant ones. The function concludes its operations by integrating the original input into this recalibrated output, forming a residual connection. This approach provides a balanced and refined feature representation and allows for the construction of deeper architectures without information loss.

**Attention mechanism.** The attention mechanism is to enhance the feature representations in the decoder by considering features at the corresponding level in the encoder. First, the input from a lower decoder level is upsampled by a factor of 5, then given the upsampled input from a lower decoder level and a shortcut connection from the encoder, both are transformed via 1D convolutions and summed. A ReLU activation is applied to this sum, which is then processed through another 1D convolution with a sigmoid activation. This forms the attention mask, which is multiplied by the encoder feature map.

**Encoder.** The encoder initiates with a CBR block and progresses through a sequence of eight Resblocks. With each stage, feature maps are condensed by a factor of five using strided convolutions, enhancing the receptive field while aggregating spatial information. Parallelly, the depth of these feature maps escalates, beginning at 32 channels beginning at 32 and adding 32 more at each step, reaching 32*4 by the last step. This simultaneous contraction of spatial resolution and channel expansion ensures a rich representation of features. Moreover, attention mechanisms at every encoder level capture crucial spatial cues, preparing for the subsequent decoding phase.

**Decoder.** In the decoder phase, spatial resolution is progressively restored across five decoder levels. At each level, feature maps are upsampled by a factor of five through the attention mechanism. These upsampled feature maps are combined with attention-refined feature maps from their corresponding encoder counterparts. Following this fusion, the concatenated feature maps pass through a CBR block, which mirrors the structure of the encoder. Simultaneously, the number of channels decreases progressively, transitioning from 32*4 down to 32*3, then 32*2, and finally 32*1 channel.

**Quantile PIA-UNET with confidence interval (Q-PIA-UNET)**
(Q-PIA-UNET) is a modified version of the PIA-UNET architecture designed to estimate the 90% confidence interval for iAP values using a quantile loss function. The quantile loss function[61] asymmetrically penalizes over- and underestimation, enabling the model to accurately predict specific quantiles (0.05, 0.5, and 0.95) by minimizing errors for each. To address the issue of quantile crossing, the modified architecture simultaneously predicts the 0.05, 0.5, and 0.95 quantiles through three parallel output layers ($V_{0.05}$, $V_{0.5}$, and $V_{0.95}$), along with corresponding physical parameter predictors. Quantile crossing occurs when the predicted lower, median, and upper quantiles are out of order, which makes the predictions unreliable and inconsistent. By using separate output layers corresponding to each quantile in one model, the model ensures that the

predicted values follow the correct order, improving both the accuracy and reliability of the results. The shared hidden layers in the Q-PIA-UNET architecture learn common underlying features from the input data, providing a consistent foundation for all quantile predictions. The separate output layers then specialize in predicting their respective quantiles, fine-tuning the shared representations to capture the unique aspects of each quantile level. Training these quantile predictions simultaneously within the same model allows for joint optimization, where the errors of all quantiles are considered together. This joint learning process implicitly enforces the natural ordering of quantiles, as the model adjusts its weights to minimize discrepancies between quantile levels. As a result, the predictions for the lower quantile (0.05) remain less than or equal to the median (0.5), which in turn remains less than or equal to the upper quantile (0.95), thus preventing quantile crossing.

**Physics informed layer (a, x and k estimation).** In our model, the parameters $k, x,$ and $a$ are estimated from either the bottleneck layer or the layer just before the final layer. This process begins with the convolution of features, where the data is passed through a convolutional layer to extract relevant information. The resulting features are then flattened and processed through a dense layer, which effectively maps the high-level features to the desired parameters. This design allows the model to adapt and learn different sets of parameters for each quantile, accommodating the unique characteristics of each quantile distribution. By following this approach, the model is better equipped to capture the relationships between the input data and the physics-informed parameters.

## Incorporating the Aliev-Panfilov model into the PIA-UNET hybrid loss function
The modified Aliev-Panfilov model for a single cell is as follows:

$$\frac{dv}{dt} = kv(1-v)(v-a) - vw \tag{12}$$

$$\frac{dw}{dt} = \varepsilon(v, a, x)(kv - w) \tag{13}$$

In these equations, v and w represent the normalized iAP and recovery variables, respectively. To ensure the differentiability of the loss function, the step function $\varepsilon(v, a, x)$ is approximated using a sigmoid activation function, which smoothly transitions between values and preserves the model's ability to backpropagate gradients effectively.

$$\varepsilon(v, a, x) \approx x.\sigma(n(a-v)) + (1-x).\sigma(n(v-a)) \tag{14}$$

in which n is a constant determining the sharpness of the stem function. In our approach, we have chosen to set n to be 1000. The parameter $a$ is the excitation threshold, while $k$ controls the magnitude of the transmembrane current. Both space units [s.u.] and time units [t.u.] are dimensionless. The parameter $x$ controls the balance between excitation and recovery dynamics by modulating the smoothness of transitions in the recovery mechanism based on the membrane potential $v$ and the threshold $a$. This flexibility allows the action potential model to adapt to a wider range of iAP shapes. See SI for further details. The loss function typically used in fully connected neural networks aims to reconstruct simulated iAPs from time and coordinates as inputs, leveraging calculable derivatives during propagation to ensure adherence to governing physical equations[62–64]. However, in this case, the input is eAP, and the pseudo-physics loss function is employed to maintain the iAP shape. Derivatives are computed numerically due to the temporal nature of the data arrays. Furthermore, the term $w$ is initially unknown but

can be simplified with certain assumptions in the Aliev-Panfilov equations.

Incorporating the value of $w$ from Eq. (12) into Eq. (13) and deriving it with respect to time (t), we obtain a single equation: for $v \neq 0$:

$$F_{AP}\left(v, \frac{dv}{dt}, \frac{d^2v}{dt^2}, a, k\right) = v^{-2}\left(\frac{dv}{dt}\right)^2 - v^{-1}\frac{d^2v}{dt^2} - \frac{dv}{dt}\left[\varepsilon(v)v^{-1} - k(1 - 2v + a)\right] - k\varepsilon(v)(v^2 - av + a) = 0 \tag{15}$$

The detailed derivation of Eq. 15 is provided in the SI. A small noise (eps = $10^{-3}$) was added to the indices where v or $\frac{dv}{dt}$ = 0. Note that the model processes temporal arrays of eAP potentials of length 8000, outputting temporal arrays of the same length for iAP potentials. The derivatives $\frac{dv}{dt}$ and $\frac{d^2v}{dt^2}$ for array $\mathbf{V} = [v_1, v_2 \ldots, v_d]$ are calculated using discrete numerical methods:

$$\frac{dv}{dt} = \frac{v_{t+1} - v_{t-1}}{2t} \tag{16}$$

$$\frac{d^2v}{dt^2} = \frac{v_{t+1} + v_{t-1} - 2v_t}{t^2} \tag{17}$$

The parameters $v, a$ and $k$ and $x$ can be estimated by compressing the bottleneck or the final layer of the PIA-UNET model, just before the iAP layer, into a single channel using a convolution operation. This is followed by flattening the output and applying a two-layer dense neural network to generate the parameter estimates. The hybrid loss function incorporates terms that account for both alignment with experimental measurements ($L_D$) and adherence to physical laws $L_p$, as described by the pseudo-physics expression in Eq. (15). This loss is computed for N predicted iAP arrays, each with $d$ dimensions ($\mathbf{V} = [v_1, v_2 \ldots, v_d]$), as follows:

$$L_D = \frac{1}{N}\sum_{n=1}^{N}\frac{1}{d}\sum_{i=1}^{d}\left|v_{n,i}^{predicted} - v_{n,i}^{actual}\right| \tag{18}$$

$$L_p = \frac{1}{N}\sum_{n=1}^{N}\frac{1}{d}\sum_{i=1}^{d}\left|F_{AP}\left(v_{n,i}, \frac{dv_{n,i}}{dt}, \frac{d^2v_{n,i}}{dt^2}, a_n, k_n\right)\right| \tag{19}$$

Here, $F_{AP}$ represents the function describing the pseudo-physics relationship with $a$ and $k$ as parameters. The overall hybrid loss function, L, combines these two components, integrating a logarithmic transformation on the physical law adherence term ($L_p$):

$$L = \alpha L_D + \beta \log(L_P) \tag{20}$$

The logarithm function is applied to the physics loss function ($L_p$) due to its monotonic nature. This transformation serves to dampen significant deviations in $L_p$, thereby harmonizing the scales of the physics-based and data-based components of the loss function. This approach ensures a more balanced optimization, considering both experimental data and physical law adherence. The respective weights for the data-based and physics-based components of the loss function are denoted as $\alpha$ and $\beta$ are determined through a grid search hyperparameter tuning technique detailed in the Hyperparameter Tuning and Model Training section.

## Q-PIA-UNET loss function
The quantile loss function for N samples and a given quantile q (0.05, 0.5, or 0.95) is defined as:

$$L_{D,q} = \frac{1}{N}\sum_{n=1}^{N}\frac{1}{d}\sum_{i=1}^{d}\max\left(q\left(v_{n,i}^{predicted} - v_{n,i}^{actual}\right), (q-1)\left(v_{n,i}^{predicted} - v_{n,i}^{actual}\right)\right) \tag{21}$$

For predictions below the actual value, the error is weighted by q, and for predictions above, by 1−q. This asymmetry is beneficial for quantile regression. In addition, the physics term and the total loss functions are:

$$L_{p,a,k} = \frac{1}{N}\sum_{n=1}^{N}\frac{1}{d}\sum_{i=1}^{d}\left|F_{AP}\left(v_{n,i}, \frac{dv_{n,i}}{dt}, \frac{d^2v_{n,i}}{dt^2}, a_n, k_n\right)\right| \tag{22}$$

$$L = \alpha(L_{D,0.05} + L_{D,0.5} + L_{D,0.95}) + \beta\log(L_{P,a1,k1} + L_{P,a2,k2} + L_{P,a3,k3}) \tag{23}$$

## Hyperparameter tuning and model training
Grid search, in conjunction with k-fold cross-validation, was employed for multi-step hyperparameter tuning. Distinctively, in each fold, one pair of neighboring channels was set aside. Five distinct pairs of neighboring channels—sourced from two different recording sets—were utilized for the training and hyperparameter tuning process. The tuning was conducted in a sequential three-step approach: initially, the kernel size, Resblock depth, and the number of channels in Convolution-Batch Normalization-ReLU (CBR) were optimized. Subsequently, the learning rate, number of epochs, and batch size were optimized. The final step involved tuning the weights assigned to the loss functions. The model, trained with these optimal hyperparameters, underwent a two-fold cross-validation. In each fold, the data from one recording set were considered as the validation set. The optimal hyperparameters for the model are set as follows: Initial learning rate at 0.001, number of epochs at 100, number of channels in CBR at 32, kernel size at 11, Resblock depth at 8, batch size at 32, and the weighting factors loss functions, α and β at 10 and 0.01, respectively.

## Action potential durations (APDs) calculation
To accurately determine the APDs, we first applied a smoothing technique to the segmented intracellular traces using a moving average filter (window size = 20) to reduce noise and enhance the detectability of the action potential features. Next, we utilized the standard deviation of the smoothed data to identify the initial upward spike of the action potential. For the quantification of the APDs, we employed the 'peak_widths' method from the SciPy Python package. This method is particularly effective in measuring the widths of action potentials at various levels of repolarization. We specifically focused on obtaining ten distinct APD measurements to comprehensively describe the shape of each action potential. For instance, APD10 and APD20 represent the widths of the intracellular action potential at 10% and 20% of repolarization, respectively.

## Statistical Analysis
All values are expressed as mean ± standard deviation (SD) unless otherwise specified.

## Reporting summary
Further information on research design is available in the Nature Portfolio Reporting Summary linked to this article.

## Data availability

All data supporting the findings of this study are available within the article and its supplementary files. Any additional requests for information can be directed to and will be fulfilled by, the corresponding authors. Source data are provided in this paper.

## Code availability

The source code of the models used in this study is available online at GitHub[65].

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

## Acknowledgements

This work was in part supported by the Air Force Office of Scientific Research under Award Number FA9550-23-1-0090, and the National Science Foundation (Grant DMR–2011924) to Z.J., the Kavli Institute for Brain and Mind award (#4729) to Z.J., B.B., D.A., and K.R and the Jacob's School of Engineering Early Career Faculty Development Award to Z.J. and R.Y. We also acknowledge the support by the National Institutes of Health (1R01HL165491, 1R35GM141598, and R01NS121934) to B.C.

## Author contributions

K.R.: Drafted the manuscript, implemented data analysis and deep learning models, and designed the experiments. Y.Y.: Designed and conducted the experiments and developed the methods section. E.P.F.: Developed deep learning models, conducted the initial study, and assisted in the methods section. C.T.T: Designed and fabricated the devices. D.P.M.: Assisted with running the experiments and contributed to the methods section. D.D.A.: Assisted with running the experiments. A.G.: Contributed to data analysis. B.C., R.Y., F.S., and B.L.B.: provided study oversight and analyzed the data. C.F.: Developed deep learning models, assisted in designing the experiments, provided study oversight, and analyzed the data. Z.J.: Secured funding, provided oversight, designed the experiments, and drafted the manuscript. All authors contributed to manuscript editing for the final version.

## Competing interests

A patent application (application number 63/717,739) has been filed by the University of California San Diego and Stanford University. The named inventors on this patent application are Zeinab Jahed, Keivan Rahmani, Bianxiao Cui, and Csaba Farro. This application covers the method for intracellular recording using an AI model trained on intra- and extracellular paired data, which is a specific aspect of the manuscript. All other authors declare no competing interest.

## Ethics

This study does not involve experiments involving animals, human participants, or clinical samples.

## Additional information

[1]Aiiso Yufeng Li Family Department of Chemical and Nano Engineering, University of California San Diego, La Jolla, CA, USA. [2]Department of Chemistry, Stanford University, Stanford, CA, USA. [3]Wu-Tsai Neuroscience Institute and ChEM-H Institute, Stanford University, Stanford, CA, USA. [4]Department of Computer Science, Stanford University, Stanford, CA, USA. [5]Department of Neurobiology, School of Biological Sciences, University of California San Diego, La Jolla, CA, USA. [6]Department of Computer Science and Engineering, Jacobs School of Engineering, University of California San Diego, La Jolla, CA, USA. [7]Stanford Cardiovascular Institute, Stanford University, Stanford, CA, USA. [8]Center for Advanced Biomaterials for Healthcare, Istituto Italiano di Tecnologia, Naples, Italy. [9]Neuroelectronic Interfaces, Faculty of Electrical Engineering and IT, RWTH Aachen, Aachen, Germany. [10]Institute of Biological Information Processing—Bioelectronics, IBI-3, Forschungszentrum, Juelich, Germany. [11]Chan Zuckerberg Biohub Chicago, Chicago, IL, USA. [12]Shu Chien-Gene Lay Department of Bioengineering, Jacobs School of Engineering, University of California, San Diego, CA, USA. ✉e-mail: cforro@stanford.edu; zjahed@ucsd.edu

