## [Transparent Peer Review file · Nature Communications]

Intelligent In-Cell Electrophysiology: Reconstructing intracellular action potentials using a physics-informed deep learning model trained on nanoelectrode array recordings

Corresponding Author: Dr Zeinab Jahed

Version 0:

Reviewer comments:

Reviewer #1

(Remarks to the Author)

In their manuscript "Intelligent In-Cell Electrophysiology: Reconstructing Action Potentials from Nanoelectrode Data Using Physics-Informed Deep Learning" Dr Jahed and colleagues describe a deep learning-based approach with which they calculate intracellular action potentials from extracellular action potential recordings obtained with nanoelectrode arrays in cardiac monolayer cell cultures. The work is interesting and describes much needed methodology, but the authors need to address concerns regarding some of the core concepts and implementation, and the presentation of the results requires revisions.

My main concern is regarding the generalizability of the technique: deep learning excels on data that is very similar to the training data, but may fail when applied to different data. It is crucial to see how the technique performs with data that comes from outside the lab or was generated with a different nanoelectrode array. In order to demonstrate the broader utility of their technique, the authors should demonstrate that it also works with cell culture data that was not generated in their lab and with other nanoelectrode arrays.

Again, the authors write in the conclusion: "While our study is limited by the size and diversity of our dataset, the potential for future research in intracellular electrophysiology is vast." They should demonstrate that their model can be applied to other data, e.g. cardiac monolayers produced in other laboratories, or cardiac monolayer cell cultures with different protocols and investigate how the iAP reconstruction quality changes when this data is included in the training vs. not included (while separating training and test data). The statement "Expanding the dataset to encompass a wider range of electrogenic cell types and drug interactions will not only refine the accuracy of our findings but also broaden their applicability." should already, at least in parts, be demonstrated in this study already.

The data comes from 'confluent' human stem-cell derived monolayer cardiomyocytes. Can they specify this a bit further? Are the cells ventricular-, atrial- or pacemaker-like?

The action potentials throughout the paper exhibit an undershoot (the baseline is higher before the upstroke and lower after repolarization), which is not what cardiac action potentials nor the Aliev Panfilov model typically exhibit. Please clarify.

What do the authors mean by super-repolarization?

Another concern is regarding the separation of training and test data, which is not clearly stated in the manuscript. Do the authors train on one cell culture and then test on a different cell culture or are they different recordings in the same cell culture? Please clarify and clearly state in the manuscript.

The study lacks references to work where the translation of extracellular electrode recordings into action potentials was learned from optical measurements.

The authors write that "Incorporating physics into the loss function improved the performance of the Attention UNET by approximately 11.7%.". Please provide a plot with a comparison of iAP traces reconstructed with the physics-informed

network and without or a encoder-decoder alone.

The authors should compute activation maps and APD maps from both the eAP and iAP data shown in Fig. 4d,e) and compare.

In the conclusion, the authors claimed that they 'discovered new relationships between features of eAP and iAP'. Could they specify this a bit further what new relationship this is? The deep learning model is a black box and establishes this relationship but it is not accessible to humans.

The authors write that they used recordings from neighboring microelectrodes that are in close physical proximity to obtain precise time-synchronized pairs of eAP and iAP. The authors also demonstrate that two iAP recordings from two neighboring electrodes are very similar. Accordingly, their deep learning approach learns from data that does not 100% correspond to each other. How was the slight spatial mismatch accounted for since the pairs are not exactly from the same physical location?

How large was the distance typically between neighboring electrodes and is it always the same? Where is data regarding inter-electrode distances?

Can't the authors record from the same electrode during repetitive stimulation and record from two subsequent identical AP instead?

What does the ion channel blocker do? Please state its effect onto the shape of the AP wherever appropriate.

Could you please clarify if the iAP used for training are normalized or not? Fig. 2d shows the normalization but Fig. 2g/h show unnormalized ones.

Fig. 3 is the key figure and should appear earlier in the manuscript.

Fig. 4 requires ground-truth comparisons.

Fig. 4f Are the scattered APD measurements in some of the panels reconstruction errors or real APD changes over time? Again, ground-truth data would help to clarify.

Fig. 1a: please also include scale bar for left panel showing 3x3 electrodes.

Fig. 1b: marks extracellular recordings as non-invasive, but the cell is deformed quite substantially. The authors should discuss whether this membrane deformation introduces artifacts or alterations in ion channel behavior (stretch-activated channels) or the electrophysiology.

Fig. 2a: there is an increasing phase shift between the upstroke of the APs and the spikes in the eAP measurements, where eAP spikes precede the upstrokes. Is this a bug or a plotting artefact? Please clarify.

Fig. 2e: What does this panel show? I do not understand.

Fig. 3e requires legends. What is the overlay supposed to show? I do not notice any differences.

Fig. 3e Is S seconds? if so, please write [sec.] or [s] instead.

Fig. 3f The cumulative error is 0.2 seconds? That would be huge and does not match the traces in 3b. Please clarify.

Fig. 4 requires legends, e.g. what are the black and orange traces in panel e) and why is orange noisier than black?

(Remarks on code availability)

The code is not available.

Reviewer #2

(Remarks to the Author)

The authors describe a method for reconstructing intracellular electrophysiological signals (iAPs) using artificial intelligence algorithms. By analyzing thousands of pairs of synchronized extracellular action potentials (eAPs) and iAPs collected from NEA-derived cardiomyocytes, the authors found a strong correlation between specific features of eAPs and iAPs waveforms, such as amplitude and peak velocity. By developing a physics-informed deep learning model trained on these datasets, successful reconstruction of iAPs waveforms through eAPs. This method holds great potential for non-invasive, long-term, and high-throughput assessment of drug-induced cardiac toxicity. However, the article lacks innovation both in sensor and system. In addition, the method of reconstructing iAPs through eAPs still requires further refinement. Therefore, it is not recommended for acceptance in this journal, and a specialized journal will be more suitable.

Additionally, the article may have the following issues:

(1) The text mentions, "We established a signal-to-noise ratio (S/N) cutoff of 90 dB.". How was this threshold selected, and what impact does it have on the results?

(2) The text mentions, "To obtain precise time-synchronized pairs of eAP and iAP, we used recordings from neighboring microelectrodes in a confluent monolayer of human stem-cell-derived cardiomyocytes that are in close physical proximity." It lacks a description of the details of the neighboring microelectrodes, including their distance and spatial layout, so it is recommended to supplement this with a diagram.

(3) Later in the text, it is mentioned that "two neighboring microelectrodes exhibit highly similar iAP waveforms." Are the parameters of these neighboring electrodes the same as those of the aforementioned electrode pairs? Moreover, during electroporation, electrical signals might be conducted through the solution to neighboring electrode locations. Would this affect the eAPs of neighboring electrodes?

(4) How were the starting point bp1 and ending point bp2 of the signals determined? It is recommended to supplement the criteria used for this determination.

(5) Regarding the section on "Application of the system for assessment of drug-induced cardiotoxicity", Figure 4 shows that eAPs can also reflect drug toxicity well. In addition, there may be loss and error in the mapping from eAPs to iAPs signals, thus reducing the ability of the system to assess drug-induced cardiotoxicity. Therefore, further exploration is needed for the application of algorithms reconstructing iAPs from eAPs to demonstrate the necessity of reconstruction.

(6) Some reference formats are incomplete; for example, [9], [10], [11] are missing the page numbers. It is recommended to check the entire document and make corrections.

(7) "Application of our proposed system for high-throughput multi-channel assessment of drug-induced cardiotoxicity." There is an extra full stop here. "Then 32*2, and finally 32 channels" is missing a full stop at the end. It is recommended to check the entire document for formatting issues.

(Remarks on code availability)

Reviewer #3

(Remarks to the Author)

The authors present an original deep learning method to reconstruct intracellular action potentials from extracellular recorded action potentials. They use human iPSC derived cardiomyocytes and plate them on a nanoelectrode array. Using electroporation they gain intracellular access at some electrodes while neighboring electrodes record the corresponding extracellular signals. This unique dataset is used for end-to-end training of a UNET variant. For regularization they introduce a physics-informed loss function based on the well established Aliev-Panfilov model which markedly reduced the mean absolute error of the prediction of the intracellular action potential waveform. In general the paper reads well and results are sound. Their results are of interest to a broader audience because the application of deep learning to reconstruct intracellular action potentials from extracellular signals (shown here for cardiomyocytes) may also be applicable to other excitable cells, particularly neurons.

Major comment

Simplified models such as Aliev-Panfilov model are developed to reduce model complexity compared to a full conductance based model. This is necessary to derive an analytic expression for the physics-informed loss. However, such a simplified model might no longer be applicable if changes to the underlying conductances lead to changes in the action potential waveform. To address this issue, the authors used the drug Dofetilide to purposely alter the shape of action potentials in order to obtain a more diverse dataset for training. This drug works by selectively blocking the IKr current thereby prolonging action potentials. Other drugs can induce more drastic changes in the waveform, e.g. Verapamil (abolishing the plateau phase by blocking calcium channels) and Flecainide (slowing the action potential upstroke by blocking sodium channels). It is unclear (1) whether the Aliev-Panfilov model can represent these waveforms and (2) whether the parameters α and k can be accurately predicted in this case for an unbiased physics-informed loss. Therefore, the network trained on Dofetilide data might not generalize well to new data, i.e. drugs or mutations that affect other cardiac channels.

Minor comments:

"Furthermore, Severe arrhythmias, such as Torsades de Pointes, are detectable only through intracellular electrophysiology techniques" (page 2) TdP is characterized by helical ventricular complexes on the electrocardiogram (ECG), i.e. intracellular recordings are not required for its detection.

"the patch-clamp technique (...) is low in throughput, manual, and remains invasive to the recorded cell." (page 2) High throughput patch clamp recording in cardiomyocytes has been described: Seibertz, F., Rapedius, M., Fakuade, F.E. et al. A modern automated patch-clamp approach for high throughput electrophysiology recordings in native cardiomyocytes. *Commun Biol* 5, 969 (2022). <https://doi.org/10.1038/s42003-022-03871-2>

What is "spiking velocity" (page 5)?

MAE, MSE are given without proper units in table 1 and elsewhere in the text. Please use either seconds or ms.

"repolarization prolongation of stem-cell-derived cardiomyocytes in the presence of a cardiotoxic drug (dofetilide)" (page 6)

Please describe the mechanism of action for this drug.

Please describe the fabrication of the nanoarray (page 7) in sufficient detail or include a reference if it has been described before. Please report the geometry of the chamber.

“This temperature was chosen because it keeps the membrane pore open for longer iAP recordings compared to 37°C.” (page 7) What is meant here by “membrane pore”?

“Furthermore, the drug’s diffusion limits resulted in gradual APD changes until stabilization (Figure-SF-2h), providing a spectrum of iAPs with APD values ranging from normal to fully impacted by the drug.” (page 8) What is meant by “diffusion limits”? What is the evidence that diffusion is the main reason for the gradual drug action? Furthermore, this argument seems to contradict the claim of “homogeneous diffusion” (page 7).

“To characterize eAP waveforms, five critical points where the waveform gradient changes significantly were identified: just before the spike (bp1), maximum point (Ymax), minimum point (Ymin), right after the minimum point (bp2), and where it starts to rise again (bp3).” (page 8) What are these measurements? Voltages?

Please reference previous literature appropriately in the material and methods section, e.g. UNET architecture, attention mechanism, physics informed loss, quantile loss etc.

I think the physics loss in equation (14) is derived from equations (12) and (13).

What are the predicted model parameters α and k for the Aliev-Panfilov model?

For the estimation of the first and second derivative finite difference methods were used, which tend to amplify noise in data and might render the physics informed loss useless. Please show the computed first and second derivative for the reconstructed and the target extracellular trace. Also report the relative contributions of the data and the physics loss.

Why is there a logarithmic transformation of the physics loss in the hybrid loss function?

Please explain the term “quantile crossing” (page 12).

What filter was used to remove noise from intracellular recordings (page 13)?

Please explain “Linear”, “DLinear”, “NLinear”, “MLP” in table legend or material and methods.

Figure 1c: Please explain the time course of drug application. Was there a wash between the drug applications (indicated by dashed boxes)?

Figure 2a: What is the origin of these Wenckebach-like rhythms?

Figure 2a: please explain “%APA”, include a proper unit on the x axis and explain the difference between eAP “voltage” and eAP “spike amplitude”

Figure 2d: Why is the distribution of APD values so different between train and test dataset?

Figure 2b, insert: What is the origin of these distorted spikes? Overlapping signals from adjacent cardiomyocytes that are not fully entrained?

Figure 2e: Dashed lines make it difficult to see the end points of the lines.

Figure 2e: The examples show the largest differences for APD10, 20, 30 whereas the plot in Figure 2f shows the opposite trend.

Figure 2g: The total APD error depends on the somewhat arbitrarily chosen number of levels and is almost as large as the APD itself. An average APD error or maximal APD error might be a better choice.

Figure 4a: Please explain the time course of drug application. Was there a wash between the drug applications (indicated by dashed boxes)?

Supplemental information, Figure S1-3a: The legend mentions 4 different drug concentrations, but only 3 drug applications are shown.

Supplemental information, page 9: Please write out the abbreviation LTFS by the first time you use it.

Although it is not overly complicated, some readers might find it helpful if a step-by-step derivation for formula (14) were included in the supporting information.

(Remarks on code availability)

I could not access the github page, perhaps it is still a "private" repository.
I would like to see the implementation before making a recommendation.

Reviewer #4

(Remarks to the Author)

The authors report a novel methodology for electrophysiological characterization of stem cell derived cardiomyocytes using nanoelectrode arrays, with the support of artificial intelligence tools. The interesting part of methodology is the noninvasive reconstruction of cardiac Intracellular Action Potentials (iAPs) from the external Action Potentials (eAP) recordings. The integration of physics-informed deep learning to reconstruct iAPs from eAPs is novel and addresses the limitations of both invasive patch-clamp techniques and lower-fidelity extracellular recordings. While the results are promising, there are significant weaknesses that need to be addressed through revisions of the manuscript and inclusion of additional data.

MAJOR CONCERNS

The biological component of this research is largely neglected. It is mentioned that studies were done using "stem cell derived cardiomyocytes", without specifying what are the stem cells (human or animal; embryonic or iPS cells), and how many lineages were used. I was unable to find a method section with detailed disclosure of the methodology. Even if broadly studied cell source and a published methodology were used, it is critical to provide extensive characterization of the molecular, ultrastructural and functional (contractility, electrophysiology, calcium handling) of cardiomyocytes and determine the level of their maturity.

Related to the previous comment, it would be also important to document reproducibility of the measurements both within the same cell preparation and batch to batch.

In Figure 2h, through SHAP, ΔT_d , DR, and IR were identified as the most important features to predict APD (especially for > APD50). How do these features correspond to the biological cardiac action potential? Are the minima (labeled as bp3) generally observed in all NEA eAP recordings? If so, is this feature an empirically determined one or one based on known cardiac biology? Was there any feature selection prior to training the XGBoost algorithm?

Are the datasets used for the feature-based model in Figure 2 the same as the ones used in Figure 3? If so, it seems like the feature-based approach is better (APD Error = 0.196s) than the proposed PIA-UNET (APD Error = 0.25s). What is the added benefit of the DL approach? It also seems that the DL approach might have an overfitting issue, with APD error of the test set diverging significantly from the train set.

Dofetilide was used to generate a more diverse set of waveform data to train the model. In validating the model, the authors use the same drug to demonstrate the reconstruction of iAP using eAP recordings. An additional drug other than dofetilide would validate the method in a more robust manner. On a similar note, does the model generalize to other classes of cardiac drugs that affect the AP waveform differently (e.g. in the QRS phases/depolarization rather than QT)? Investigating the effects of another set of drugs (such as a Na class IC blocker that perturbs the QRS phase or a class Ib drug that shortens QT) would strengthen the generalizability of this model to capture perturbations to additional parts of the AP waveform.

MINOR COMMENTS AND SUGGESTIONS

References to Fig 1A and B are missing in the manuscript.

In the model evaluations shown in Fig 2d-g and Fig 3c-g, it would be more compelling to resplit the dataset from the first 2 sets of experiments to create another test set rather than comparing the train set (which I assume the model has already seen before) vs test set as the authors currently show. The current test set is from an independent 3rd experiment set, which is great, but it would be even more compelling to assess performance differences between data from the first 2 sets of experiments that the model has not yet seen compared to the current test set to assess generalizability across experimental setup and hardware.

Please include the number of points in the correlation plot in 3C, as the coefficient of determination is dependent on the number of plotted points.

Please add a color legend for Fig 4F.

(Remarks on code availability)

Reviewer #5

(Remarks to the Author)

I co-reviewed this manuscript with one of the reviewers who provided the listed reports. This is part of the Nature Communications initiative to facilitate training in peer review and to provide appropriate recognition for Early Career Researchers who co-review manuscripts. Here are my additional comments:

The paper presents a promising methodology in electrophysiology by combining high-throughput NEA recordings with a physics-informed loss to train the DL model to enable noninvasive reconstruction of cardiac iAPs using eAP recordings. The integration of physics-informed deep learning to reconstruct iAPs from eAPs is novel and addresses the limitations of both

invasive patch-clamp techniques and lower-fidelity extracellular recordings. Addressing the identified weaknesses and conducting additional experiments will further strengthen the methodology and broaden its applicability.

1. References to Fig 1A and B are missing in the manuscript.
2. In Figure 2h, through SHAP, ΔT_d , DR, and IR were identified as the most important features to predict APD (especially for $> APD_{50}$). How do these features correspond to the biological cardiac action potential? Are the minima (labeled as bp3) generally observed in all NEA eAP recordings? If so, is this feature an empirically determined one or one based on known cardiac biology? Was there any feature selection prior to training the XGBoost algorithm?
3. Are the datasets used for the feature-based model in Figure 2 the same as the ones used in Figure 3? If so, it seems like the feature-based approach is better (APD Error = 0.196s) than the proposed PIA-UNET (APD Error = 0.25s). What is the added benefit of the DL approach? It also seems that the DL approach might have an overfitting issue, with APD error of the test set diverging significantly from the train set.
4. In the model evaluations shown in Fig 2d-g and Fig 3c-g, it would be more compelling to resplit the dataset from the first 2 sets of experiments to create another test set rather than comparing the train set (which I assume the model has already seen before) vs test set as the authors currently show. The current test set is from an independent 3rd experiment set, which is great, but it would be even more compelling to assess performance differences between data from the first 2 sets of experiments that the model has not yet seen compared to the current test set to assess generalizability across experimental setup and hardware.
5. Please include the number of points in the correlation plot in 3C, as the coefficient of determination is dependent on the number of plotted points.
6. Please add a color legend for Fig 4F.
7. Dofetilide was used to generate a more diverse set of waveform data to train the model. In validating the model, the authors use the same drug to demonstrate the reconstruction of iAP using eAP recordings. An additional drug other than dofetilide or a disease model would validate the method in a more robust manner. On a similar note, does the model generalize to other classes of cardiac drugs that affect the AP waveform differently (e.g. in the QRS phases/depolarization rather than QT)? Investigating the effects of another set of drugs (such as a Na class IC blocker that perturbs the QRS phase or a class Ib drug that shortens QT) would strengthen the generalizability of this model to capture perturbations to additional parts of the AP waveform.

(Remarks on code availability)

The GitHub page has not been made publicly available. There is a page not found error.

Version 1:

Reviewer comments:

Reviewer #1

(Remarks to the Author)

I would like to thank the authors for addressing my questions and the other reviewers questions so thoroughly. My concerns were all sufficiently well addressed, and I support the publication of this manuscript if the concerns of the other reviewers were addressed as well. There are only a few remaining formal items that need to be addressed:

- please include the information of your response to question 12 in the supplementary information
- please include the information regarding your normalization and the rationale behind it as explained in your response to question 13 in the supplementary information

(Remarks on code availability)

There is code in the repository, but I did not have the time to verify whether it is usable.

Reviewer #2

(Remarks to the Author)

The authors have made appropriate revisions to address my concerns. In the revised version, the authors supplemented data from different nanoelectrode arrays (NEA) recordings of eAP/iAP to validate the feasibility of the intracellular action potentials reconstruction algorithm. The results showed that the algorithm could be applied to different NEA systems, verifying the algorithm's generality and enhancing the innovation of the paper. Furthermore, the intracellular action potentials algorithm was validated in eAP/iAP obtained from hiPSC-CMs exposed to different drugs, including multiriot, quinine, nifedipine, flumequine, and lidocaine, demonstrating the algorithm's promising application prospects. Finally, the authors refined the details of the algorithm and writing to meet publication requirements. Therefore, I would recommend the acceptance of the work in Nature Communication.

(Remarks on code availability)

Reviewer #3

(Remarks to the Author)

The authors have made significant improvements to the manuscript by addressing most of my suggestions, as well as those of the other reviewers. In particular, testing additional drugs has strengthened the evaluation of PIA-UNET's benefits and predictive power. However, I still have a few minor comments that need to be addressed:

1) Regarding my previous comment 9:

While the explanation provided is now sufficient, I would recommend the following improvements to the notation for clarity and standardization:

1a) It is unconventional to use "x" and "y" for time and voltage, respectively. The more standard symbols would be "t" for time and "V" for voltage. Using the familiar "t" and "V" will make it easier for readers to follow.

1b) The derivatives should be expressed explicitly as "dV/dt" and "d²V/dt²" rather than just "d/dt" or "d²/dt²", which are missing the dependent variable.

1c) There seems to be inconsistency in the use of the variables "u" and "v", where sometimes "1/u" and sometimes "1/v" is written. Please ensure uniformity in the variable naming throughout the text.

2. Regarding my previous comment 15:

The explanation of "quantile crossing" is now clear, but the rationale for how separate output layers for the 0.05, 0.5, and 0.95 quantiles prevent this phenomenon remains unclear. Since these output layers are independent, one might expect them to increase the risk of quantile crossing. Could you provide a more detailed explanation or add some discussion to clarify how the separate layers address this issue?

(Remarks on code availability)

Reviewer #4

(Remarks to the Author)

The authors provided very long and detailed responses to the concerns raised during the review (over 80 pages, not counting references). While some issues have been addressed (for example inclusion of experiments with additional drugs), my main concerns remain. I copy here my major comments followed by the assessment of the authors' responses.

Concern 1: The biological component of this research is largely neglected. It is mentioned that studies were done using "stem cell derived cardiomyocytes", without specifying what are the stem cells (human or animal; embryonic or iPSC cells), and how many lineages were used. I was unable to find a method section with detailed disclosure of the methodology. Even if broadly studied cell source and a published methodology were used, it is critical to provide extensive characterization of the molecular, ultrastructural and functional (contractility, electrophysiology, calcium handling) of cardiomyocytes and determine the level of their maturity.

Characterization of commercially obtained iPSC-CMs is still lacking, and it is really important to document their properties at the time when the APs were measured. Likewise, the "human iPSC-derived cardiomyocytes that were heterogeneously composed, with a predominance of ventricular-like cells (57%), as well as atrial-like and nodal-like cells" are not characterized either. It is also not disclosed how the cells were classified into the ventricular, atrial-like and nodal-like cells in terms of both the methods and the criteria used, and how reproducible is the cell derivation. The authors only state that "the derivation of these cells followed a protocol approved by the Stanford University Human Subjects Research Institutional Review Board, which ensures ethical standards and informed consent from participants (Burrige et al. 2014)". IRB is certainly needed if the cells were derived from the patients' samples, but their detailed characterization is also very important. The authors seem to have limited appreciation for the spectrum of the phenotypes and maturity levels for iPSC-derived cardiomyocytes, both of which can markedly change the APs that are being measured. The proposed methodology cannot be agnostic to the biological properties of the cells.

Concern 2: Related to the previous comment, it would be also important to document reproducibility of the measurements both within the same cell preparation and batch to batch.

The authors acknowledge that "there are several batch-to-batch variabilities, and even heterogeneity within a single culture in the electrophysiological recordings of hiPSC-CMs", but do not report this variability as suggested. Furthermore, they say that this variability is "further highlighting the need for high-throughput approaches like the one proposed in our manuscript to obtain statistically significant results". I do not agree that this is a better approach than optimizing and standardizing the cell derivation to minimize random changes in cell properties. Finally, the authors state: "Our new results included in our

revised manuscript demonstrate that our model can accurately predict iAP across various cell batches under different conditions" without properly documenting and explaining this statement.

Concern 3: In Figure 2h, through SHAP, ΔT_d , DR, and IR were identified as the most important features to predict APD (especially for $> APD_{50}$). How do these features correspond to the biological cardiac action potential? Are the minima (labeled as bp3) generally observed in all NEA eAP recordings? If so, is this feature an empirically determined one or one based on known cardiac biology? Was there any feature selection prior to training the XGBoost algorithm?

This question (also raised by other reviewers) was aimed at clarifying to which extent the culture iPSC-derived cardiomyocytes recapitulate cardiac electrophysiology. Instead of trying to benchmark their recordings against the biological data, they discuss relationships between different parameters extracted from the recordings.

Concern 4: Are the datasets used for the feature-based model in Figure 2 the same as the ones used in Figure 3? If so, it seems like the feature-based approach is better (APD Error = 0.196s) than the proposed PIA-UNET (APD Error = 0.25s). What is the added benefit of the DL approach? It also seems that the DL approach might have an overfitting issue, with APD error of the test set diverging significantly from the train set.

This is another important question for any AI/ML algorithm: clear separation of the training datasets from the experimental datasets. The authors do not clearly respond to this question and instead say: "Yes, the datasets used in Figure 2 and Figure 3 are from the same source. However, for the XGBoost method, we applied screening methods to only retain signals where the critical points were computed accurately by our algorithm and exhibited measurable ΔT_s in their spikes. While XGBoost demonstrated robust performance, it is limited to predicting APD values and cannot reconstruct the entire iAP shape. This makes it suitable for simple studies but insufficient for more comprehensive analysis. Additionally, XGBoost requires intensive data preparation to extract waveform features, whereas the deep learning (DL) method does not. The DL approach can also handle distorted spikes, providing greater versatility".

Concern 5: Dofetilide was used to generate a more diverse set of waveform data to train the model. In validating the model, the authors use the same drug to demonstrate the reconstruction of iAP using eAP recordings. An additional drug other than dofetilide would validate the method in a more robust manner. On a similar note, does the model generalize to other classes of cardiac drugs that affect the AP waveform differently (e.g. in the QRS phases/depolarization rather than QT)? Investigating the effects of another set of drugs (such as a Na class IC blocker that perturbs the QRS phase or a class Ib drug that shortens QT) would strengthen the generalizability of this model to capture perturbations to additional parts of the AP waveform.

In response to this comment, the authors extended their studies to additional drugs, which is an improvement.

(Remarks on code availability)

Reviewer #5

(Remarks to the Author)

(Remarks on code availability)

While I did not run the code myself, there seems to be adequate documentation and code on the repository to run the training myself if desired based on browsing the repository.

Version 2:

Reviewer comments:

Reviewer #4

(Remarks to the Author)

The criticism raised during the reviews has been largely addressed and the manuscript is markedly improved. I would like to suggest to remove any overlap between the training and evaluation datasets, and to include at least some evidence for the biological properties of the cells.

(Remarks on code availability)

Reviewer #1 (Remarks to the Author):

In their manuscript "Intelligent In-Cell Electrophysiology: Reconstructing Action Potentials from Nanoelectrode Data Using Physics-Informed Deep Learning" Dr Jahed and colleagues describe a deep learning-based approach with which they calculate intracellular action potentials from extracellular action potential recordings obtained with nanoelectrode arrays in cardiac monolayer cell cultures. The work is interesting and describes much needed methodology, but the authors need to address concerns regarding some of the core concepts and implementation, and the presentation of the results requires revisions. My main concern is regarding the generalizability of the technique: deep learning excels on data that is very similar to the training data, but may fail when applied to different data. It is crucial to see how the technique performs with data that comes from outside the lab or was generated with a different nanoelectrode array. In order to demonstrate the broader utility of their technique, the authors should demonstrate that it also works with cell culture data that was not generated in their lab and with other nanoelectrode arrays. Again, the authors write in the conclusion: "While our study is limited by the size and diversity of our dataset, the potential for future research in intracellular electrophysiology is vast." They should demonstrate that their model can be applied to other data, e.g. cardiac monolayers produced in other laboratories, or cardiac monolayer cell cultures with different protocols and investigate how the iAP reconstruction quality changes when this data is included in the training vs. not included (while separating training and test data). The statement "Expanding the dataset to encompass a wider range of electrogenic cell types and drug interactions will not only refine the accuracy of our findings but also broaden their applicability." should already, at least in parts, be demonstrated in this study already.

1. generalizability of the technique : In order to demonstrate the broader utility of their technique, the authors should demonstrate that it also works with cell culture data that was not generated in their lab and with other nanoelectrode arrays.

Response:

Thank you for your comment. We have expanded our training and test sets to comprehensively study the generalizability of both our machine learning (XGBoost) and deep learning models. Our updated training dataset includes eAP/iAP recordings from nanoelectrode arrays (NEA) using human iPSC-derived cardiomyocytes (hiPSC-CMs) exposed to dofetilide, quinidine, nifedipine, flecainide, and lidocaine. All these data used for training were collected from hiPSC-CMs, differentiated and recorded in the Cui Lab at Stanford University.

To validate the generalizability of our approach, we employed three different test sets:

1. **Test 1** : hiPSC-CMs exposed to dofetilide, a drug included in the training set, were tested on a different NEA device that was not used for training recordings, to obtain eAP/iAP pairs.
2. **Test 2**: hiPSC-CMs exposed to Propranolol (a drug not included in the training set) was tested on a different NEA from a separate device to obtain eAP/iAP pairs.
3. **Test 3**: eAP/iAP pairs were recorded in a different laboratory using commercial hiPSC-CMs on a commercial microelectrode array (MEA) instead of an NEA. Simultaneous patch clamp recordings from the same cells were used to obtain iAPs. The data were recorded at the Bloodgood Lab at UCSD.

We reported and compared the performance of both the machine learning and deep learning models on these three test sets, achieving high accuracy in both cases.

For the **XGBoost model**, the mean absolute APD error from APD10 to APD100 was:

- Test 1: Mean error of 0.020 ± 0.007 s
- Test 2: Mean error of 0.039 ± 0.006 s
- Test 3: Mean error of 0.047 ± 0.038 s
- The training set had a mean error of 0.0022 ± 0.0015 s with a maximum error of 0.0115 s.

For the deep learning model, the mean absolute APD error from APD10 to APD100 was:

- Test 1: 0.030 ± 0.024 s
- Test 2: 0.027 ± 0.023 s
- Test 3: 0.020 ± 0.019 s

The model exhibited a MAE of:

- Test 1: 0.032 ± 0.041
- Test 2: 0.051 ± 0.050
- Test 3: 0.038 ± 0.045

The training set had a MAE of 0.021 ± 0.031 , indicating lower error during training.

Figures 2 and 3 below showcase the model's performance on the respective test sets.

“Figure 2

Figure 2. Quantitative relationships between eAPs and iAP waveform features. a) A representative recording of

simultaneous eAP and iAP from arrhythmic cells (top), and overlay of eAP amplitude with the iAP spike velocity (represented as iAP spike change percentage over time) showing a strong correlation ($R^2=0.903$). This association is further evident as oscillations in the extracellular recordings appear to reflect the action potential's repolarization phase. **b)** Key points were identified on the eAP (left) and iAP (right) recordings to describe their waveforms. Key points on the eAP include: break point 1 (bp_1) – just before the spike where its derivative notably rises; and break point 2 (bp_2) – immediately post its minimum, as the derivative starts decreasing. Other defining points are: y_{max} (the peak positive value), y_{min} (negative spike minimum following the positive peak), and bp_3 (following the first minimum, marking where the eAP derivative becomes positive). The vertical distance from bp_1 to y_{max} is termed ΔV_1 ; ΔV_2 defines the vertical distance from bp_2 to y_{min} ; ΔT_1 defines the width of the positive spike from bp_1 to its rightmost pre-minima point; ΔT_2 signifies the width of the negative spike from bp_2 to its leftmost pre-minima point; ΔT_d represents the horizontal distance between bp_2 and bp_3 . The decay rate (DR, v/s) is the average slope between bp_2 and bp_3 , reflecting the voltage change over time, while the increase rate defines the slope just after y_{min} . In the context of iAP signals, APD 10 to APD 100 describe the duration it takes for the action potential to return to 10% and 100% of its amplitude, respectively. The increase rate (IR, v/s) is determined by the slope from iAP bp_1 to its peak, while the decay rate gauges the slope from the peak to bp_2 . ΔV_1 is the vertical span between bp_1 and the peak, and ΔT_s charts the vertical distance between bp_1 and bp_2 . Additionally, examples of eAPs with distorted spikes are also presented **c)** Correlation Analysis of eAP Features with iAP APD Values: Showcasing the relationship between various features of eAPs and iAP APD values ranging from APD10 to APD100. This section provides insights into how eAP characteristics correlate with the corresponding iAP APD metrics. **d)** Violin plot distribution of normalized eAP and iAP waveform features: This plot illustrates the distribution of specific eAP features (ΔT_s , ΔV_d , DR, and ΔV_1) and iAP features (APD30, APD50, APD70, and APD90). **e)** Comparison of Predicted and Actual APD: Showcases reconstructed APD lines for the shortest and longest action potential durations from the test set, illustrating model accuracy across varying AP durations. The XGBoost model underwent optimization through hyperparameter tuning on a validation set derived from the training set and was evaluated on the training set and three test sets to assess the model's performance. Test 1 involved hiPSC-CMs exposed to dofetilide, a drug included in the training set, which were tested on a different NEA from a separate device to obtain eAP/iAP pairs. Test 2 involved propranolol, a drug not used in the training set, tested on a different NEA device from a separate batch and wafer to obtain eAP/iAP pairs. Test 3 involved eAP/iAP pairs recorded from a different laboratory using separate hiPSC-CMs (commercial hiPSC-CMs) on a commercial MEA, where simultaneous patch clamp recordings of the same cells were used to obtain iAPs. **f)** Comparative Analysis of Predicted APD Error Values for Test and Training Data: This section evaluates the average prediction errors for APD-30, APD-50, APD-70, and APD-90 across different test sets. For Test Set 1, the errors were 0.010 ± 0.006 s, 0.016 ± 0.012 s, 0.019 ± 0.011 s, and 0.033 ± 0.018 s, respectively. In Test Set 2, the errors were 0.061 ± 0.014 s, 0.054 ± 0.011 s, 0.025 ± 0.007 s, and 0.028 ± 0.011 s. For Test Set 3, the errors were 0.011 ± 0.007 s, 0.055 ± 0.045 s, 0.059 ± 0.070 s, and 0.058 ± 0.063 s. When expressed as percentages, the prediction errors for the APD benchmarks are as follows. For Test Set 1, the errors were $2.390 \pm 1.358\%$ for APD-30, $2.460 \pm 1.706\%$ for APD-50, $2.373 \pm 1.393\%$ for APD-70, and $3.584 \pm 2.059\%$ for APD-90. In Test Set 2, the errors were $13.320 \pm 2.978\%$ for APD-30, $8.154 \pm 1.732\%$ for APD-50, $3.474 \pm 0.908\%$ for APD-70, and $3.655 \pm 1.368\%$ for APD-90. For Test Set 3, the errors were $2.695 \pm 1.671\%$ for APD-30, $11.093 \pm 9.079\%$ for APD-50, $10.598 \pm 12.540\%$ for APD-70, and $9.326 \pm 10.222\%$ for APD-90. These figures demonstrate the robustness of the XGBoost model in accurately predicting APD values ranging from APD 10 to APD 100. **g)** Violin Plot of Average APD Error on Test and Training Sets. The mean absolute APD error from APD10 to APD100 for the test sets were as follows: Test 1 had a mean of 0.020 ± 0.007 s, Test 2 had a mean of 0.040 ± 0.006 s, and Test 3 had a mean of 0.047 ± 0.038 s. For the training set, the mean error was 0.002 ± 0.001 s. **h)** Feature significance based on their SHapley Additive exPlanations (SHAP) values indicates respectively, ΔT_d , $\Delta V_1/\Delta V_2$, ΔT_s , ΔV_1 , and IR₁ as the most important eAP features to predict iAP features. SHAP values illustrate how features alter the predicted value from the model's average output. Feature importance, defined as the average of absolute changes imposed on the predicted values by varying features within their range. **i)** The ranked significance of eAP signal features in predicting ADP 30, APD 50, APD 70, and APD 90 is illustrated, along with their local partial dependency. Each dot signifies a single predicted APD value, its color indicates the feature's value, and its position on the X-axis represents its SHAP value reflects the expected deviation in APD prediction. For example, as the part APD 90 shows, increasing ΔT_s (going from blue to red color), results in increasing the predicted APD 90 value, while ΔT_s and APD 30 follow the opposite direction. The observed SHAP values could be influenced by local minima, potentially limiting their representation of the global relationship between features. This underscores the importance of considering broader contextual factors when interpreting these dependencies.”

“Figure 3

Figure 3. Physics-Informed Attention Unet (PIA-UNET) to reconstruct the entire iAP waveform from the eAP signal. a) PIA-UNET Model Schematic: A visual representation of the PIA-UNET model utilized in the study. The proposed Attention Physics informed UNET (PIA-UNET) model consists of an encoder and decoder, both constructed with Residual Block (ResBlock) functions. The encoder compresses eAP features while the decoder reconstructs the iAP. Each ResBlock contains convolutional, batch normalization, and ReLU (CBR) sequences with squeeze-and-excitation (SE) blocks for enhanced feature representation. The architecture also incorporates skip connections and attention mechanisms, enhancing gradient flow and focus on significant data features for precise iAP reconstruction from eAPs. The Aliev-Panfilov is Incorporated into the PIA-UNET as the physics part of the hybrid loss function. **b)** Comparison between reconstructed iAPs with PIA-UNET (with and without physic incorporation in loss function- orange dashed line for with physics and blue dashed line for without physics in loss) and the actual iAPs derived from corresponding eAPs (black line), on examples of test sets 1, test sets 2 and test sets . Comparison of reconstructed iAPs using PIA-UNET with (orange dashed line) and without (blue dashed line) physics-informed loss functions, alongside the actual iAPs derived from corresponding eAPs (black line), was performed on three test sets. Test 1 involved hiPSC-CMs exposed to dofetilide, a drug included in the training set, tested on a different NEA from a separate device. Test 2 used propranolol, an unseen drug, tested on a different NEA from a separate batch and wafer. Test 3 comprised eAP/iAP pairs recorded from a different laboratory using commercial hiPSC-CMs on a commercial MEA, where simultaneous patch

clamp recordings provided iAPs. Incorporating physics corrected the brief undershoot at the end of repolarization in some NEA iAP recordings (Test 2), and resulted in more accurate predictions on eAPs recorded with the MEA (Test 3). c) iAP Normalized Potential Comparison: Shows the model's predicted potential values closely aligning with the actual values, with a correlation of determination ($r = 0.99$). d) The model exhibits a MAE of 0.032 ± 0.041 on Test 1, 0.051 ± 0.050 on Test 2, and 0.038 ± 0.045 on Test 3. The MAE on the training set is 0.021 ± 0.031 , indicating lower error during training. The higher MAE on Test 2 is attributed to the correction of NEA iAP through the incorporation of physics into the loss function. e) Reconstructed iAPs are compared with actual ones in terms of APD. For **Test 1**, the APD errors are **APD30: 0.029 ± 0.020 s, APD50: 0.028 ± 0.022 s, APD70: 0.017 ± 0.010 s, and APD90: 0.045 ± 0.021 s.** For **Test 2**, the APD errors are **APD30: 0.036 ± 0.026 s, APD50: 0.024 ± 0.028 s, APD70: 0.015 ± 0.025 s, and APD90: 0.027 ± 0.013 s.** For **Test 3**, the APD errors are **APD30: 0.012 ± 0.009 s, APD50: 0.016 ± 0.017 s, APD70: 0.021 ± 0.022 s, and APD90: 0.029 ± 0.018 s.** For the **training set**, the APD errors are **APD30: 0.017 ± 0.015 s, APD50: 0.018 ± 0.018 s, APD70: 0.017 ± 0.021 s, and APD90: 0.017 ± 0.022 s.** In terms of percentage error, for **Test 1**, the model shows errors of $7.33 \pm 5.05\%$ at APD30, $4.27 \pm 3.31\%$ at APD50, $2.15 \pm 1.17\%$ at APD70, and $4.90 \pm 2.00\%$ at APD90. For **Test 2**, the percentage errors are $7.62 \pm 5.47\%$ at APD30, $3.49 \pm 4.02\%$ at APD50, $1.97 \pm 3.21\%$ at APD70, and $3.47 \pm 1.70\%$ at APD90. For **Test 3**, the percentage errors are $3.15 \pm 2.50\%$ at APD30, $3.27 \pm 3.45\%$ at APD50, $3.66 \pm 3.91\%$ at APD70, and $4.66 \pm 2.95\%$ at APD90. On the **training set**, the percentage errors are $4.26 \pm 4.26\%$ at APD30, $3.07 \pm 3.07\%$ at APD50, $2.50 \pm 2.50\%$ at APD70, and $2.41 \pm 2.41\%$ at APD90. f) The mean absolute APD error from APD10 to APD100 was 0.030 ± 0.024 s for Test 1, 0.027 ± 0.023 s for Test 2, and 0.020 ± 0.019 s for Test 3, with the training set showing a lower error of 0.017 ± 0.019 s. In terms of percentage error, these values correspond to $5.63 \pm 4.50\%$ for Test 1, $5.56 \pm 5.91\%$ for Test 2, $4.33 \pm 4.03\%$ for Test 3, and $3.61 \pm 4.26\%$ for the training set. g) The MAE comparison on the test set versus eAP shows that the eAP waveform amplitude/noise ratio is the primary factor influencing prediction error. Signals with lower noise levels are expected to be reconstructed more accurately. This ratio is calculated by dividing the eAP maximum value by the noise level."

2. Q: The data comes from 'confluent' human stem-cell derived monolayer cardiomyocytes. Can they specify this a bit further? Are the cells ventricular-, atrial- or pacemaker-like?

Response:

The data used in this study comes from human induced pluripotent stem cell (iPSC)-derived cardiomyocytes. For the microelectrode array (MEA) recordings, we utilized commercially available Human iPSC-derived ventricular cardiomyocytes (Celo.Cardiomyocytes, Celogics, CAT # C50), which are specifically ventricular-like. For the nanoelectrode array (NEA) recordings, we employed human iPSC-derived cardiomyocytes that were heterogeneously composed, with a predominance of ventricular-like cells (57%), as well as atrial-like and nodal-like cells. These heterogeneous cardiomyocytes were recorded at Stanford University. The derivation of these cells followed a protocol approved by the Stanford University Human Subjects Research Institutional Review Board, which ensures ethical standards and informed consent from participants (Burrige et al. 2014).

3. Q: The action potentials throughout the paper exhibit an undershoot (the baseline is higher before the upstroke and lower after repolarization), which is not what cardiac action potentials nor the Aliev Panfilov model typically exhibit. Please clarify.

Response:

Thank you for your observation regarding the undershoot in the action potentials. Indeed, some NEA intracellular recordings exhibit an undershoot when compared to actual intracellular action potential (iAP) recordings obtained via patch clamp. However, we have compared the NEA and patch clamp iAP recordings in the Supplementary Information (SI) Section 1. The analysis shows that this undershoot does not significantly impact our results. For cells where iAPs were simultaneously recorded using both patch clamp and NEA, the errors for iAP signals with a signal-to-noise ratio (S/N) greater than 90 dB were minimized. Specifically, the Mean Absolute Error (MAE) decreased to 0.024 ± 0.006 , and the average correlation coefficient (r) improved to 0.996 ± 0.003 . The average errors for Action Potential Duration at 50% repolarization (APD50) and 90% repolarization (APD90) between NEA iAPs and patch clamp

recordings were reduced to 0.017 ± 0.020 seconds and 0.004 ± 0.007 seconds, respectively. In percentage terms, these correspond to $4.815 \pm 2.342\%$ for APD50 and $1.343 \pm 0.993\%$ for APD90. Additionally, the mean cycle time difference between two consecutive spikes was recorded at 0.007 ± 0.006 seconds. The following is from SI-Section 1:

“To determine the accuracy of iAP waveforms obtained from NEAs, we performed simultaneous NEA and patch clamp recording from the same cell and compared various features of their waveforms. Nano Crown-shaped NEAs (referred to simply as NEAs herein) were fabricated as previously reported (Jahed et al. 2022). iPSC-CM cells were then seeded onto the NEAs employing a differentiation and culture protocol as outlined in the Methods section.

We performed 15 sets of simultaneous iAP recordings from single cells, utilizing both patch clamp and NEA methods, with a comparative analysis of their waveforms shown in Figure-SI-1a. Each recording set, as depicted in Figure-SI-1b, demonstrated that although NEA iAP traces exhibit a lower amplitude, they closely align with the patch-clamp recordings when normalized. This normalization process allowed for a more accurate comparison between the two techniques. The total duration of these simultaneous recordings was approximately 46 minutes. To quantify the similarity between waveforms, we initially applied high-pass and low-pass filtering, as detailed in the Methods section, to decompose the waveform into isolated noise and refined action potential waveform. To account for various iAP shapes collected during recording, the signals were then segregated using a window length of 8000 timepoints or 1.6 seconds.

This process yielded a total of 3363 pairs of iAP recordings from both NEA and patch clamp methods. The S/N for both NEA and patch recordings was calculated by utilizing the filtered signal and noise vectors, assigning an S/N value to each window of the respective recording methods. Each window, containing pairs of action potential signals, was then normalized to a scale ranging from 0 to 1 to facilitate a more precise comparison. Furthermore, we quantified the APD values and their differences within each window, along with the cycle time for each recording (Figure-SI-1c). We also calculated the discrepancy in cycle times between the NEA and patch clamp methods. Additionally, we determined the Mean Absolute Error (MAE) and analyzed the correlation (r) within each window (Figure-SI-1d). This comprehensive analysis, which included comparisons of NEA and patch clamp recordings as depicted in the Figure-SI-1, provided an in-depth evaluation of the compatibility between these two recording techniques.

The measured low MAE of 0.046 ± 0.028 and a high average correlation (r) of 0.989 ± 0.012 pointed to the near-perfect agreement between the NEA iAP and patch-clamp recordings. Further, key parameters linked to drug-induced heart rhythm abnormalities were investigated by assessing iAP measurements from NEAs against patch clamping, focusing on APD50, and APD90 (cell repolarization markers) and cycle time. The average errors for APD50, and APD90 between NEA iAPs and patch clamps during the experiment were 0.032 ± 0.034 (s), and 0.011 ± 0.016 (s), respectively. Expressed as percentage errors, these values equate to $11.516 \pm 10.010\%$, and $3.875 \pm 3.425\%$, respectively. For cycle time, the mean difference between two consecutive spikes was 0.011 ± 0.024 (s). Figure-SI-1e illustrates how the MAE, APD 50, and APD 90 errors vary across a range of values depending on the S/N in NEA recording traces.

The figure highlights that as the NEA S/N ratio increases, the maximum values of these errors decrease. This trend suggests that low S/N ratios may cause distortions in iAP shape, potentially due to an imperfect cell-to-NEA seal (Jahed et al. 2022). These distortions seem to follow a probability distribution dependent on the S/N value, with deviations diminishing at higher S/N ratios. Based on these findings, to ensure waveform accuracy, we set maximum acceptable thresholds at 0.05 for MAE, and 10% for both APD 50 and APD 90 percentage errors. Our analysis revealed that applying a stringent threshold of 90 dB ($S/N^ = 90$ dB) and filtering signals with S/N ratios above S/N^* ensured that the APD 50 percentage error and MAE remained below 10% and 0.1, respectively, as demonstrated in Figure-SI-1e. The figure also presents examples of eAP and iAP pairs that illustrate the maximum errors observed at or above the S/N^* .*

Upon comparing NEA iAPs that satisfied the S/N^* with corresponding patch clamp iAP recordings, we observed a significant reduction in errors. MAE decreased to 0.024 ± 0.006 , and the average correlation coefficient (r) increased to 0.996 ± 0.003 . The average errors for APD50 and APD90 between NEA iAPs and patch clamps improved to 0.017 ± 0.020 seconds and 0.004 ± 0.007 seconds, respectively. When expressed as percentage errors, these values correspond to $4.815 \pm 2.342\%$ for APD50 and $1.343 \pm 0.993\%$ for APD90. Additionally, the mean cycle time difference between two consecutive spikes was recorded at 0.007 ± 0.006 seconds. The distributions of aforementioned errors are shown in Figure-SI-1f. Furthermore, the experiment level comparison of normalized iAPs from NEA recording and patch clamp is provided in Figure-SI-2. These findings align with a comprehensive comparison of NEA and patch clamp iAPs, particularly when various drugs were introduced during recordings (Jahed et al. 2022). Our findings revealed a near-perfect match between iAPs from the two recording methods when the S/N^* was exceeded. We used this S/N^* threshold for processing of data used to train our model in subsequent steps.

Figure-SI-1. a) Simultaneous iAP recording from iPSC-CMs using patch clamp (PC) and nano electrode array (NEA) via electroporation. **b)** Comparison of iAP recordings from PC and NEA: Scaling between 0 to 1 and segmenting into arrays of length 8000 indices or 1.6s. Includes important features such as cycle time, APD50 (action potential duration at 50% repolarization), and APD90. **c)** Comparison between windows of iAP from PC and NEA by cycle time, APD50, and APD90. **d)** Illustration of mean absolute error (MAE), correlation coefficient (r), and NEA signal to noise ratio (S/N) changes over time during one set of the experiment. **e)** Comparison between MAE, APD50% error, and APD90% error vs NEA S/N . It highlights the changes in three critical errors describing the similarity of NEA-recorded and PC-recorded normalized iAPs (MAE, APD50 percentage error, and APD90 percentage error), with thresholds set at 0.05, 10%, and 10% respectively to ensure reasonable similarity between iAP pairs. Also included is the comparison between iAP pairs with the highest MAE with NEA $S/N > S/N^*$ threshold (S/N^*), shown in blue, as well as examples of iAP pairs with error exceeding the threshold from the region with NEA $S/N < 90$. **f)** Box plot distribution of cycle time difference (dCT(s), r , MAE, APD50%, and APD90% errors between PC and NEA normalized iAP pairs with NEA $S/N > S/N^*$ ($= 90$)).

Regarding the second part of your question, although the undershoot is present, the NEA system enabled high-throughput, simultaneous iAP/eAP recordings, which are essential for our study. These recordings provide valuable insights despite the limitations. Regarding the use of the Aliev-Panfilov model, as you correctly noted, this model does not naturally exhibit an undershoot. Our purpose for incorporating the Aliev-Panfilov model was to explore its ability to help reconstruct the iAP shape more accurately, particularly given that some training data include recordings with minimal undershoot. The results demonstrate that the Aliev-Panfilov model effectively aids in reconstructing iAPs that closely resemble those obtained via patch-clamp recordings, significantly enhancing the model's utility in our analyses. Notably, as shown in **part b of Figure 3**, the samples exhibit an undershoot issue in iAP reconstruction. However, by incorporating the Aliev-Panfilov model, this problem is mitigated, as shown in the figure. The model successfully avoids the undershoots commonly seen in other reconstruction methods, improving the accuracy and reliability of iAP estimation from extracellular action potentials (eAPs).

“**Figure 3b**-Comparison between reconstructed iAPs with PIA-UNET (with and without physic incorporation in loss function- orange dashed line for with physics and blue dashed line for without physics in loss) and the actual iAPs derived from corresponding eAPs (black line), on examples of test sets 1, test sets 2 and test sets 3. Comparison of reconstructed iAPs using PIA-UNET with (orange dashed line) and without (blue dashed line) physics-informed loss functions, alongside the actual iAPs derived from corresponding eAPs (black line), was performed on three test sets. Test 1 involved hiPSC-CMs exposed to dofetilide, a drug included in the training set, tested on a different NEA from a separate device. Test 2 used propranolol, an unseen drug, tested on a different NEA from a separate batch and wafer. Test 3 comprised eAP/iAP pairs recorded from a different laboratory using commercial hiPSC-CMs on a commercial MEA, where simultaneous patch clamp recordings provided iAPs. Incorporating physics corrected the brief undershoot at the end of repolarization in some NEA iAP recordings (Test 2), and resulted in more accurate predictions on eAPs recorded with the MEA (Test 3).”

4. What do the authors mean by super-repolarization?

Response:

Thank you for your comment. By "super-repolarization," we refer to the slight undershoot observed at the end of the repolarization phase in the intracellular action potential (iAP) data recorded using the Nanoelectrode Array (NEA). It is characterized by the membrane potential briefly becoming more negative than the resting potential after the repolarization phase, before returning to its baseline level. We also added that to the manuscript. *“The model captures iAP waveforms and prevents super-repolarization (a brief undershoot at the end of repolarization) in the system.”*

5. Another concern is regarding the separation of training and test data, which is not clearly stated in the manuscript. Do the authors train on one cell culture and then test on a different cell culture or are they different recordings in the same cell culture? Please clarify and clearly state in the manuscript.

Response:

Thank you very much for your comment regarding the separation of training and test data. We have clarified this point in the manuscript : “ *The training dataset included eAP/iAP recordings from NEA using hiPSC-CMs exposed to dofetilide, quinidine, nifedipine, flecainide, and lidocaine. To validate the generalizability of our study, three different test sets were used: Test 1: hiPSC-CMs exposed to dofetilide, a drug included in the training set, were tested on a different NEA from a separate batch and wafer to obtain eAP/iAP pairs. Test 2: Propranolol, not used in the training set, was tested on a different NEA from a separate batch and wafer to obtain eAP/iAP pairs. Test 3: eAP/iAP pairs were recorded from a different laboratory using commercial hiPSC-CMs on a commercial MEA. For this test, simultaneous patch clamp recordings of the same cells were used to obtain iAPs.*” This clarification ensures that the training and test datasets are distinct, with different cell cultures or recordings used for each test set, as outlined above.

6. The study lacks references to work where the translation of extracellular electrode recordings into action potentials was learned from optical measurements.

Response:

Thank you for your comment. We have incorporated the following into the manuscript: “Optical recording of iAPs using voltage-sensitive dyes or proteins is versatile and scalable but limited by low sampling rates, reduced sensitivity, and potential cytotoxicity from photobleaching, which may affect cellular behavior (Hortigon-Vinagre et al. 2016; Liu and Miller 2020).”

7. The authors write that "Incorporating physics into the loss function improved the performance of the Attention UNET by approximately 11.7%.". Please provide a plot with a comparison of iAP traces reconstructed with the physics-informed network and without or a encoder-decoder alone.

Response:

Thank you for your comment. It was a great suggestion. We have incorporated the comparison in Figure 3b.

“**Figure 3b**-Comparison between reconstructed iAPs with PIA-UNET (with and without physic incorporation in loss function- orange dashed line for with physics and blue dashed line for without physics in loss) and the actual iAPs derived from corresponding eAPs (black line), on examples of test sets 1, test sets 2 and test sets 3. Comparison of reconstructed iAPs using PIA-UNET with (orange dashed line) and without (blue dashed line) physics-informed loss functions, alongside the actual iAPs derived from corresponding eAPs (black line), was performed on three test sets. Test 1 involved hiPSC-CMs exposed to

dofetilide, a drug included in the training set, tested on a different NEA from a separate device. Test 2 used propranolol, an unseen drug, tested on a different NEA from a separate batch and wafer. Test 3 comprised eAP/iAP pairs recorded from a different laboratory using commercial hiPSC-CMs on a commercial MEA, where simultaneous patch clamp recordings provided iAPs. Incorporating physics corrected the brief undershoot at the end of repolarization in some NEA iAP recordings (Test 2), and resulted in more accurate predictions on eAPs recorded with the MEA (Test 3).“

8. The authors should compute activation maps and APD maps from both the eAP and iAP data shown in Fig. 4d,e) and compare.

Response:

Thank you for your comment. Both the eAP Activation Map and APD 50 Map, derived from the data shown in Fig. 4d,e, are provided in the Supplementary Information in Figure-SI-8. However, it is important to highlight that the unique star-shaped arrangement of the NEA channels introduces complexities in direct comparison. Unlike the more straightforward applications of Activation Map vs. APD Map comparisons in tissues or uniformly patterned MEAs, the non-uniform layout of the NEA complicates spatial correlation. Despite these challenges, the maps offer valuable insights into the spatiotemporal dynamics of the recorded signals, even with the added complexity of the electrode geometry. We believe this highlights the robustness of our approach in handling more diverse and intricate setups, which could extend the applicability of our method beyond standard configurations.

“Figure-SI-8. a) NEA-recorded eAP Activation Map vs. b) its reconstructed APD 50 Map. Note that the channels are arranged in a star-shaped pattern, as illustrated in part c of the figure.”

9. In the conclusion, the authors claimed that they 'discovered new relationships between features of eAP and iAP'. Could they specify this a bit further what new relationship this is? The deep learning model is a black box and establishes this relationship but it is not accessible to humans

Response:

Thank you for your comment regarding the new relationships discovered between features of eAP and iAP. It is true that deep learning models often function as "black boxes," making it challenging to directly interpret the relationships they establish. To address this, we included an XGBoost model to predict iAP APD values based on eAP features. This approach allows us to perform SHAP (SHapley Additive exPlanations) analysis, which provides insights into which eAP features most significantly impact various APD values. As shown in Figure 2-h and -i, our analysis revealed specific relationships between eAP features and iAP APD values. For example, ΔT_s significantly impacts lower APD values, such as APD30, while ΔT_d is more strongly associated with higher APD values like APD50 to APD90. The plots indicate that higher ΔT_d values (represented by a more reddish color) correlate with an increase in APD50, APD70, and APD90. In contrast, for APD30, ΔT_s is the main contributing factor. Low ΔT_s values do not significantly impact the median predicted APD30, whereas higher ΔT_s values tend to decrease the predicted APD30. By integrating the XGBoost model and SHAP analysis, we were able to gain some valuable insights into the eAP features that most influence different APD values, helping us interpret the relationships between eAPs and iAPs.

c

eAP ΔT_1	-0.50	-0.44	-0.30	-0.08	0.13	0.29	0.40	0.46	0.49	0.51
eAP ΔT_2	-0.65	-0.56	-0.37	-0.10	0.16	0.35	0.48	0.54	0.58	0.60
eAP ΔT_3	-0.61	-0.53	-0.35	-0.10	0.15	0.33	0.45	0.52	0.55	0.58
eAP ΔT_d	-0.03	0.21	0.50	0.76	0.91	0.97	0.97	0.97	0.96	0.94
eAP IR	-0.50	-0.45	-0.37	-0.26	-0.14	-0.07	-0.03	-0.01	-0.01	0.00
eAP DR	-0.19	-0.19	-0.23	-0.27	-0.30	-0.32	-0.34	-0.35	-0.36	-0.38
eAP ΔV_1	-0.34	-0.19	-0.07	0.03	0.08	0.10	0.09	0.07	0.05	0.03
eAP ΔV_2	-0.22	-0.14	-0.06	0.01	0.05	0.05	0.04	0.02	0.00	-0.03
eAP ΔV_d	0.08	0.02	0.09	0.03	-0.04	-0.12	-0.18	-0.22	-0.24	-0.27
eAP $\Delta V_i / \Delta V_2$	-0.16	-0.08	-0.04	-0.02	-0.01	-0.01	0.00	0.00	0.00	0.01
eAP $\Delta V_d / \Delta V_2$	0.09	0.07	0.01	-0.04	-0.09	-0.12	-0.13	-0.14	-0.14	-0.14
eAP $\Delta V_d / \Delta V_1$	0.31	0.21	0.06	-0.09	-0.20	-0.28	-0.31	-0.33	-0.34	-0.34
APD 10										
APD 20										
APD 30										
APD 40										
APD 50										
APD 60										
APD 70										
APD 80										
APD 90										
APD 100										

“Figure 2. Quantitative relationships between eAPs and iAP waveform features. a) A representative recording of simultaneous eAP and iAP from arrhythmic cells (top), and overlay of eAP amplitude with the iAP spike velocity (represented as iAP spike change percentage over time) showing a strong correlation ($R^2=0.903$). This association is further evident as oscillations in the extracellular recordings appear to reflect the action potential's repolarization phase. **b)** Key points were identified on the eAP (left) and iAP (right) recordings to describe their waveforms. Key points on the eAP include: break point 1 (bp_1) – just before the spike where its derivative notably rises; and break point 2 (bp_2) – immediately post its minimum, as the derivative starts decreasing. Other defining points are: y_{max} (the peak positive value), y_{min} (negative spike minimum following the positive peak), and b_{p3} (following the first minimum, marking where the eAP derivative becomes positive). The vertical distance from bp_1 to y_{max} is termed ΔV_1 ; ΔV_2 defines the vertical distance from bp_2 to y_{min} ; ΔT_1 defines the width of the positive spike from bp_1 to its rightmost pre-minima point; ΔT_2 signifies the width of the negative spike from bp_2 to its leftmost pre-minima point; ΔT_d represents the horizontal distance between bp_2 and bp_3 . The decay rate (DR, v/s) is the average slope between bp_2 and bp_3 , reflecting the voltage change over time, while the increase rate defines the slope just after y_{min} . In the context of iAP signals, APD 10 to APD 100 describe the duration it takes for the action potential to return to 10% and 100% of its amplitude, respectively. The increase rate (IR, v/s) is determined by the slope from iAP bp_1 to its peak, while the decay rate gauges the slope from the peak to bp_2 . ΔV_1 is the vertical span between bp_1 and the peak, and ΔT_s charts the vertical distance between bp_1 and bp_2 . Additionally, examples of eAPs with distorted spikes are also presented **c)** Correlation Analysis of eAP Features with iAP APD Values: Showcasing the relationship between various features of eAPs and iAP APD values ranging from APD10 to APD100. This section provides insights into how eAP characteristics correlate with the corresponding iAP APD metrics. **d)** Violin plot distribution of normalized eAP and iAP waveform features: This plot illustrates the distribution of specific eAP features (ΔT_s , ΔV_b , DR, and ΔV_1) and iAP features (APD30, APD50, APD70, and APD90). **e)** Comparison of Predicted and Actual APD: Showcases reconstructed APD lines for the shortest and longest action potential durations from the test set, illustrating model accuracy across varying AP durations. The XGBoost model underwent optimization through hyperparameter tuning on a validation set derived from the training set and was evaluated on the training set and three test sets to assess the model's performance. Test 1 involved hiPSC-CMs exposed to dofetilide, a drug included in the training set, which were tested on a different NEA from a separate device to obtain eAP/iAP pairs. Test 2 involved propranolol, a drug not used in the training set, tested on a different NEA device from a separate batch and wafer to obtain eAP/iAP pairs. Test 3 involved eAP/iAP pairs recorded from a different laboratory using separate hiPSC-CMs (commercial hiPSC-CMs) on a commercial MEA, where simultaneous patch clamp recordings of the same cells were used to obtain iAPs. **f)** Comparative Analysis of Predicted APD Error Values for Test and Training Data: This section evaluates the average prediction errors for APD-30, APD-50, APD-70, and APD-90 across different test sets. For Test Set 1, the errors were 0.010 ± 0.006 s, 0.016 ± 0.012 s, 0.019 ± 0.011 s, and 0.033 ± 0.018 s, respectively. In Test Set 2, the errors were 0.061 ± 0.014 s, 0.054 ± 0.011 s, 0.025 ± 0.007 s, and 0.028 ± 0.011 s. For Test Set 3, the errors were 0.011 ± 0.007 s, 0.055 ± 0.045 s, 0.059 ± 0.070 s, and 0.058 ± 0.063 s. When expressed as percentages, the prediction errors for the APD benchmarks are as follows. For Test Set 1, the errors were $2.390 \pm 1.358\%$ for APD-30, $2.460 \pm 1.706\%$ for APD-50, $2.373 \pm 1.393\%$ for APD-70, and $3.584 \pm 2.059\%$ for APD-90. In Test Set 2, the errors were $13.320 \pm 2.978\%$ for APD-30, $8.154 \pm 1.732\%$ for APD-50, $3.474 \pm 0.908\%$ for APD-70, and $3.655 \pm 1.368\%$ for APD-90. For Test Set 3, the errors were $2.695 \pm 1.671\%$ for APD-30, $11.093 \pm 9.079\%$ for APD-50, $10.598 \pm 12.540\%$ for APD-70, and $9.326 \pm 10.222\%$ for APD-90. These figures demonstrate the robustness of the XGBoost model in accurately predicting APD values ranging from APD 10 to APD 100. **g)** Violin Plot of Average APD Error on Test and Training Sets. The average APD error for the test sets was as follows: Test 1 had a mean of 0.0205 ± 0.0074 s with a maximum error of 0.0435 s, Test 2 had a mean of 0.0395 ± 0.0063 s with a maximum error of 0.0475 s, and Test 3 had a mean of 0.0472 ± 0.0381 s with a maximum error of 0.1721 s. For the training set, the mean error was 0.0022 ± 0.0015 s with a maximum error of 0.0115 s. **h)** Feature significance based on their SHapley Additive exPlanations (SHAP) values indicates respectively, ΔT_s , $\Delta V_1/\Delta V_2$, ΔT_s , ΔV_1 , and IR₁ as the most important eAP features to predict iAP features. SHAP values illustrate how features alter the predicted value from the model's average output. Feature importance, defined as the average of absolute changes imposed on the predicted values by varying features within their range. **i)** The ranked significance of eAP signal features in predicting ADP 30, APD 50, APD 70, and APD 90 is illustrated, along with their local partial dependency. Each dot signifies a single predicted APD value, its color indicates the feature's value, and its position on the X-axis represents its SHAP value reflects the expected deviation in APD prediction. For example, as the part APD 90 shows, increasing ΔT_s (going from blue to red color), results in increasing the predicted APD 90 value, while ΔT_s and APD 30 follow the opposite direction. The observed SHAP values could be influenced by local minima, potentially limiting their representation of the global relationship between features. This underscores the importance of considering broader contextual factors when interpreting these dependencies.”

10. The authors write that they used recordings from neighboring microelectrodes that are in close physical proximity to obtain precise time-synchronized pairs of eAP and iAP. The authors also demonstrate that two iAP recordings from two neighboring electrodes are very similar. Accordingly, their deep learning approach learns from data that does not 100% correspond to each other. How was the slight spatial mismatch accounted for since the pairs are not exactly from the same physical location?

Response:

Thank you for your comment regarding the slight spatial mismatch between eAP and iAP recordings from neighboring microelectrodes. As discussed in the Supplementary Information, under Section II: "Two neighboring channels on NEA show similar iAPs," we demonstrate that while iAP waveforms from two neighboring channels are not exactly identical, they are highly similar. This similarity allows us to reliably collect synchronized pairs of eAP and iAP.

From 22 pairs of simultaneous recordings from neighboring channels, we collected a total of 2661 iAP spikes with a signal-to-noise ratio (S/N) greater than 90 dB. To assess the similarity between iAPs from neighboring channels, we applied the same analysis used for comparing NEA-recorded iAPs to those obtained via patch clamp. Specifically, we scaled the iAPs from neighboring NEA channels to a range of 0 to 1 (Figure SI-3b) and calculated the mean absolute error (MAE) and correlation coefficients between them, as well as compared their action potential durations at 50% (APD50) and 90% (APD90) repolarization, along with the cycle time (Figure SI-3c).

The analysis showed that the average MAE between iAP traces from two neighboring channels was 0.011 ± 0.006 . The correlation coefficients (r) between the iAPs from the two neighboring channels were quantified as 0.999 ± 0.001 . The average errors for APD50 and APD90 between iAPs from two neighboring NEA channels were 0.015 ± 0.009 s and 0.006 ± 0.005 s, respectively. These errors, expressed as percentages, equate to $3.943 \pm 2.233\%$ for both APD50 and APD90. Additionally, the mean difference in cycle time between two consecutive spikes was 0.002 ± 0.002 s.

Figure SI-3d further illustrates pairs of iAPs from neighboring channels, showing the maximum observed MAE, APD50 percentage error, and APD90 percentage error. This evidence demonstrates the consistency of iAP traces from neighboring NEA channels when the S/N ratio is above a certain threshold (S/N*). This consistency supports the approach of using two neighboring channels to record synchronously from two adjacent cells in a confluent monolayer, effectively capturing both eAP from one channel and iAP from the other, as if both were recorded from a single cell. Building on this understanding, we implemented an approach where a cell was electroporated through one channel while recording eAP from an adjacent cell. This method enabled the simultaneous capture of both eAP and iAP data, which was then used as input and corresponding output data for our machine learning and deep learning models.

“Figure-SI-3. **a**) Simultaneous iAP recording from neighboring channels via non-invasive extracellular action potential (NEA) electroporation. This section details the multi-step addition process of Dofetilide, administered in concentrations of 0.3, 1, 3, and finally 10 nM. The process was conducted in four stages at approximately 400, 800, 1200, and 1700 seconds during the recording session. This method was employed to collect a diverse range of iAP shapes, facilitating a more comprehensive analysis. **b**) Comparison process similar to PC vs NEA iAP recording: Scaling between 0 to 1 and segmenting into arrays of length 8000 indices or 1.6s. **c**) Box plot distribution of dCT(s), correlation coefficient (r), MAE, and APD50% and APD90% errors between neighboring and NEA normalized iAP pairs with NEA S/N > S/N* (= 90). **d**) Examples of iAP pairs from neighboring channels with the highest MAE, APD50, and APD90% errors, as indicated in the box plots.”

11. How large was the distance typically between neighboring electrodes and is it always the same? Where is data regarding inter-electrode distances?

Response:

Response: We typically record from neighboring channels, and the distance between two neighboring electrodes in our NEA layout is approximately 120 microns. The following figure illustrates the NEA layout and provides an example of the distance between two neighboring channels. Additionally, we have included the layout details in the supplementary information under Section VI.

“Figure-SI-7. a) NEA layout showing an example of the distance between two neighboring channels.”

12. Can't the authors record from the same electrode during repetitive stimulation and record from two subsequent identical AP instead?

Response:

Thank you for your suggestion regarding recording from the same electrode during repetitive stimulation to obtain two subsequent identical APs. However, there are two main challenges with this approach:

- A. **Gradual Change in iAP Shapes:** When cells are exposed to drugs, the iAP shapes can gradually change over time. As a result, two subsequent iAPs may not necessarily be similar, especially if the drug effects are still evolving. This makes it difficult to ensure that the recorded iAPs are truly identical or representative of a steady-state condition.
- B. **Electroporation Recovery Time:** After a cell is electroporated, it typically takes around 30 to 60 minutes for the membrane to reseal and recover fully. If we were to record an eAP) followed by the next iAP after electroporation, the 30-minute waiting period would significantly reduce the throughput of data recording. This slow process would hinder our ability to collect the large number of synchronized eAP/iAP pairs necessary for training deep learning models, which require substantial amounts of data.

13. What does the ion channel blocker do? Please state its effect onto the shape of the AP wherever appropriate

Response:

Thank you for your comment. We have added a section in the supplementary information discussing the effects of the drugs used in the study on cardiac ion channels and their impact on iAP shape, as outlined below:

“Section III: Drug test experiments.

The following provides a detailed overview of the drugs used in the experiments, their interactions with cardiac ion channels, and their impact on the shape of action potentials.

Dofetilide: *A Class III antiarrhythmic drug, dofetilide primarily blocks the hERG (IKr)(Saliba 2001) potassium channels. By inhibiting this channel, it prolongs the repolarization phase, resulting in an increased action potential duration (APD) and a longer QT interval (Jaiswal and Goldbarg 2014).*

Quinidine: *As a Class IA antiarrhythmic, quinidine blocks sodium (Na⁺) channels (INa), effectively depressing the rapid initial depolarization phase of the action potential. This mechanism reduces the sodium current and affects the overall electrical activity of the heart (Imaizumi and Giles 1987)*

Nifedipine: *A calcium channel blocker in the dihydropyridine class, nifedipine selectively blocks L-type calcium channels (ICa-L)(Hadley and Lederer 1995). This results in a shortened plateau phase of the action potential, leading to a reduced action potential duration (Zemzemi and Rodriguez 2015)*

Flecainide: *A Class IC antiarrhythmic drug, flecainide blocks both Na⁺ channels (INa) and, to a lesser extent, potassium (IKr) channels (O’ Brien et al. 2022; Melgari et al. 2015) . Its overall effect is to prolong the duration of the action potential and the effective refractory period in ventricular fibers, while both are shortened in the Purkinje system, which is likely consistent with its sodium channel blockade (Lavalle et al. 2021)*

Lidocaine: *A Class IB antiarrhythmic, lidocaine primarily blocks Na⁺ channels (INa)(Bean, Cohen, and Tsien 1983) . It may shorten the APD, though the effect observed in the experiments varied based on the sample conditions.*

Propranolol: *Known as a beta-blocker (Wang et al. 2010) and classified as a Class II antiarrhythmic, propranolol also exhibits sodium (INa) channel blocking properties, further influencing cardiac electrical activity (Kirkorian, Touboul, and Atallah 1988). It can result in shortening APD 90, by increasing the repolarization slope(Hirose et al. 2020).*

This comprehensive understanding of the drug-channel interactions and their resultant impact on action potentials aids in evaluating the electrophysiological responses during experiments.

Figure-SI-5. a) The drugs used in the study, their dosages, and their impact on normalized iAP shapes, along with the wafer (W) and device (D) numbers of the unique Nano Electrode Arrays used in the study. The color gradient from red to blue indicates later experiments, with the bluer tones representing data from more recent experiments.”

14. Could you please clarify if the iAP used for training are normalized or not? Fig. 2d shows the normalization but Fig. 2g/h show unnormalized ones.

Response:

Thank you for your comment regarding the normalization of iAP data used for training. We confirm that all iAP data used for training are normalized between 0 and 1, with a starting value at 0.1. This normalization is applied because the features of iAPs are primarily based on their shape rather than their absolute amplitudes. We have updated Figures 2d, 2g, and 2h to consistently use normalized data.

For the eAP data, we also applied a normalization process to ensure consistency. The eAP signals are normalized by dividing the amplitude by the standard deviation of the noise level, and then adjusting the signal to have an initial value of zero. This normalization is crucial to bring all data to a comparable scale and to emphasize relative changes in signal features over absolute values.

c

eAP ΔT_1	-0.50	-0.44	-0.30	-0.08	0.13	0.29	0.40	0.46	0.49	0.51
eAP ΔT_2	-0.65	-0.56	-0.37	-0.10	0.16	0.35	0.48	0.54	0.58	0.60
eAP ΔT_3	-0.61	-0.53	-0.35	-0.10	0.15	0.33	0.45	0.52	0.55	0.58
eAP ΔT_d	-0.03	0.21	0.50	0.76	0.91	0.97	0.97	0.97	0.96	0.94
eAP IR	-0.50	-0.45	-0.37	-0.26	-0.14	-0.07	-0.03	-0.01	-0.01	0.00
eAP DR	-0.19	-0.19	-0.23	-0.27	-0.30	-0.32	-0.34	-0.35	-0.36	-0.38
eAP ΔV_1	-0.34	-0.19	-0.07	0.03	0.08	0.10	0.09	0.07	0.05	0.03
eAP ΔV_2	-0.22	-0.14	-0.06	0.01	0.05	0.05	0.04	0.02	0.00	-0.03
eAP ΔV_d	0.08	0.12	0.09	0.03	-0.04	-0.12	-0.18	-0.22	-0.24	-0.27
eAP $\Delta V_i / \Delta V_2$	-0.16	-0.08	-0.04	-0.02	-0.01	-0.01	0.00	0.00	0.00	0.01
eAP $\Delta V_j / \Delta V_2$	0.09	0.07	0.01	-0.04	-0.09	-0.12	-0.13	-0.14	-0.14	-0.14
eAP $\Delta V_j / \Delta V_1$	0.31	0.21	0.06	-0.09	-0.20	-0.28	-0.31	-0.33	-0.34	-0.34
	APD 10	APD 20	APD 30	APD 40	APD 50	APD 60	APD 70	APD 80	APD 90	APD 100

15. Fig. 3 is the key figure and should appear earlier in the manuscript.

Response:

Thank you for your comment and for highlighting the importance of Figure 3. We agree that it is a key figure in our study. However, we have structured the manuscript to follow a logical narrative that first establishes the validity of our methods before delving into the deep learning results presented in Figure 3. In Figure 1h, we outline our approach to applying deep learning. Later in the study we proceed by first validating the NEA iAP recordings against patch clamp recordings. We then validate the similarity of iAP recordings from neighboring channels, followed by the use of simpler machine learning algorithms to predict APD values from eAP features. By presenting Figure 3 later in the manuscript, we ensure that the reader understands the foundational validation steps that underpin the deep learning analysis. Moving Figure 3 to an earlier position would disrupt this narrative and might make it challenging for readers to follow the progression of our study.

16. Fig. 4 requires ground-truth comparisons.

Response:

Thank you for your comment. We have added a ground-truth iAP channel (*) recorded using electroporation to the predicted values. Although this comparison cannot be applied to all channels, it should demonstrate similar trends.

“Figure 4. Demonstration of high-throughput pharmacology by reconstructing iAPs from multi-channel eAP recordings and predicting the drug dose response of cardiac myocytes. a) An example of a one-channel recording for an extended duration, with 10 nM of Dofetilide added to the dish around $t \approx 300$ s. b) Reconstructed iAPs by PIA-UNET from eAP recording samples at 200, 350, 500, and 850 seconds. c) Changes in APD 50, 70, and 90 values over time, calculated from the reconstructed iAP signals at the corresponding sample times. d) Snapshot of simultaneous eAP recordings from 50 channels on NEA at $t \approx 445$ s (7.42 mins), along with iAP recordings from the neighboring channel next to the second channel on NEA using electroporation (*). e) Reconstructed iAPs from simultaneous eAP recordings across 49 channels on nano-electrode arrays, captured at $t = 445$ s, with actual iAP recording (*) from the neighboring channel next to the second channel. f) Changes in APD values (50, 70, and 90) calculated from reconstructed iAPs, showing variation over the recording period (~19 minutes), with 10 nM of Dofetilide added to the dish around $t \approx 300$ s. g) Box plot distribution showing maximum drug-induced APD changes at 40%, 50%, 70%, and 90% repolarization levels upon exposure to 10 nM Dofetilide, expressed in milliseconds and as a percentage. *** indicates p -value < 0.001 compared to no change.”

17. Fig. 4f Are the scattered APD measurements in some of the panels reconstruction errors or real APD changes over time? Again, ground-truth data would help to clarify.

Response:

Thank you for your comment. We have added a ground-truth iAP channel (*) recorded using electroporation to the predicted values. While this comparison cannot be applied to all channels, it should demonstrate similar trends. The remaining channels and their corresponding iAP and APD values are reconstructed by the deep learning model.

18. Fig. 1a: please also include scale bar for left panel showing 3x3 electrodes.

Response:

Thank you for your comment, we have added that.

19. Fig. 1b: marks extracellular recordings as non-invasive, but the cell is deformed quite substantially. The authors should discuss whether this membrane deformation introduces artifacts or alterations in ion channel behavior (stretch-activated channels) or the electrophysiology.

Response:

Thank you for pointing out the potential concern regarding membrane deformation during recordings on the NEA. While it is true that the cells experience some deformation, which could potentially activate stretch-activated ion channels or alter electrophysiological behavior, we have taken this into account in our study. It's important to note that both intracellular action potentials (iAPs) and extracellular action

potentials (eAPs) are recorded from cells positioned on top of the NEA, experiencing the same deformation conditions. As a result, any deformation-induced effects would be consistent across both types of recordings. This consistency allows us to ensure that the iAP and eAP pairs remain corresponding, which is crucial for training the deep learning model effectively. The primary objective is to provide the deep learning model with a diverse set of corresponding iAP and eAP pairs, covering a wide range of waveform shapes. This helps the model learn the mapping from eAP to iAP and accurately reconstruct iAPs from eAPs, regardless of the deformation condition. Additionally, we have included simultaneous eAP recordings from MEA and iAP recordings from patch clamp techniques on a single cell, which do not undergo the deformation condition. The accurate predictions made in these conditions further confirm that the model has successfully learned the relationship between eAP and iAP, independent of the deformation effects.

20. Fig. 2a: there is an increasing phase shift between the upstroke of the APs and the spikes in the eAP measurements, where eAP spikes precede the upstrokes. Is this a bug or a plotting artefact? Please clarify.

Response:

Thank you for pointing that out. There is indeed a time lag between the spikes, particularly as the distance between channels, and hence the distance between the cells increases. In this specific experiment with the cells exhibiting arrhythmia, successful recordings were only obtained on a few channels, and the arrhythmic eAPs and iAPs were recorded from two cells on non-adjacent channels, which explains the observed phase shift. We studied the spike time differences between neighboring channels (SI-SectionII) and found that it can vary by up to 0.05 seconds, with the disparity increasing for more distant channels. This example was shown as an initial step in our study to demonstrate for the first time some clear correlations between iAP features such as spiking velocity and eAP features such as amplitude, which was only detectable because of the paired intra and extracellular recording and the arrhythmic behaviors. .

21. Fig. 2e: What does this panel show? I do not understand.

Response:

Thank you for your question regarding Figure 2e. This panel illustrates the reconstruction of actual APD (action potential duration) values ranging from 10% to 100% (APD10–APD100) against their predicted values using the XGBoost model. In our study, XGBoost is employed to predict APD values based on the shape of action potentials derived from eAP features. Figure 2e demonstrates how closely the predicted APD values resemble the actual values, providing an example of XGBoost's performance in reconstructing APD values from eAP features. It is important to note that, in our study, XGBoost is used solely to reconstruct APD values, not the entire iAP waveform. This figure highlights how well the actual and reconstructed iAPs approximate each other when the model is given only the APD values.

“Figure 2. Quantitative relationships between eAPs and iAP waveform features.” “e) Comparison of Predicted and Actual APD: Showcases reconstructed APD lines for the shortest and longest action potential durations from the test set, illustrating model accuracy across varying AP durations. The XGBoost model underwent optimization through hyperparameter tuning on a validation set derived from the training set and was evaluated on the training set and three test sets to assess the model's performance. Test 1 involved hiPSC-CMs exposed to dofetilide, a drug included in the training set, which were tested on a different NEA from a separate device to obtain eAP/iAP pairs. Test 2 involved propranolol, a drug not used in the training set, tested on a different NEA device from a separate batch and wafer to obtain eAP/iAP pairs. Test 3 involved eAP/iAP pairs recorded from a different laboratory using separate hiPSC-CMs (commercial hiPSC-CMs) on a commercial MEA, where simultaneous patch clamp recordings of the same cells were used to obtain iAPs.”

22. Fig. 3e requires legends. What is the overlay supposed to show? I do not notice any differences.

Response:

Thank you for your comment regarding Figure 3e. We have added a legend to the figure for clarity. Figure 3e shows the mean APD error for different APD values (from APD10 to APD100) between the reconstructed iAPs using deep learning and the actual iAPs, for both the training and test sets. This panel demonstrates the performance of the deep learning model in accurately reconstructing APDs across the full range of repolarization percentages. If you were referring to Figure 3b, this panel provides examples of actual iAPs recorded using NEA overlaid with the reconstructed iAPs from eAPs using our deep learning model. The overlay is intended to show the accuracy of the model in reconstructing iAPs from eAPs, highlighting how closely the reconstructed iAPs match the actual recorded iAPs.

“Figure 3. Physics-Informed Attention Unet (PIA-UNET) to reconstruct the entire iAP waveform from the eAP signal. a) PIA-UNET Model Schematic: A visual representation of the PIA-UNET model utilized in the study. The proposed Attention Physics informed UNET (PIA-UNET) model consists of an encoder and decoder; both constructed with Residual Block (ResBlock) functions. The encoder compresses eAP features while the decoder reconstructs the iAP. Each ResBlock contains convolutional, batch normalization, and ReLU (CBR) sequences with squeeze-and-excitation (SE) blocks for enhanced feature representation. The architecture also incorporates skip connections and attention mechanisms, enhancing gradient flow and focus on significant data features for precise iAP reconstruction from eAPs. The Aliev-Panfilov is Incorporated into the PIA-UNET as the physics part of the hybrid loss function. **b)** Comparison between reconstructed iAPs with PIA-UNET (with and without physics incorporation in loss function- orange dashed line for with physics and blue dashed line for without physics in loss) and the actual iAPs derived from corresponding eAPs (black line), on examples of test sets 1, test sets 2 and test sets 3. Comparison of reconstructed iAPs using PIA-UNET with (orange dashed line) and without (blue dashed line) physics-informed loss functions, alongside the actual iAPs derived from corresponding eAPs (black line), was performed on three test sets. Test 1 involved hiPSC-CMs exposed to dofetilide, a drug included in the training set, tested on a different NEA from a separate device. Test 2 used propranolol, an unseen drug, tested on a different NEA from a separate batch and wafer. Test 3 comprised eAP/iAP pairs recorded from a different laboratory using commercial hiPSC-CMs on a commercial MEA, where simultaneous patch clamp recordings provided iAPs. Incorporating physics corrected the brief undershoot at the end of repolarization in some NEA iAP recordings (Test 2), and resulted in more accurate predictions on eAPs recorded with the MEA (Test 3). **c)** iAP Normalized Potential Comparison: Shows the model's predicted potential values closely aligning with the actual values, with a correlation of

determination ($r = 0.99$). **d)** The model exhibits a MAE of 0.032 ± 0.041 on Test 1, 0.051 ± 0.050 on Test 2, and 0.038 ± 0.045 on Test 3. The MAE on the training set is 0.021 ± 0.031 , indicating lower error during training. The higher MAE on Test 2 is attributed to the correction of NEA iAP through the incorporation of physics into the loss function. **e)** Reconstructed iAPs are compared with actual ones in terms of APD. For **Test 1**, the APD errors are **APD30: 0.029 ± 0.020 s**, **APD50: 0.028 ± 0.022 s**, **APD70: 0.017 ± 0.010 s**, and **APD90: 0.045 ± 0.021 s**. For **Test 2**, the APD errors are **APD30: 0.036 ± 0.026 s**, **APD50: 0.024 ± 0.028 s**, **APD70: 0.015 ± 0.025 s**, and **APD90: 0.027 ± 0.013 s**. For **Test 3**, the APD errors are **APD30: 0.012 ± 0.009 s**, **APD50: 0.016 ± 0.017 s**, **APD70: 0.021 ± 0.022 s**, and **APD90: 0.029 ± 0.018 s**. For the **training set**, the APD errors are **APD30: 0.017 ± 0.015 s**, **APD50: 0.018 ± 0.018 s**, **APD70: 0.017 ± 0.021 s**, and **APD90: 0.017 ± 0.022 s**. In terms of percentage error, for **Test 1**, the model shows errors of $7.33 \pm 5.05\%$ at APD30, $4.27 \pm 3.31\%$ at APD50, $2.15 \pm 1.17\%$ at APD70, and $4.90 \pm 2.00\%$ at APD90. For **Test 2**, the percentage errors are $7.62 \pm 5.47\%$ at APD30, $3.49 \pm 4.02\%$ at APD50, $1.97 \pm 3.21\%$ at APD70, and $3.47 \pm 1.70\%$ at APD90. For **Test 3**, the percentage errors are $3.15 \pm 2.50\%$ at APD30, $3.27 \pm 3.45\%$ at APD50, $3.66 \pm 3.91\%$ at APD70, and $4.66 \pm 2.95\%$ at APD90. On the **training set**, the percentage errors are $4.26 \pm 4.26\%$ at APD30, $3.07 \pm 3.07\%$ at APD50, $2.50 \pm 2.50\%$ at APD70, and $2.41 \pm 2.41\%$ at APD90. **f)** The mean absolute APD error from **APD10 to APD100** was 0.030 ± 0.024 s for **Test 1**, 0.027 ± 0.023 s for **Test 2**, and 0.020 ± 0.019 s for **Test 3**, with the training set showing a lower error of 0.017 ± 0.019 s. In terms of percentage error, these values correspond to $5.63 \pm 4.50\%$ for **Test 1**, $5.56 \pm 5.91\%$ for **Test 2**, $4.33 \pm 4.03\%$ for **Test 3**, and $3.61 \pm 4.26\%$ for the **training set**. **g)** The MAE comparison on the test set versus eAP shows that the eAP waveform amplitude/noise ratio is the primary factor influencing prediction error. Signals with lower noise levels are expected to be reconstructed more accurately. This ratio is calculated by dividing the eAP maximum value by the noise level.”

23. Fig. 3e Is S seconds? if so, please write [sec.] or [s] instead.

Response:

Thank you for your comment. We have updated Figure 3e to use [s] to indicate seconds for clarity.

24. Fig. 3f The cumulative error is 0.2 seconds? That would be huge and does not match the traces in 3b. Please clarify.

Response:

Thank you for your comment. You are correct in noting that a cumulative error of 0.2 seconds would be quite large and does not match the traces shown in Figure 3b. We apologize for the confusion. Instead of reporting the cumulative APD error, we have updated Figure 3f to display the average APD error. The cumulative error was previously calculated as the total APD error summed from APD10 to APD100. The updated figure now shows the average APD error, which is a more accurate representation of the model's performance. This correction ensures consistency with the data shown in Figure 3b.

25. Fig. 4 requires legends, e.g. what are the black and orange traces in panel e) and why is orange noisier than black?

Response:

Thank you for your comment. We have updated Figure 4 using the deep learning model trained on a new dataset and added the legend. In panel e, the orange color represents the 90% confidence interval for the predictions, while the black line represents the 50th quantile (median) predicted value from the deep learning model based on eAPs.

“Figure 4. Demonstration of high-throughput pharmacology by reconstructing iAPs from multi-channel eAP recordings and predicting the drug dose response of cardiac myocytes. a) An example of a one-channel recording for an extended duration, with 10 nM of Dofetilide added to the dish around $t \approx 300$ s . **b)** Reconstructed iAPs by PIA-UNET from eAP recording samples at 200, 350, 500, and 850 seconds. **c)** Changes in APD 50, 70, and 90 values over time, calculated from the reconstructed iAP signals at the corresponding sample times. **d)** Snapshot of simultaneous eAP recordings from 50 channels on NEA at $t \approx 445$ s (7.42 mins), along with iAP recordings from the neighboring channel next to the second channel on NEA using electroporation (*). **e)** Reconstructed iAPs from simultaneous eAP recordings across 49 channels on nano-electrode arrays, captured at $t = 445$ s, with actual iAP recording (*) from the neighboring channel next to the second channel. **f)** Changes in APD values (50, 70, and 90) calculated from reconstructed iAPs, showing variation over the recording period (~19 minutes) , with 10 nM of Dofetilide added to the dish around $t \approx 300$ s. **g)** Box plot distribution showing maximum drug-induced APD changes at 40%, 50%, 70%, and 90% repolarization levels upon exposure to 10 nM Dofetilide, expressed in milliseconds and as a percentage. *** indicates p -value < 0.001 compared to no change.”

Reviewer #2 (Remarks to the Author):

The authors describe a method for reconstructing intracellular electrophysiological signals (iAPs) using artificial intelligence algorithms. By analyzing thousands of pairs of synchronized extracellular action potentials (eAPs) and iAPs collected from NEA-derived cardiomyocytes, the authors found a strong correlation between specific features of eAPs and iAPs waveforms, such as amplitude and peak velocity. By developing a physics-informed deep learning model trained on these datasets, successful reconstruction of iAPs waveforms through eAPs. This method holds great potential for non-invasive, long-term, and high-throughput assessment of drug-induced cardiac toxicity. However, the article lacks innovation both in sensor and system. In addition, the method of reconstructing iAPs through eAPs still requires further refinement. Therefore, it is not recommended for acceptance in this journal, and a specialized journal will be more suitable.

Additionally, the article may have the following issues:

1. The text mentions, "We established a signal-to-noise ratio (S/N) cutoff of 90 dB." How was this threshold selected, and what impact does it have on the results?

Response:

Thank you for your comment regarding the selection of the signal-to-noise ratio (S/N) cutoff of 90 dB. This S/N threshold was chosen based on a detailed comparison between iAP recordings from NEA and patch clamp from a single cell. As described in Supplementary Information Section 1, we analyzed the correlation between the S/N of NEA-recorded iAPs and the maximum error between normalized pairs of iAPs recorded by patch clamp and NEA. We observed that iAP waveforms recorded using NEA tend to have a lower amplitude compared to those recorded via patch clamp (approximately 2.5 mV vs. 20 mV). This difference in amplitude is logarithmically correlated with the S/N ratio, meaning that a higher amplitude (with the same level of noise) results in a higher S/N ratio. In our analysis of 15 sets of simultaneous iAP recordings from single cells using both patch clamp and NEA methods, we collected a total of 3363 pairs of iAP recordings. For each recording method, the S/N was calculated using filtered signal and noise vectors, and an S/N value was assigned to each recording window containing pairs of action potential signals. These signals were normalized to a scale of 0 to 1 to enable precise comparison. We then quantified the APD values and their differences within each window, along with the cycle time for each recording (Figure SI-1c). Additionally, we calculated the discrepancy in cycle times between the NEA and patch clamp methods, as well as the mean absolute error (MAE) and correlation coefficient (r) within each window (Figure SI-1d). Our analysis showed a low MAE of 0.046 ± 0.028 and a high average correlation (r) of 0.989 ± 0.012 , indicating a strong agreement between the NEA iAP and patch-clamp recordings. Further, the average errors for APD50 and APD90 between NEA iAPs and patch clamps were 0.032 ± 0.034 s and 0.011 ± 0.016 s, respectively. These correspond to percentage errors of $11.516 \pm 10.010\%$ for APD50 and $3.875 \pm 3.425\%$ for APD90. The mean difference in cycle time between two consecutive spikes was 0.011 ± 0.024 s. Figure SI-1e illustrates how the MAE, APD50, and APD90 errors vary with the S/N ratio in NEA recording traces. The figure shows that as the NEA S/N ratio increases, the maximum errors decrease, suggesting that low S/N ratios may cause distortions in iAP shape, possibly due to an imperfect cell-to-NEA seal. These distortions appear to follow a probability distribution dependent on the S/N value, with deviations diminishing at higher S/N ratios. To ensure waveform accuracy, we set maximum acceptable thresholds at 0.05 for MAE and 10% for both APD50 and APD90 percentage errors. We found that applying a stringent threshold of 90 dB ($S/N^* = 90$ dB) and filtering signals with S/N ratios above this threshold ensured that the APD50 percentage error and MAE remained

below 10% and 0.1, respectively, as shown in Figure SI-1e. The figure also presents examples of eAP and iAP pairs illustrating the maximum errors observed at or above the S/N^* . When comparing NEA iAPs that met the S/N^* threshold with corresponding patch clamp iAP recordings, we observed a significant reduction in errors. The MAE decreased to 0.024 ± 0.006 , and the average correlation coefficient (r) increased to 0.996 ± 0.003 . The average errors for APD50 and APD90 between NEA iAPs and patch clamps improved to 0.017 ± 0.020 seconds and 0.004 ± 0.007 seconds, respectively. Expressed as percentage errors, these values were $4.815 \pm 2.342\%$ for APD50 and $1.343 \pm 0.993\%$ for APD90. Additionally, the mean cycle time difference between two consecutive spikes was recorded at 0.007 ± 0.006 seconds. The distributions of these errors are shown in Figure SI-1f.

“Figure-SI-1. a) Simultaneous iAP recording from iPSC-CMs using patch clamp (PC) and nano electrode array (NEA) via electroporation. **b)** Comparison of iAP recordings from PC and NEA: Scaling between 0 to 1 and segmenting into arrays of length 8000 indices or 1.6s. Includes important features such as cycle time, APD50 (action potential duration at 50% repolarization), and APD90. **c)** Comparison between windows of iAP from PC and NEA by cycle time, APD50, and APD90. **d)** Illustration of mean absolute error (MAE), correlation coefficient (r), and NEA signal to noise ratio (S/N) changes over time during one set of the experiment. **e)** Comparison between MAE, APD50% error, and APD90% error vs NEA S/N . It highlights the changes in three critical errors describing the similarity of NEA-recorded and PC-recorded normalized iAPs (MAE, APD50 percentage error, and APD90 percentage error), with thresholds set at 0.05, 10%, and 10% respectively to ensure reasonable similarity between iAP pairs. Also included is the comparison between iAP pairs with the highest MAE with NEA $S/N > S/N^*$, shown in blue, as well as examples of iAP pairs with error exceeding the threshold from the region with NEA $S/N < 90$. **f)** Box plot distribution of cycle time difference (dCT (s), r , MAE, APD50%, and APD90% errors between PC and NEA normalized iAP pairs with NEA $S/N > S/N^*$ ($= 90$). “

2. The text mentions, "To obtain precise time-synchronized pairs of eAP and iAP, we used recordings from neighboring microelectrodes in a confluent monolayer of human stem-cell-derived cardiomyocytes that are in close physical proximity." It lacks a

description of the details of the neighboring microelectrodes, including their distance and spatial layout, so it is recommended to supplement this with a diagram.

Response:

Response: Thank you for your comment. We typically record from neighboring channels, and the distance between two neighboring electrodes in our NEA layout is approximately 120 microns. The following figure illustrates the NEA layout and provides an example of the distance between two neighboring channels. Additionally, we have included the layout details in the supplementary information under Section VI.

"Figure-SI-7. a) NEA layout showing an example of the distance between two neighboring channels."

3. Later in the text, it is mentioned that "two neighboring microelectrodes exhibit highly similar iAP waveforms." Are the parameters of these neighboring electrodes the same as those of the aforementioned electrode pairs? Moreover, during electroporation, electrical signals might be conducted through the solution to neighboring electrode locations. Would this affect the eAPs of neighboring electrodes?

Response:

Thank you for your insightful questions. Regarding the similarity of parameters between neighboring electrodes: The neighboring electrodes mentioned in the manuscript are indeed the same as those discussed earlier in the context of synchronized eAP and iAP recordings. Both nanoelectrode channels used for recording iAP and eAP are in close physical proximity within a confluent monolayer of human stem-cell-derived cardiomyocytes. This proximity ensures that the iAP waveforms recorded from neighboring nanoelectrodes are highly similar, as demonstrated in our analysis using metrics such as

differences in cycle times (dCT), action potential durations (APD), correlation (r), and Mean Absolute Error (MAE). This similarity validates the use of one nanoelectrode's iAP recording as a reliable training target for reconstructing the eAP from its neighboring channel. Regarding the potential conduction of electrical signals through the solution during electroporation: While there is a possibility of electrical signals being conducted through the solution to neighboring electrode locations, several factors mitigate this effect. First, the nanoelectrodes (nano crowns) are closely surrounded by cardiac cells, which minimizes the spread of electrical signals through the solution. Additionally, the electroporation process is carefully controlled, with the applied voltage and duration optimized to achieve electroporation only at the target electrode without significantly affecting adjacent ones. The voltage used is relatively low ($\pm 4V$ with a $200 \mu s$ duration per phase), which limits the spread of the electrical field. Furthermore, when collecting pairs of eAPs and iAPs, we incorporate a 20-second buffer period after electroporation before starting the actual recordings. This buffer time allows any temporary, short-term effects of electroporation to dissipate, ensuring that the eAPs and iAPs used for training the deep learning model are not affected by transient electrical conduction to neighboring electrodes.

4. How were the starting point bp1 and ending point bp2 of the signals determined? It is recommended to supplement the criteria used for this determination.

Response:

Thank you for your comment. We have added the following explanation to Section VIII: eAP Features Determination Methods in Supplementary Information.

*“**Determination of Starting Point bp₁:** bp₁ is identified through a series of signal processing steps aimed at detecting significant changes in the signal's behavior:*

1. **Signal Smoothing:** *The initial portion of the extracellular action potential (eAP) signal, from the beginning up to the point of maximum voltage (x1), is extracted for analysis. To reduce noise and highlight the essential features of the signal, a Savitzky-Golay filter is applied. This filtering technique smooths the signal while preserving its overall shape, which is critical for accurately detecting transitions in the signal.*
2. **Gradient Calculation:** *After smoothing, the gradient of the signal is computed to accentuate the areas where rapid changes occur, which are often indicative of important transitions such as the start of the signal. This gradient is further smoothed to reduce noise, which ensures that only the most significant changes are considered in the next step.*
3. **Change Point Detection:** *The processed gradient signal is analyzed using a method based on Jenks natural breaks optimization, which is designed to find natural divisions in data by minimizing the variance within groups and maximizing the variance between groups. In this context, the method is used to identify distinct changes in the signal that likely correspond to the initiation of the eAP. Specifically, the algorithm segments the gradient into three distinct classes, and the most relevant change point for bp₁ is selected as the last point where the gradient signal shifts from one class to another.*

***Determination of bp₂:** bp₂ is determined using a similar methodology, tailored to identify the conclusion of the signal.*

1. **Signal Segmentation:** *A specific segment of the eAP signal, starting just beyond the point of*

maximum voltage and extending a short distance thereafter, is isolated for this analysis. This region is chosen because it is expected to contain the signal's descent back to the baseline, marking the end of the action potential.

2. **Signal Smoothing and Gradient Calculation:** As with the determination of *bp1*, the signal segment is smoothed using the Savitzky-Golay filter, and its gradient is calculated and smoothed. This step is crucial for emphasizing the downward trend as the signal returns to baseline levels.
3. **Change Point Detection:** The processed gradient of this segment is then analyzed using a slightly modified version of the Jenks natural breaks optimization, this time dividing the data into two classes. This adjustment reflects the simpler nature of the signal at this stage, which primarily involves a transition back to the baseline. The algorithm identifies the most significant change point as the first point where the gradient signal transitions between classes, corresponding to the point where the signal's descent begins to level off, marking the end of the eAP.””

5. Regarding the section on "Application of the system for assessment of drug-induced cardiotoxicity", Figure 4 shows that eAPs can also reflect drug toxicity well. In addition, there may be loss and error in the mapping from eAPs to iAPs signals, thus reducing the ability of the system to assess drug-induced cardiotoxicity. Therefore, further exploration is needed for the application of algorithms reconstructing iAPs from eAPs to demonstrate the necessity of reconstruction.

Response:

Thank you for your comment. We have expanded our training and test sets to comprehensively study the generalizability of both our machine learning (XGBoost) and deep learning models. Our updated training dataset includes eAP/iAP recordings from nanoelectrode arrays (NEA) using human iPSC-derived cardiomyocytes (hiPSC-CMs) exposed to dofetilide, quinidine, nifedipine, flecainide, and lidocaine. All these data used for training were collected from hiPSC-CMs, differentiated and recorded in the Cui Lab at Stanford University.

To validate the generalizability of our approach, we employed three different test sets:

4. **Test 1 :** hiPSC-CMs exposed to dofetilide, a drug included in the training set, were tested on a different NEA device that was not used for training recordings, to obtain eAP/iAP pairs.
5. **Test 2:** hiPSC-CMs exposed to Propranolol (a drug not included in the training set) was tested on a different NEA from a separate device to obtain eAP/iAP pairs.
6. **Test 3:** eAP/iAP pairs were recorded in a different laboratory using commercial hiPSC-CMs on a commercial microelectrode array (MEA) instead of an NEA. Simultaneous patch clamp recordings from the same cells were used to obtain iAPs. The data were recorded at the Bloodgood Lab at UCSD.

We reported and compared the performance of both the machine learning and deep learning models on these three test sets, achieving high accuracy in both cases.

For the **XGBoost model**, , the mean absolute APD error from APD10 to APD100 was:

- Test 1: Mean error of 0.020 ± 0.007 s
- Test 2: Mean error of 0.039 ± 0.006 s
- Test 3: Mean error of 0.047 ± 0.038 s
- The training set had a mean error of 0.0022 ± 0.0015 s with a maximum error of 0.0115 s.

For the deep learning model, the mean absolute APD error from APD10 to APD100 was:

- Test 1: 0.030 ± 0.024 s
- Test 2: 0.027 ± 0.023 s
- Test 3: 0.020 ± 0.019 s

The model exhibited a MAE of:

- Test 1: 0.032 ± 0.041
- Test 2: 0.051 ± 0.050
- Test 3: 0.038 ± 0.045

The training set had a MAE of 0.021 ± 0.031 , indicating lower error during training.

Figures 2 and 3 below showcase the model's performance on the respective test sets.

“Figure 2

Figure 2. Quantitative relationships between eAPs and iAP waveform features. a) A representative recording of

simultaneous eAP and iAP from arrhythmic cells (top), and overlay of eAP amplitude with the iAP spike velocity (represented as iAP spike change percentage over time) showing a strong correlation ($R^2=0.903$). This association is further evident as oscillations in the extracellular recordings appear to reflect the action potential's repolarization phase. **b)** Key points were identified on the eAP (left) and iAP (right) recordings to describe their waveforms. Key points on the eAP include: break point 1 (bp_1) – just before the spike where its derivative notably rises; and break point 2 (bp_2) – immediately post its minimum, as the derivative starts decreasing. Other defining points are: y_{max} (the peak positive value), y_{min} (negative spike minimum following the positive peak), and bp_3 (following the first minimum, marking where the eAP derivative becomes positive). The vertical distance from bp_1 to y_{max} is termed ΔV_1 ; ΔV_2 defines the vertical distance from bp_2 to y_{min} ; ΔT_1 defines the width of the positive spike from bp_1 to its rightmost pre-minima point; ΔT_2 signifies the width of the negative spike from bp_2 to its leftmost pre-minima point; ΔT_d represents the horizontal distance between bp_2 and bp_3 . The decay rate (DR, v/s) is the average slope between bp_2 and bp_3 , reflecting the voltage change over time, while the increase rate defines the slope just after y_{min} . In the context of iAP signals, APD 10 to APD 100 describe the duration it takes for the action potential to return to 10% and 100% of its amplitude, respectively. The increase rate (IR, v/s) is determined by the slope from iAP bp_1 to its peak, while the decay rate gauges the slope from the peak to bp_2 . ΔV_1 is the vertical span between bp_1 and the peak, and ΔT_s charts the vertical distance between bp_1 and bp_2 . Additionally, examples of eAPs with distorted spikes are also presented **c)** Correlation Analysis of eAP Features with iAP APD Values: Showcasing the relationship between various features of eAPs and iAP APD values ranging from APD10 to APD100. This section provides insights into how eAP characteristics correlate with the corresponding iAP APD metrics. **d)** Violin plot distribution of normalized eAP and iAP waveform features: This plot illustrates the distribution of specific eAP features (ΔT_s , ΔV_d , DR, and ΔV_1) and iAP features (APD30, APD50, APD70, and APD90). **e)** Comparison of Predicted and Actual APD: Showcases reconstructed APD lines for the shortest and longest action potential durations from the test set, illustrating model accuracy across varying AP durations. The XGBoost model underwent optimization through hyperparameter tuning on a validation set derived from the training set and was evaluated on the training set and three test sets to assess the model's performance. Test 1 involved hiPSC-CMs exposed to dofetilide, a drug included in the training set, which were tested on a different NEA from a separate device to obtain eAP/iAP pairs. Test 2 involved propranolol, a drug not used in the training set, tested on a different NEA device from a separate batch and wafer to obtain eAP/iAP pairs. Test 3 involved eAP/iAP pairs recorded from a different laboratory using separate hiPSC-CMs (commercial hiPSC-CMs) on a commercial MEA, where simultaneous patch clamp recordings of the same cells were used to obtain iAPs. **f)** Comparative Analysis of Predicted APD Error Values for Test and Training Data: This section evaluates the average prediction errors for APD-30, APD-50, APD-70, and APD-90 across different test sets. For Test Set 1, the errors were 0.010 ± 0.006 s, 0.016 ± 0.012 s, 0.019 ± 0.011 s, and 0.033 ± 0.018 s, respectively. In Test Set 2, the errors were 0.061 ± 0.014 s, 0.054 ± 0.011 s, 0.025 ± 0.007 s, and 0.028 ± 0.011 s. For Test Set 3, the errors were 0.011 ± 0.007 s, 0.055 ± 0.045 s, 0.059 ± 0.070 s, and 0.058 ± 0.063 s. When expressed as percentages, the prediction errors for the APD benchmarks are as follows. For Test Set 1, the errors were $2.390 \pm 1.358\%$ for APD-30, $2.460 \pm 1.706\%$ for APD-50, $2.373 \pm 1.393\%$ for APD-70, and $3.584 \pm 2.059\%$ for APD-90. In Test Set 2, the errors were $13.320 \pm 2.978\%$ for APD-30, $8.154 \pm 1.732\%$ for APD-50, $3.474 \pm 0.908\%$ for APD-70, and $3.655 \pm 1.368\%$ for APD-90. For Test Set 3, the errors were $2.695 \pm 1.671\%$ for APD-30, $11.093 \pm 9.079\%$ for APD-50, $10.598 \pm 12.540\%$ for APD-70, and $9.326 \pm 10.222\%$ for APD-90. These figures demonstrate the robustness of the XGBoost model in accurately predicting APD values ranging from APD 10 to APD 100. **g)** Violin Plot of Average APD Error on Test and Training Sets. The mean absolute APD error from APD10 to APD100 for the test sets were as follows: Test 1 had a mean of 0.020 ± 0.007 s, Test 2 had a mean of 0.040 ± 0.006 s, and Test 3 had a mean of 0.047 ± 0.038 s. For the training set, the mean error was 0.002 ± 0.001 s. **h)** Feature significance based on their SHapley Additive exPlanations (SHAP) values indicates respectively, ΔT_d , $\Delta V_1/\Delta V_2$, ΔT_s , ΔV_1 , and IR₁ as the most important eAP features to predict iAP features. SHAP values illustrate how features alter the predicted value from the model's average output. Feature importance, defined as the average of absolute changes imposed on the predicted values by varying features within their range. **i)** The ranked significance of eAP signal features in predicting ADP 30, APD 50, APD 70, and APD 90 is illustrated, along with their local partial dependency. Each dot signifies a single predicted APD value, its color indicates the feature's value, and its position on the X-axis represents its SHAP value reflects the expected deviation in APD prediction. For example, as the part APD 90 shows, increasing ΔT_s (going from blue to red color), results in increasing the predicted APD 90 value, while ΔT_s and APD 30 follow the opposite direction. The observed SHAP values could be influenced by local minima, potentially limiting their representation of the global relationship between features. This underscores the importance of considering broader contextual factors when interpreting these dependencies.”

“Figure 3

Figure 3. Physics-Informed Attention Unet (PIA-UNET) to reconstruct the entire iAP waveform from the eAP signal. a) PIA-UNET Model Schematic: A visual representation of the PIA-UNET model utilized in the study. The proposed Attention Physics informed UNET (PIA-UNET) model consists of an encoder and decoder, both constructed with Residual Block (ResBlock) functions. The encoder compresses eAP features while the decoder reconstructs the iAP. Each ResBlock contains convolutional, batch normalization, and ReLU (CBR) sequences with squeeze-and-excitation (SE) blocks for enhanced feature representation. The architecture also incorporates skip connections and attention mechanisms, enhancing gradient flow and focus on significant data features for precise iAP reconstruction from eAPs. The Aliev-Panfilov is Incorporated into the PIA-UNET as the physics part of the hybrid loss function. **b)** Comparison between reconstructed iAPs with PIA-UNET (with and without physic incorporation in loss function- orange dashed line for with physics and blue dashed line for without physics in loss) and the actual iAPs derived from corresponding eAPs (black line), on examples of test sets 1, test sets 2 and test sets . Comparison of reconstructed iAPs using PIA-UNET with (orange dashed line) and without (blue dashed line) physics-informed loss functions, alongside the actual iAPs derived from corresponding eAPs (black line), was performed on three test sets. Test 1 involved hiPSC-CMs exposed to dofetilide, a drug included in the training set, tested on a different NEA from a separate device. Test 2 used propranolol, an unseen drug, tested on a different NEA from a separate batch and wafer. Test 3 comprised eAP/iAP pairs recorded from a different laboratory using commercial hiPSC-CMs on a commercial MEA, where simultaneous patch

clamp recordings provided iAPs. Incorporating physics corrected the brief undershoot at the end of repolarization in some NEA iAP recordings (Test 2), and resulted in more accurate predictions on eAPs recorded with the MEA (Test 3). c) iAP Normalized Potential Comparison: Shows the model's predicted potential values closely aligning with the actual values, with a correlation of determination ($r = 0.99$). d) The model exhibits a MAE of 0.032 ± 0.041 on Test 1, 0.051 ± 0.050 on Test 2, and 0.038 ± 0.045 on Test 3. The MAE on the training set is 0.021 ± 0.031 , indicating lower error during training. The higher MAE on Test 2 is attributed to the correction of NEA iAP through the incorporation of physics into the loss function. e) Reconstructed iAPs are compared with actual ones in terms of APD. For **Test 1**, the APD errors are **APD30: 0.029 ± 0.020 s**, **APD50: 0.028 ± 0.022 s**, **APD70: 0.017 ± 0.010 s**, and **APD90: 0.045 ± 0.021 s**. For **Test 2**, the APD errors are **APD30: 0.036 ± 0.026 s**, **APD50: 0.024 ± 0.028 s**, **APD70: 0.015 ± 0.025 s**, and **APD90: 0.027 ± 0.013 s**. For **Test 3**, the APD errors are **APD30: 0.012 ± 0.009 s**, **APD50: 0.016 ± 0.017 s**, **APD70: 0.021 ± 0.022 s**, and **APD90: 0.029 ± 0.018 s**. For the **training set**, the APD errors are **APD30: 0.017 ± 0.015 s**, **APD50: 0.018 ± 0.018 s**, **APD70: 0.017 ± 0.021 s**, and **APD90: 0.017 ± 0.022 s**. In terms of percentage error, for **Test 1**, the model shows errors of $7.33 \pm 5.05\%$ at APD30, $4.27 \pm 3.31\%$ at APD50, $2.15 \pm 1.17\%$ at APD70, and $4.90 \pm 2.00\%$ at APD90. For **Test 2**, the percentage errors are $7.62 \pm 5.47\%$ at APD30, $3.49 \pm 4.02\%$ at APD50, $1.97 \pm 3.21\%$ at APD70, and $3.47 \pm 1.70\%$ at APD90. For **Test 3**, the percentage errors are $3.15 \pm 2.50\%$ at APD30, $3.27 \pm 3.45\%$ at APD50, $3.66 \pm 3.91\%$ at APD70, and $4.66 \pm 2.95\%$ at APD90. On the **training set**, the percentage errors are $4.26 \pm 4.26\%$ at APD30, $3.07 \pm 3.07\%$ at APD50, $2.50 \pm 2.50\%$ at APD70, and $2.41 \pm 2.41\%$ at APD90. f) The mean absolute APD error from APD10 to APD100 was 0.030 ± 0.024 s for Test 1, 0.027 ± 0.023 s for Test 2, and 0.020 ± 0.019 s for Test 3, with the training set showing a lower error of 0.017 ± 0.019 s. In terms of percentage error, these values correspond to $5.63 \pm 4.50\%$ for Test 1, $5.56 \pm 5.91\%$ for Test 2, $4.33 \pm 4.03\%$ for Test 3, and $3.61 \pm 4.26\%$ for the training set. g) The MAE comparison on the test set versus eAP shows that the eAP waveform amplitude/noise ratio is the primary factor influencing prediction error. Signals with lower noise levels are expected to be reconstructed more accurately. This ratio is calculated by dividing the eAP maximum value by the noise level."

6. Some reference formats are incomplete; for example, [9], [10], [11] are missing the page numbers. It is recommended to check the entire document and make corrections.

Response:

Thank you for identifying these formatting issues. We thoroughly checked all the references and corrected the missing page numbers for references [9], [10], [11], as well as any other formatting inconsistencies throughout the document. We appreciate your attention to detail and have ensured that the reference section is now complete and properly formatted.

7. "Application of our proposed system for high-throughput multi-channel assessment of drug-induced cardiotoxicity." There is an extra full stop here. "Then 32*2, and finally 32 channels" is missing a full stop at the end. It is recommended to check the entire document for formatting issues.

Response:

Thank you for identifying these formatting issues. We have corrected the extra full stop and added the missing full stop. We have also thoroughly reviewed the entire document for any additional formatting inconsistencies and have made the necessary corrections.

Reviewer #3 (Remarks to the Author):

The authors present an original deep learning method to reconstruct intracellular action potentials from extracellular recorded action potentials. They use human iPSC derived cardiomyocytes and plate them on a nanoelectrode array. Using electroporation they gain intracellular access at some electrodes while neighboring electrodes record the corresponding extracellular signals. This unique dataset is used for end-to-end training of a UNET variant. For regularization they introduce a physics-informed loss function based on the well established Aliev-Panfilov model which markedly reduced the mean absolute error of the prediction of the intracellular action potential waveform. In general the paper reads well and results are sound. Their results are of interest to a broader audience because the application of deep learning to reconstruct intracellular action potentials from extracellular signals (shown here for cardiomyocytes) may also be applicable to other excitable cells, particularly neurons.

Major comment

Simplified models such as Aliev-Panfilov model are developed to reduce model complexity compared to a full conductance based model. This is necessary to derive an analytic expression for the physics-informed loss. However, such a simplified model might no longer be applicable if changes to the underlying conductances lead to changes in the action potential waveform. To address this issue, the authors used the drug Dofetilide to purposely alter the shape of action potentials in order to obtain a more diverse dataset for training. This drug works by selectively blocking the IKr current thereby prolonging action potentials. Other drugs can induce more drastic changes in the waveform, e.g. Verapamil (abolishing the plateau phase by blocking calcium channels) and Flecainide (slowing the action potential upstroke by blocking sodium channels). It is unclear (1) whether the Aliev-Panfilov model can represent these waveforms and (2) whether the parameters α and k can be accurately predicted in this case for an unbiased physics-informed loss.

1. Therefore, the network trained on Dofetilide data might not generalize well to new data, i.e. drugs or mutations that affect other cardiac channels.

Response:

Thank you for your insightful comment. We appreciate the suggestion to include additional drugs and drug classes to better validate the generalizability of our model. Based on your feedback, we have expanded our study to incorporate more diverse drugs in both the training and test sets, including drugs that affect different phases of the action potential (AP).

To ensure a broad spectrum of synchronized eAP and iAP pairs for our training dataset, we introduced various ion-channel blockers in a dose-dependent manner to hiPSC-derived cardiomyocytes (hiPSC-CMs), which allowed us to achieve a wide range of action potential shapes. The drugs used in the training dataset included:

- **Dofetilide:** A Class III antiarrhythmic drug, dofetilide primarily blocks the hERG (IKr)³⁷ potassium channels. By inhibiting this channel, it prolongs the repolarization phase, resulting in an increased action potential duration (APD) and a longer QT interval (Jaiswal and Goldbarg

2014).

- **Quinidine:** As a Class IA antiarrhythmic, quinidine blocks sodium (Na^+) channels (INa), effectively depressing the rapid initial depolarization phase of the action potential. This mechanism reduces the sodium current and affects the overall electrical activity of the heart (Imaizumi and Giles 1987)
- **Nifedipine:** A calcium channel blocker in the dihydropyridine class, nifedipine selectively blocks L-type calcium channels (ICa-L)⁴¹. This results in a shortened plateau phase of the action potential, leading to a reduced action potential duration (Zemzemi and Rodriguez 2015)
- **Flecainide:** A Class IC antiarrhythmic drug, flecainide blocks both Na^+ channels (INa) and, to a lesser extent, potassium (IKr) channels^{42,43}. Its overall effect is to prolong the duration of the action potential and the effective refractory period in ventricular fibers, while both are shortened in the Purkinje system, which is likely consistent with its sodium channel blockade (Lavalle et al. 2021)
- **Lidocaine:** A Class IB antiarrhythmic, lidocaine primarily blocks Na^+ channels (INa)⁴⁴. It may shorten the APD, though the effect observed in the experiments varied based on the sample conditions.
- **Propranolol:** Known as a beta-blocker⁴⁵ and classified as a Class II antiarrhythmic, propranolol also exhibits sodium (INa) channel blocking properties, further influencing cardiac electrical activity (Kirkorian et al. 1988). It can result in shortening APD 90, by increasing the repolarization slope (Hirose et al. 2020).

In addition to these, we incorporated recordings from an unseen device, the commercial Microelectrode Array (MEA), instead of the NEA, using different commercially available hiPSC-derived cardiomyocytes. This was done to further test the model's generalizability. We collected a total of 2364 synchronized eAP/iAP pairs from ten independent experiments using different cell cultures on NEA, sourced from different wafers and batches. For the validation of our study, we used three distinct test sets:

- **Test 1:** hiPSC-CMs exposed to dofetilide (a drug included in the training set) were tested on a different NEA from a separate batch and wafer to obtain eAP/iAP pairs.
- **Test 2:** Propranolol (not used in the training set) was tested on a different NEA from a separate batch and wafer to obtain eAP/iAP pairs.
- **Test 3:** eAP/iAP pairs were recorded from a different laboratory using commercial hiPSC-CMs on a commercial MEA ("Pharmacology (cardio)," n.d.). For this test, simultaneous patch clamp recordings of the same cells were used to obtain iAPs.

By including a broader range of drugs and an unseen device in our testing, we have aimed to enhance the robustness and generalizability of our model, ensuring it can capture perturbations to various parts of the AP waveform, not just the QT interval.

2. whether the Aliev-Panfilov model can represent these waveforms including other drugs?

Response:

Thank you for your comment. As you correctly mentioned, the Aliev-Panfilov model has limitations in the range of action potential shapes it can generate. However, we can address this issue through the following three points:

- A. **We have modified Aliev-Panfilov by replacing the step function in Equation (1) with a sigmoid function that includes a variable x .** This allows the model to capture a wider variety of action potential shapes. Here is the detailed Aliev-Panfilov model and our modification:

$$\frac{dv}{dt} = kv(1 - v)(v - a) - vw \quad (1)$$

$$\frac{dw}{dt} = \varepsilon(v, a, x)(kv - w) \quad (2)$$

In these equations, v and w represent the normalized iAP and recovery variables, respectively. To ensure the differentiability is loss function, the step function $\varepsilon(v, a, x)$ is approximated using a sigmoid activation function, which smoothly transitions between values and preserves the model's ability to backpropagate gradients effectively.

$$\varepsilon(v, a, x) \approx x \cdot \sigma(n(a - v)) + (1 - x) \cdot \sigma(n(v - a)) \quad (3)$$

in which n is a constant determining the sharpness of the stem function. In our approach, we have chosen to set n to be 1000. The parameter a is the excitation threshold, while k controls the magnitude of the transmembrane current. The parameter x controls the balance between excitation and recovery dynamics by modulating the smoothness of transitions in the recovery mechanism based on the membrane potential v and the threshold a . This flexibility allows the action potential model to adapt to a wider range of iAP shapes. While we acknowledge that these shapes may still not fully match all potential action potential patterns and remain somewhat limited, it's important to note that the contribution of this modified pseudo-physics function in our study is accompanied by a low weighting factor, and the primary factor remains the data loss function. The physics-informed component, however, serves to ensure that the deep learning model does not predict unrealistic iAPs when exposed to new eAP types. Additionally, we provide examples of iAPs that can be reconstructed purely by solving the modified Aliev-Panfilov model using the corresponding parameters in the figure below (Figure 2).

Aliev Panfilov
Result

Actual
iAPS

Figure 2: Examples of iAPs and their corresponding Aliev-Panfilov model outputs when hiPSC-CMs are exposed to different drugs, along with their predicted parameters (a , k , and x).

B. The results show that incorporating physics into the loss function led to a notable improvement in APD error, enhancing the waveform similarity. However, this adjustment caused a higher mean absolute error due to the correction of super-repolarization—a brief undershoot at the end of repolarization observed in some NEA recordings. In terms of generalizability, the physics-based approach performed significantly better in Test 3, which involved eAPS on MEA, when compared to patch clamp iAPS recordings. This suggests improved robustness across different test conditions despite the increase in absolute error.

Table S1- Comparison of Model Performance With and Without Physics-based Loss Incorporation

PHYSICS	MEA Test1	MEA Test2	MEA Test3	M-APD Test1 [ms]	M-APD Test2 [ms]	M-APD Test3 [ms]	M-APD Test1 [%]	M-APD Test2 [%]	M-APD Test3 [%]	Runs
ON	0.036 ± 0.004	0.051 ± 0.004	0.040 ± 0.002	0.030 ± 0.001	0.022 ± 0.002	0.023 ± 0.006	5.782 ± 0.582	4.216 ± 0.315	4.915 ± 1.199	3
OFF	0.035 ± 0.001	0.029 ± 0.003	0.048 ± 0.002	0.033 ± 0.002	0.023 ± 0.002	0.033 ± 0.006	7.902 ± 0.394	4.772 ± 0.360	6.579 ± 1.090	3

C. Finally, while some NEA recordings might display a slight undershoot, incorporating the Aliev-Panfilov model, which does not naturally exhibit this behavior, helps refine the iAP reconstruction process. The model effectively generates iAPs that closely resemble those obtained from patch-clamp recordings, enhancing its utility in our analysis. For example, in part b of Figure 3, we observe a minor undershoot in some samples during iAP reconstruction. By leveraging the Aliev-Panfilov model, this issue is minimized, resulting in smoother reconstructions without the undershoot that may appear in other methods, improving the overall accuracy of iAP estimations from eAP data.

“Figure 3b-Comparison between reconstructed iAPs with PIA-UNET (with and without physic incorporation in loss function- orange dashed line for with physics and blue dashed line for without physics in loss) and the actual iAPs derived from corresponding eAPs (black line), on examples of test sets 1, test sets 2 and test sets 3. Comparison of reconstructed iAPs using PIA-UNET with (orange dashed line) and without (blue dashed line) physics-informed loss functions, alongside the actual iAPs

derived from corresponding eAPs (black line), was performed on three test sets. Test 1 involved hiPSC-CMs exposed to dofetilide, a drug included in the training set, tested on a different NEA from a separate device. Test 2 used propranolol, an unseen drug, tested on a different NEA from a separate batch and wafer. Test 3 comprised eAP/iAP pairs recorded from a different laboratory using commercial hiPSC-CMs on a commercial MEA, where simultaneous patch clamp recordings provided iAPs. Incorporating physics corrected the brief undershoot at the end of repolarization in some NEA iAP recordings (Test 2), and resulted in more accurate predictions on eAPs recorded with the MEA (Test 3).“

3. whether the parameters α and k can be accurately predicted in this case for an unbiased physics-informed loss.

Response: Thank you for your comment. We have added a section in the Supplementary Information (SI) that includes the physics parameters (a , k , and x) predicted by the model, as well as the reconstruction of iAPs using those parameters in the Aliev-Panfilov model. While the reconstructed iAPs may not be perfectly accurate, the figure demonstrates that the iAP values reconstructed using the predicted a , k , and x parameters resemble the iAPs predicted by the deep learning model.

“Section V: Physics loss in PIA-UNET.

Figure-SI-7. A) Overlay of predicted iAPs using PIA-UNET, actual iAPs, and iAPs generated by the Aliev-Panfilov model based on PIA-UNET physics parameter output values, for all three test sets. b) Overlay of the physics-informed loss values from the PIA-UNET for all three test set predictions. The parameter “a” was predicted to be 0.057, “k” was fixed at 100.00 for all samples, and the distribution of “x” is shown in part c. The initial membrane potential u_0 was set to 0.1, with du/dt at t_0 initialized to 0.4 and the equation was solved for $t = 1.6$ sec.

Minor comments:

1. “Furthermore, Severe arrhythmias, such as Torsades de Pointes, are detectable only through intracellular electrophysiology techniques” (page 2) TdP is characterized by helical ventricular complexes on the electrocardiogram (ECG), i.e. intracellular recordings are not required for its detection.

Response:

Thank you for your valuable comment. You are correct that Torsades de Pointes (TdP) is characterized by helical ventricular complexes on the electrocardiogram (ECG), and intracellular recordings are not required for its detection. We have revised the introduction accordingly.

“Electrophysiology, which investigates the electrical characteristics of biological cells and tissues, is crucial for understanding drug mechanisms, developing cardiac and neurological therapies, and evaluating cardiotoxicity across various drugs⁸. Preclinical proarrhythmic assessments currently rely on in vitro hERG potassium channel assays or in vivo electrocardiogram measurements in large animals⁹. However, these methods are costly and labor-intensive, so they are usually employed late in development, limiting the feasibility of making chemical modifications. Furthermore, evaluating a drug's effect solely on the hERG channel can be insufficient, as drugs can affect multiple ion channels. For example, Verapamil, a strong hERG blocker, is clinically safe due to its calcium channel blocking effect, which offsets its impact on hERG channels¹⁰. Intracellular action potentials (iAPs) reflect the activity of multiple cardiac ion channels, and subtle changes in iAPs can indicate cardiotoxicity or serve as biomarkers⁸. Advances in cardiomyocyte culture and the use of human induced pluripotent stem cell-derived cardiomyocytes (hiPSC-CMs) have greatly enhanced the availability of human cardiomyocytes for in vitro studies. Consequently, the CiPA initiative, launched by the FDA and other agencies, proposes in vitro screening of cardiac iAPs^{11,12}. This approach may offer a more accurate assessment of cardiac toxicity than the hERG assay and facilitate toxicity evaluation of early drug candidates.”

2. “the patch-clamp technique (...) is low in throughput, manual, and remains invasive to the recorded cell.” (page 2) High throughput patch clamp recording in cardiomyocytes has been described: Seibertz, F., Rapedius, M., Fakuade, F.E. et al. A modern automated patch-clamp approach for high throughput electrophysiology recordings in native cardiomyocytes. *Commun Biol* 5, 969 (2022). <https://doi.org/10.1038/s42003-022-03871-2>

Response:

Thank you for your valuable comment. You are correct. We have revised the introduction accordingly *“The gold standard for intracellular electrophysiology technique currently used in all areas of medical sciences is the patch-clamp technique. This method is very powerful and can measure intracellular potentials with very high precision. However, this technique is low in throughput, manual, and remains invasive to the recorded cell (Figure 1b). Automated patch-clamp systems, while improving throughput, require enzymatic dissociation of cells, which can alter cardiomyocyte electrophysiological properties^{13,14}. Optical recording of iAPs using voltage-sensitive dyes or proteins is versatile and scalable but limited by low sampling rates, reduced sensitivity, and potential cytotoxicity from photobleaching, which may affect cellular behavior^{15,16}.”*

3. What is “spiking velocity” (page 5)?

Response:

Thank you for your comment. We apologize for the oversight. The term "iAP spike velocity" refers to the rate of change in intracellular action potential (iAP) during the spiking phase, expressed as the percentage change in iAP potential over time (% change in iAP per second). We have clarified this in the manuscript and added the explanation to the Figure 2 caption for better understanding.

“**Figure 2. Quantitative relationships between eAPs and iAP waveform features.** a) A representative recording of simultaneous eAP and iAP from arrhythmic cells (top), and an overlay of eAP amplitude (representing its maximum voltage during the spiking phase [mv]) with iAP spike velocity (represented as the percentage change in iAP voltage during its spiking phase over time [% change in iAP/s]), showing a strong correlation ($r = 0.903$). This association is further evident as oscillations in the extracellular recordings appear to reflect the action potential's repolarization phase.”

4. MAE, MSE are given without proper units in table 1 and elsewhere in the text. Please use either seconds or ms.

Response:

Thank you for your comment. The MAE and MSE values in Table S-1 and elsewhere in the text are reported to compare the predicted iAPs against the normalized actual iAPs. Since the iAPs are normalized with a maximum value of 1 and a starting point of 0.1, it would not be appropriate to use units such as seconds or millivolts (mV). The normalization process allows us to compare the shapes of the action potentials rather than their absolute values, and therefore these metrics are presented as dimensionless values reflecting this normalized scale.

5. “repolarization prolongation of stem-cell-derived cardiomyocytes in the presence of a cardiotoxic drug (dofetilide)” (page 6) Please describe the mechanism of action for this drug.

Response:

Thank you for your comment. We have added a section in the supplementary information discussing the effects of the drugs used in the study on cardiac ion channels and their impact on iAP shape, as outlined below:

“Section III: Drug test experiments.

The following provides a detailed overview of the drugs used in the experiments, their interactions with cardiac ion channels, and their impact on the shape of action potentials.

Dofetilide: A Class III antiarrhythmic drug, dofetilide primarily blocks the hERG (IKr)(Saliba 2001) potassium channels. By inhibiting this channel, it prolongs the repolarization phase, resulting in an increased action potential duration (APD) and a longer QT interval (Jaiswal and Goldberg 2014).

Quinidine: As a Class IA antiarrhythmic, quinidine blocks sodium (Na^+) channels (INa), effectively depressing the rapid initial depolarization phase of the action potential. This mechanism reduces the sodium current and affects the overall electrical activity of the heart (Imaizumi and Giles 1987)

Nifedipine: A calcium channel blocker in the dihydropyridine class, nifedipine selectively blocks L-type

calcium channels ($ICa-L$)(Hadley and Lederer 1995). This results in a shortened plateau phase of the action potential, leading to a reduced action potential duration (Zemzemi and Rodriguez 2015)

Flecainide: A Class IC antiarrhythmic drug, flecainide blocks both Na^+ channels (INa) and, to a lesser extent, potassium (IKr) channels (O' Brien et al. 2022; Melgari et al. 2015) . Its overall effect is to prolong the duration of the action potential and the effective refractory period in ventricular fibers, while both are shortened in the Purkinje system, which is likely consistent with its sodium channel blockade (Lavalley et al. 2021)

Lidocaine: A Class IB antiarrhythmic, lidocaine primarily blocks Na^+ channels (INa)(Bean, Cohen, and Tsien 1983) . It may shorten the APD, though the effect observed in the experiments varied based on the sample conditions.

Propranolol: Known as a beta-blocker (Wang et al. 2010) and classified as a Class II antiarrhythmic, propranolol also exhibits sodium (INa) channel blocking properties, further influencing cardiac electrical activity (Kirkorian, Touboul, and Atallah 1988). It can result in shortening APD 90, by increasing the repolarization slope(Hirose et al. 2020).

This comprehensive understanding of the drug-channel interactions and their resultant impact on action potentials aids in evaluating the electrophysiological responses during experiments.

Figure-SI-5. a) The drugs used in the study, their dosages, and their impact on normalized iAP shapes, along with the wafer (W) and device (D) numbers of the unique Nano Electrode Arrays used in the study. The color gradient from red to blue indicates later experiments, with the bluer tones representing data from more recent experiments.”

- Please describe the fabrication of the nanoarray (page 7) in sufficient detail or include a reference if it has been described before. Please report the geometry of the chamber.

Response:

Thank you for your comment. We have briefly described the fabrication process in the Methods section and cited the main paper for the Nano Electrode Array fabrication details. The following is what we added:

“Fabrication of Nano Electrode Arrays

The fabrication process is fully explained in the paper by Jahed et al(Jahed et al. 2022). Briefly, we used maskless photolithography followed by deep reactive ion etching to develop vertical SiO₂ nanopillars, which were then coated with Pt metal to achieve conductivity. The metal was etched from the tip of the pillars using a directional dry etch to achieve the nano crown shape.”

7. “This temperature was chosen because it keeps the membrane pore open for longer iAP recordings compared to 37°C.” (page 7) What is meant here by “membrane pore”?

Response:

Thank you for your comment. By "membrane pore," we are referring to the pores created by electroporation. The temperature and electroporation conditions were optimized in a previous study (Jahed et al. 2022), and the chosen temperature allows these electroporated pores to remain open longer, facilitating extended intracellular action potential (iAP) recordings. We changed the manuscript to make it more clear:

“This method, though more complex, uses a short electric pulse to temporarily create localized pores in the cell membrane, allowing the electrode to gain intracellular access and record iAPs at specific intervals. While these pores are highly localized, they can still disrupt cellular physiology at the nanoscale, and they eventually reseal, limiting the duration of intracellular recordings.”

8. “Furthermore, the drug’s diffusion limits resulted in gradual APD changes until stabilization (Figure-SF-2h), providing a spectrum of iAPs with APD values ranging from normal to fully impacted by the drug.” (page 8) What is meant by “diffusion limits”? What is the evidence that diffusion is the main reason for the gradual drug action? Furthermore, this argument seems to contradict the claim of “homogeneous diffusion” (page 7).

Response:

You're absolutely right. The mistake in writing caused the misunderstanding. What we meant to convey is that by adding the drug in a high dose, rather than incrementally with lower doses, we are still able to capture the full range of iAP shapes, from no drug to high-dose drug impact, while maintaining strong signal-to-noise (S/N) ratios. This allows us to observe gradual changes in iAPs despite the single-step addition of the drug. We have updated the paragraph accordingly: *“Obtaining a diverse dataset of eAP and iAP pairs. We tried two approaches for data collection: incremental drug addition and single high-dose drug application. In the first approach, we added the drugs in multiple steps to capture a spectrum of iAP shapes and their corresponding eAPs. However, we also applied a relatively high concentration of the drugs in a single step during each recording session (Figure 1g), which enabled us to achieve a wider range of iAP durations (Figure-SF-2a). The single high-dose approach minimized the noise typically associated with incremental drug addition.*

Given that both amplitude and S/N ratio tend to decrease over time—resulting in significantly lower S/N ratios and amplitudes in the final traces compared to the initial ones (Figures SF-2b to SF-2e, with paired t-test p-values of 8.6×10^{-15} and 1.1×10^{-8} , respectively)—this strategy allowed us to collect high S/N ratio data at higher drug dosages. Moreover, the APD values changed gradually until stabilization (Figure-SF-2h), providing a spectrum of iAPs with APD values ranging from normal to fully impacted by the drug. Consequently, this approach, similar to multi-step drug addition, resulted in a diverse range of eAP and iAP pairs, and as shown in Figures SF-2h to SF-2i, it facilitated the acquisition of longer APDs in a shorter time with improved S/N ratios while allowing the APD values to change incrementally.”

9. “To characterize eAP waveforms, five critical points where the waveform gradient changes significantly were identified: just before the spike (bp1), maximum point (Ymax),

minimum point (Ymin), right after the minimum point (bp2), and where it starts to rise again (bp3).” (page 8) What are these measurements? Voltages?

Response:

Thank you for your comment. The measurements referred to are not voltages per se but are rather their corresponding x and y coordinates, representing time points (x) and voltages (y), respectively. These critical points are used to characterize the eAP waveforms based on their locations on the time-voltage graph. Specifically, these points help in calculating the eAP characteristics by measuring either the voltage difference or the time difference between them. The parameters with ΔV represent voltage differences between the critical points, while those with Δt correspond to the time differences between these points. We also added a section in the Supplementary Information (SI) discussing the methods used to detect the main points bp1 and bp2 as follows:

“ *Section IX: eAP Features Determination Methods.*

Determination of Starting Point bp₁: *bp₁ is identified through a series of signal processing steps aimed at detecting significant changes in the signal's behavior.*

1. **Signal Smoothing:** *The initial portion of the extracellular action potential (eAP) signal, from the beginning up to the point of maximum voltage (x1), is extracted for analysis. To reduce noise and highlight the essential features of the signal, a Savitzky-Golay filter is applied. This filtering technique smooths the signal while preserving its overall shape, which is critical for accurately detecting transitions in the signal.*
2. **Gradient Calculation:** *After smoothing, the gradient of the signal is computed to accentuate the areas where rapid changes occur, which are often indicative of important transitions such as the start of the signal. This gradient is further smoothed to reduce noise, which ensures that only the most significant changes are considered in the next step.*
3. **Change Point Detection:** *The processed gradient signal is analyzed using a method based on Jenks natural breaks optimization, which is designed to find natural divisions in data by minimizing the variance within groups and maximizing the variance between groups. In this context, the method is used to identify distinct changes in the signal that likely correspond to the initiation of the eAP. Specifically, the algorithm segments the gradient into three distinct classes, and the most relevant change point for bp1 is selected as the last point where the gradient signal shifts from one class to another.*

Determination of bp₂: *bp₂ is determined using a similar methodology, tailored to identify the conclusion of the signal.*

1. **Signal Segmentation:** *A specific segment of the eAP signal, starting just beyond the point of maximum voltage and extending a short distance thereafter, is isolated for this analysis. This region is chosen because it is expected to contain the signal's descent back to the baseline, marking the end of the action potential.*
2. **Signal Smoothing and Gradient Calculation:** *As with the determination of bp1, the signal segment is smoothed using the Savitzky-Golay filter, and its gradient is calculated and smoothed. This step is crucial for emphasizing the downward trend as the signal returns to baseline levels.*
3. **Change Point Detection:** *The processed gradient of this segment is then analyzed using a slightly modified version of the Jenks natural breaks optimization, this time dividing the data into two classes. This adjustment reflects the simpler nature of the signal at this stage, which primarily*

involves a transition back to the baseline. The algorithm identifies the most significant change point as the first point where the gradient signal transitions between classes, corresponding to the point where the signal's descent begins to level off, marking the end of the eAP.

“

10. Please reference previous literature appropriately in the material and methods section, e.g. UNET architecture, attention mechanism, physics informed loss, quantile loss etc.

Response:

Thank you for your comment. We have now appropriately referenced the relevant literature in the Materials and Methods section, including references for the UNET architecture, attention mechanism, physics-informed loss, quantile loss, and other methodologies.

11. I think the physics loss in equation (14) is derived from equations (12) and (13).

Response:

You are absolutely right! Thank you for your comment. We have corrected the mistake and updated the manuscript accordingly.

“Incorporating the value of w from equation (12) into equation (13) and deriving it with respect to time (t), we obtain a single equation”

12. What are the predicted model parameters α and k for the Aliev-Panfilov model?

Response:

Thank you for your comment. We have added a section in the Supplementary Information (SI) showing the physics parameters (a , k , and x) predicted by the model and the reconstruction of iAPs using those parameters when plugged into the Aliev-Panfilov model.

“Section V: Physics loss in PIA-UNET.

Figure-SI-7. A) Overlay of predicted iAPs using PIA-UNET, actual iAPs, and iAPs generated by the Aliev-Panfilov model based on PIA-UNET physics parameter output values, for all three test sets. b) Overlay of the physics-informed loss values from the PIA-UNET for all three test set predictions. The parameter “a” was predicted to be 0.057, “k” was fixed at 100.00 for all samples, and the distribution of “x” is shown in part c. The initial membrane potential u_0 was set to 0.1, with du/dt at t_0 initialized to 0.4 and the equation was solved for $t = 1.6$ sec.

You are absolutely right! Thank you for your comment. We have corrected the mistake and updated the manuscript accordingly.

“Incorporating the value of w from equation (12) into equation (13) and deriving it with respect to time (t), we obtain a single equation”

13. For the estimation of the first and second derivative finite difference methods were used, which tend to amplify noise in data and might render the physics informed loss useless. Please show the computed first and second derivative for the reconstructed and the target extracellular trace. Also report the relative contributions of the data and the physics loss.

Response:

Thank you for your insightful comment. During our training process, we implemented Gaussian smoothing to mitigate the impact of random noise on the calculations of the first and second derivatives. Finite difference methods, as you mentioned, can amplify noise, especially when used for estimating derivatives from noisy data. By smoothing the predictions, we ensured that the noise was minimized, which allowed the physics-informed loss function to effectively optimize the model.

Below, we showcase the comparison of actual and predicted intracellular action potentials (iAPs), along with their first and second derivatives, using samples corresponding to the 30th, 60th, and 90th quantiles of recording time. These figures illustrate the differences between the reconstructed and target extracellular traces for the iAP, its first derivative (d/dt), and its second derivative (d^2/dt^2), demonstrating the performance of our model across various stages of the recordings.

Figure 1: Comparison of Actual and Predicted iAPs, First Derivatives, and Second Derivatives Row 1: Actual vs. Predicted iAPs for 30th, 60th, and 90th quantiles. Row 2: Actual vs. Predicted First Derivatives (d/dt) for the same quantiles. Row 3: Actual vs. Predicted Second Derivatives (d^2/dt^2). In Test 3, we excluded the second spike region to better compare model performance, as the primary goal of the model is to predict the main spike within an input length of 8000.

The inclusion of these comparisons highlights how well the model captured the dynamics of the signals, not just in the amplitude of the iAPs but also in their rate of change and acceleration (first and second derivatives), even with the challenges presented by noisy data. The model was optimized using a loss function that combines both data-driven and physics-informed terms. The weight factor for the physics-informed part of the loss function was set to 0.05, while the data loss function was weighted by a factor of 10. This weighting was necessary due to the high values present in the physics part of the loss

function, such as d/dt , d^2/dt^2 and $1/v$. By adjusting the relative weights, we ensured smooth training while maintaining adherence to the physical principles that govern the system. These weight factors helped balance the scale of the data loss, which could otherwise dominate the training process due to the larger magnitude of some terms in the physics-informed loss function. This approach not only stabilized the training but also allowed the model to align with the underlying physical dynamics of the signals, ensuring more accurate and reliable predictions. We hope that this clarification addresses your concerns and highlights the robustness of our approach in predicting iAPs, including their first and second derivatives, from extracellular recordings.

14. Why is there a logarithmic transformation of the physics loss in the hybrid loss function?

Response:

Thank you for your question regarding the logarithmic transformation of the physics loss in the hybrid loss function. The logarithmic transformation was applied to address the issue of extremely high values in the physics-informed loss, which can occur due to the presence of derivatives and terms like $1/u$. These large values can cause the loss function to escalate, making the model difficult to train effectively. By applying a logarithmic transformation, we dampen these effects, ensuring a smoother training process and improving model convergence.

15. Please explain the term “quantile crossing” (page 12).

Response:

Quantile crossing happens when predicted values for different quantiles don't follow the natural order. For example, the 0.05 quantile, which represents the lower bound, should always be less than or equal to the 0.5 quantile (the median), and the 0.5 quantile should always be less than or equal to the 0.95 quantile (the upper bound). If these values overlap or flip, it causes a problem because it makes the predictions confusing and unreliable. We have updated the text in the manuscript, and added the explanation

“*Quantile PIA-UNET with confidence interval (Q-PIA-UNET)*. (Q-PIA-UNET) is a modified version of the PIA-UNET architecture designed to estimate the 90% confidence interval for iAP values using a quantile loss function (Ben-Or, Kolomenkin, and Shabat 2020) asymmetrically penalizes over- and underestimation, enabling the model to accurately predict specific quantiles (0.05, 0.5, and 0.95) by minimizing errors for each. To address the issue of quantile crossing, the modified architecture simultaneously predicts the 0.05, 0.5, and 0.95 quantiles through three parallel output layers ($V_{0.05}$, $V_{0.5}$, and $V_{0.95}$), along with corresponding physical parameter predictors. Quantile crossing occurs when the predicted lower, median, and upper quantiles are out of order, which makes the predictions unreliable and inconsistent. By using separate output layers corresponding to each quantile in one model, the model ensures that the predicted values follow the correct order, improving both the accuracy and reliability of the results.

“

16. What filter was used to remove noise from intracellular recordings (page 13)?

Response:

Thank you for your comment. We used a Simple Moving Average (SMA) filter to remove noise from the intracellular recordings. This filter smooths the data by taking the average of values within a sliding

window. The window moves across the data, and at each step, it computes the average of the values in that window. For this study, we used a window size of 20. We have updated the manuscript accordingly to include this detail.

“Action Potential Durations (APDs) calculation. To accurately determine the APDs, we first applied a smoothing technique to the segmented intracellular traces using a moving average filter (window size = 20) to reduce noise and enhance the detectability of the action potential features. Next, we utilized the standard deviation of the smoothed data to identify the initial upward spike of the action potential. For the quantification of the APDs, we employed the `peak_widths` method from the SciPy Python package. This method is particularly effective in measuring the widths of action potentials at various levels of repolarization. We specifically focused on obtaining ten distinct APD measurements to comprehensively describe the shape of each action potential. For instance, APD10 and APD20 represent the widths of the intracellular action potential at 10% and 20% of repolarization, respectively.

17. Please explain “Linear”, “DLinear”, “NLinear”, “MLP” in table legend or material and methods.

Response:

Thank you for your insightful comment. Based on our findings, the performance of **Linear**, **DLinear**, **NLinear**, and **MLP** methods was substantially lower compared to the UNET-based models. To maintain the focus of the manuscript on the most relevant and impactful comparisons, we decided not to include these methods in the updated version. We believe this helps to avoid diverting attention from the primary objective of the study, which is to highlight the strengths of the Physics-Informed Attention UNET and Attention UNET models. For the sake of completeness, we have provided concise explanations of these methods below:

- **Linear**: A baseline model applying a simple linear transformation to the input data.
- **DLinear**: A decomposed linear model that separately processes trend and seasonal components for better time series representation.
- **NLinear**: A model combining linear transformations with nonlinear components for more complex pattern detection.
- **MLP**: A feedforward neural network with multiple layers, often used for capturing both linear and nonlinear relationships in data.

By narrowing the scope to the most relevant comparisons, we aim to provide a clearer and more focused narrative that aligns with the objectives of our research.

18. Figure 1c: Please explain the time course of drug application. Was there a wash between the drug applications (indicated by dashed boxes)?

Response:

Thank you for your comment and for pointing this out. We have updated the figure caption to include the drug application time course. There was no wash performed; however, when a drug was added, the signal became distorted for a few seconds. This distorted segment was excluded from both the training and test sets. The box plot demonstrates that drug addition impacts a range of time during the recording. We have corrected and updated Figure 1 to accurately reflect all drug addition steps.

“Figure 1. Nanoelectrode eAP and iAP data collection and pre-processing. a) Scanning Electron Microscope (SEM) image of an NEA channel with nine nanocrown electrodes, alongside a detailed schematic of the NEA setup. **b)** A methodological comparison for capturing cardiac action potentials, presenting NEA alongside the reference patch clamp technique, ordered by their degree of invasiveness. The NEA’s functionality to convert extracellular action potentials (eAPs) into intracellular action potentials (iAPs) through precise biphasic electric pulses (electroporation) is demonstrated. **c)** Simultaneous iAP recording from neighboring channels via non-invasive extracellular action potential (NEA) electroporation. **This section details the multi-step addition process of Dofetilide, administered in concentrations of 0.3, 1, 3, and finally 10 nM. The process was conducted in three stages at approximately 400, 800, and 1200 seconds during the recording session.** This method was employed to collect a diverse range of iAP shapes, facilitating a more comprehensive analysis. **d)** Comparison process similar to PC vs NEA iAP recording: Scaling between 0 to 1 and segmenting into arrays of length 8000 indices or 1.6s. **e)** Box plot distribution of dCT(s),

correlation coefficient (r), MEA, and APD50% and APD90% errors between neighboring and NEA normalized iAP pairs with NEA $S/N > S/N^*$ ($= 90$). **f)** Examples of iAP pairs from neighboring channels with the highest MEA, APD50, and APD90% errors, as indicated in the box plots. **g)** Collection of diverse iAP waveforms and corresponding eAPs from neighboring channels on NEA in iPSC-CMs cardiac myocytes, following the addition of 10 nM Dofetilide. **h)** This process involves applying electroporation to obtain iAPs and non-electroporation techniques for eAPs. Similar to the PC vs NEA iAP recording method, iAPs are scaled between 0 to 1, but this scaling is different from what was used in the training data. For eAPs, a specific segment of the signal, particularly values between indices 1150 and 1350, is selected to characterize signal noise. The function calculates the standard deviation (std) of this segment. The eAP signal is then processed by subtracting the mean of the entire eAP signal from each value, followed by normalization relative to its noise level. This is achieved by dividing the mean-adjusted values by 60 times the calculated standard deviation. The signals are then segmented into windows of 800 indices or 1.6 seconds. The figure presents a wide range of collected eAPs and iAPs, including an overlay of the non-normalized eAP and iAP pairs.”

19. Figure 2a: What is the origin of these Wenckebach-like rhythms?

Response:

Thank you for your comment. We have not yet fully characterized the biological origin of these rhythms. However, we reported them as an initial insight into how eAP and iAP properties may be related. The overlay of eAP amplitude (representing its maximum voltage during the spiking phase [mv]) with iAP spike velocity (represented as the percentage change in iAP potential during its spiking phase over time [% change in iAP/s]) shows a strong correlation ($r = 0.903$). This association is further supported by the observation that oscillations in the extracellular recordings seem to reflect the action potential's repolarization phase. This observation was possible due to the simultaneous nature of eAP and iAP recording and we have included this as an initial piece of data that motivated our study. We have not used this specific recording in any of our training or test data; In all the data used for our model, the changes in eAP/iAP shape were controlled by adding drugs.

20. Figure 2a: please explain “%APA”, include a proper unit on the x axis and explain the difference between eAP “voltage” and eAP “spike amplitude”

Response:

Thank you for your comment. We have updated the figure caption to address this issue.

“**Figure 2. Quantitative relationships between eAPs and iAP waveform features.** **a)** A representative recording of simultaneous eAP and iAP from arrhythmic cells (top), and an overlay of eAP amplitude (representing its maximum voltage during the spiking phase [mv]) with iAP spike velocity (represented as the percentage change in iAP voltage during its spiking phase over time [% change in iAP/s]), showing a strong correlation ($r = 0.903$). This association is further evident as oscillations in the extracellular recordings appear to reflect the action potential's repolarization phase.”

21. Figure 2d: Why is the distribution of APD values so different between train and test dataset?

Response:

Thank you for your valuable comment regarding the distribution of APD values in Figure 2d. While we used the same drugs for both the training and test datasets, the data were sourced from different experiments and cell cultures, which introduced variability. Specifically, the differences in initial APD values, biological variability, subtle variations in experimental conditions, and the use of lab-derived iPSC cardiomyocytes for NEA data, rather than commercial ones, contributed to this discrepancy. To address this and enhance the generalizability of our study, we have updated the test set to cover a broader range of scenarios and adjusted Figure 2d to better reflect this diversity. This ensures that our model is robust across different experimental conditions and better captures real-world variability.

First, to ensure diversity in the training data, we introduced several ion-channel blockers in a dose-dependent manner to hiPSC-derived cardiomyocytes (hiPSC-CMs), resulting in a range of action potential shapes. The drugs included dofetilide (blocks hERG (IKr) potassium channels), quinidine (blocks Na⁺, K⁺, and Ca²⁺ channels), nifedipine (selectively blocks L-type Ca²⁺ channels), flecainide (blocks Na⁺ channels, and to a lesser extent, K⁺ channels), lidocaine (primarily blocks Na⁺ channels), and propranolol (a beta-blocker that also blocks Na⁺ channels). In total, we collected 2364 synchronized eAP/iAP pairs across ten independent experiments using different cell cultures, wafers, and batches of NEA. For data splitting:

- **Training set:** eAP/iAP recordings from NEA using hiPSC-CMs exposed to dofetilide, quinidine, nifedipine, flecainide, and lidocaine.
- **Test sets:**
 1. **Test 1:** eAP/iAP pairs from hiPSC-CMs exposed to dofetilide (a drug in the training set), but recorded on a different NEA from a separate batch and wafer.
 2. **Test 2:** eAP/iAP pairs from hiPSC-CMs exposed to propranolol (not included in the training set), recorded on a different NEA from a separate batch and wafer.
 3. **Test 3:** eAP/iAP pairs recorded using commercial hiPSC-CMs on a commercial MEA in a different laboratory, with simultaneous patch clamp recordings for iAP validation.

This approach enhances the generalizability of our study by testing across various conditions and experimental setups. We have updated Figure 2d accordingly to reflect these changes, ensuring the results *better capture the variability across different experiments. We have updated the manuscript as follows:*

“To collect a diverse spectrum of synchronized eAP and iAP pairs for our training dataset, we introduced various ion-channel blockers in a dose-dependent manner to hiPSC-derived cardiomyocytes (hiPSC-CMs) to achieve different action potential shapes. The drugs used included dofetilide (primarily blocks hERG (IKr) potassium channels)(Saliba 2001), quinidine (blocks Na⁺ channels (INa), K⁺ channels (IKr and IKs), and Ca²⁺ channels (ICa))(Salata and Wasserstrom 1988; Nenov et al. 1998; Clark et al. 2022), nifedipine (selectively blocks L-type Ca²⁺ channels (ICa-L)) (Hadley and Lederer 1995), flecainide (blocks Na⁺ channels (INa) and, to a lesser extent, K⁺ channels (IKr)) (O’ Brien et al. 2022; Melgari et al. 2015), lidocaine (primarily blocks Na⁺ channels (INa))(Bean, Cohen, and Tsien 1983) and propranolol, a beta-blocker that also blocks Na⁺ channels (INa) (Wang et al. 2010). We collected a total of 2364 synchronized eAP/iAP pairs from independent recording sets (see Methods). Data were obtained from ten independent experiments using different cell cultures on NEA from different wafers and different batches.”

For data splitting:

“The training dataset included eAP/iAP recordings from NEA using hiPSC-CMs exposed to dofetilide, quinidine, nifedipine, flecainide, and lidocaine. To validate the generalizability of our study, three different test sets were used:

1. **Test 1:** hiPSC-CMs exposed to dofetilide, a drug included in the training set, were tested on a different NEA from a separate batch and wafer to obtain eAP/iAP pairs.
2. **Test 2:** Propranolol, not used in the training set, was tested on a different NEA from a separate batch and wafer to obtain eAP/iAP pairs.

3. **Test 3:** eAP/iAP pairs were recorded from a different laboratory using commercial hiPSC-CMs on a commercial MEA (“Pharmacology (cardio),” n.d.). For this test, simultaneous patch clamp recordings of the same cells were used to obtain iAPs.”

We have updated Figure 2d accordingly to reflect these changes, ensuring the results better capture the variability across different experiments.

c

eAP ΔT_1	-0.50	-0.44	-0.30	-0.08	0.13	0.29	0.40	0.46	0.49	0.51
eAP ΔT_2	-0.65	-0.56	-0.37	-0.10	0.16	0.35	0.48	0.54	0.58	0.60
eAP ΔT_3	-0.61	-0.53	-0.35	-0.10	0.15	0.33	0.45	0.52	0.55	0.58
eAP ΔT_d	-0.03	0.21	0.50	0.76	0.91	0.97	0.97	0.96	0.94	0.94
eAP IR	-0.50	-0.45	-0.37	-0.26	-0.14	-0.07	-0.03	-0.01	-0.01	0.00
eAP DR	-0.19	-0.19	-0.23	-0.27	-0.30	-0.32	-0.34	-0.35	-0.36	-0.38
eAP ΔV_1	-0.34	-0.19	-0.07	0.03	0.08	0.10	0.09	0.07	0.05	0.03
eAP ΔV_2	-0.22	-0.14	-0.06	0.01	0.05	0.05	0.04	0.02	0.00	-0.03
eAP ΔV_d	0.08	0.12	0.09	0.03	-0.04	-0.12	-0.18	-0.22	-0.24	-0.27
eAP $\Delta V_i / \Delta V_2$	-0.16	-0.08	-0.04	-0.02	-0.01	-0.01	0.00	0.00	0.00	0.01
eAP $\Delta V_d / \Delta V_2$	0.09	0.07	0.01	-0.04	-0.09	-0.12	-0.13	-0.14	-0.14	-0.14
eAP $\Delta V_d / \Delta V_1$	0.31	0.21	0.06	-0.09	-0.20	-0.28	-0.31	-0.33	-0.34	-0.34
	APD 10	APD 20	APD 30	APD 40	APD 50	APD 60	APD 70	APD 80	APD 90	APD 100

“Figure 2. Quantitative relationships between eAPs and iAP waveform features.”

22. Figure 2b, insert: What is the origin of these distorted spikes? Overlapping signals from adjacent cardiomyocytes that are not fully entrained?

Response:

Thank you for your comment. Yes, it could be due to overlapping signals from adjacent cardiomyocytes that are not fully entrained. Additionally, the lower sampling rate used in this study (5000 Hz) compared to patch clamp studies (typically 10,000 Hz) might also contribute to these distortions. Another possibility is interference from charges that may have accumulated on the nano pillars during the measurements. However, we cannot confidently confirm any specific cause at this point, and further studies are required to investigate these distorted spikes in more detail.

23. Figure 2e: Dashed lines make it difficult to see the end points of the lines.

Response:

Thank you for your comment. We have adjusted Figure 2e to improve visibility by modifying the dashed lines to make the end points clearer. Please check the figure

“Figure 2. Quantitative relationships between eAPs and iAP waveform features.” panels e, f and g.

24. Figure 2e: The examples show the largest differences for APD10, 20, 30 whereas the plot in Figure 2f shows the opposite trend.

Response:

Thank you for your comment. We acknowledge that the examples in Figure 2e showed the largest differences for APD10, 20, and 30, while the plot in Figure 2f suggested the opposite trend. The examples were initially chosen at random; however, we agree with your observation. To address this, we have updated the test set to incorporate a broader range of scenarios and replaced the examples in Figure 2e to better reflect the trends shown in Figure 2f.

25. Figure 2g: The total APD error depends on the somewhat arbitrarily chosen number of levels and is almost as large as the APD itself. An average APD error or maximal APD error might be a better choice.

Response:

Thank you for your comment regarding the total APD error in Figure 2g. We agree that the total APD error can be influenced by the arbitrarily chosen number of levels and that it may not provide the most accurate representation. In response, we have replaced the total APD error with the average APD error in the figure.

26. Figure 4a: Please explain the time course of drug application. Was there a wash between the drug applications (indicated by dashed boxes)?

Response:

Thank you for your comment and for pointing this out. We have updated the figure caption to include the drug application time course. There was no wash performed; however, when a drug was added, the signal became distorted for a few seconds. This distorted segment was excluded from both the training and test sets. The box plot demonstrates that drug addition impacts a range of time during the recording. We have corrected and updated Figure 1 to accurately reflect all drug addition steps.

“Figure 1. Nanoelectrode eAP and iAP data collection and pre-processing. a) Scanning Electron Microscope (SEM) image of an NEA channel with nine nanocrown electrodes, alongside a detailed schematic of the NEA setup. **b)** A methodological comparison for capturing cardiac action potentials, presenting NEA alongside the reference patch clamp technique, ordered by their degree of invasiveness. The NEA’s functionality to convert extracellular action potentials (eAPs) into intracellular action potentials (iAPs) through precise biphasic electric pulses (electroporation) is demonstrated. **c)** Simultaneous iAP recording from neighboring channels via non-invasive extracellular action potential (NEA) electroporation. **This section details the multi-step addition process of Dofetilide, administered in concentrations of 0.3, 1, 3, and finally 10 nM. The process was conducted in three stages at approximately 400, 800, and 1200 seconds during the recording session.** This method was employed to collect a diverse range of iAP shapes, facilitating a more comprehensive analysis. **d)** Comparison process similar to PC vs NEA iAP recording: Scaling between 0 to 1 and segmenting into arrays of length 8000 indices or 1.6s. **e)** Box plot distribution of dCT(s), correlation coefficient (r), MEA, and APD50% and APD90% errors between neighboring and NEA normalized iAP pairs with NEA $S/N > S/N^*$ ($= 90$). **f)** Examples of iAP pairs from neighboring channels with the highest MEA, APD50, and APD90%

errors, as indicated in the box plots. **g)** Collection of diverse *iAP* waveforms and corresponding *eAPs* from neighboring channels on NEA in iPSC-CMs cardiac myocytes, following the addition of 10 nM Dofetilide. **h)** This process involves applying electroporation to obtain *iAPs* and non-electroporation techniques for *eAPs*. Similar to the PC vs NEA *iAP* recording method, *iAPs* are scaled between 0 to 1, but this scaling is different from what was used in the training data. For *eAPs*, a specific segment of the signal, particularly values between indices 1150 and 1350, is selected to characterize signal noise. The function calculates the standard deviation (*std*) of this segment. The *eAP* signal is then processed by subtracting the mean of the entire *eAP* signal from each value, followed by normalization relative to its noise level. This is achieved by dividing the mean-adjusted values by 60 times the calculated standard deviation. The signals are then segmented into windows of 800 indices or 1.6 seconds. The figure presents a wide range of collected *eAPs* and *iAPs*, including an overlay of the non-normalized *eAP* and *iAP* pairs.”

27. Supplemental information, Figure S1-3a: The legend mentions 4 different drug concentrations, but only 3 drug applications are shown.

Response:

Thank you for pointing this out. We have reviewed the figure and now we have now corrected the figure to accurately reflect all four drug applications, as mentioned in the legend. We appreciate your careful review and attention to detail.

28. Supplemental information, page 9: Please write out the abbreviation LTFS by the first time you use it.

Response:

Thank you for your insightful comment. Based on our findings, the performance of **Linear**, **DLinear**, **NLinear**, and **MLP** methods was substantially lower compared to the UNET-based models. To maintain the focus of the manuscript on the most relevant and impactful comparisons, we decided not to include these methods in the updated version. We believe this helps to avoid diverting attention from the primary objective of the study, which is to highlight the strengths of the Physics-Informed Attention UNET and

Attention UNET models. For the sake of completeness, we have provided concise explanations of these methods below:

- **Linear**: A baseline model applying a simple linear transformation to the input data.
- **DLinear**: A decomposed linear model that separately processes trend and seasonal components for better time series representation.
- **NLinear**: A model combining linear transformations with nonlinear components for more complex pattern detection.
- **MLP**: A feedforward neural network with multiple layers, often used for capturing both linear and nonlinear relationships in data.

By narrowing the scope to the most relevant comparisons, we aim to provide a clearer and more focused narrative that aligns with the objectives of our research.

29. Although it is not overly complicated, some readers might find it helpful if a step-by-step derivation for formula (14) were included in the supporting information.

Response:

Thank you for your comment. We have added that to the Supplementary Information.

“1. Model Equations

We start with the following governing equations:

$$\frac{dv}{dt} = kv(1 - u)(u - a) - uw \quad (1)$$

$$\frac{dw}{dt} = \varepsilon(v, a, x)(ku - w) \quad (2)$$

with $\varepsilon(v, a, x)$ approximated by:

$$\varepsilon(v, a, x) \approx x \cdot \sigma(n(a - v)) + (1 - x) \cdot \sigma(n(v - a)) \quad (3)$$

2. Isolating w from Equation 2

We solve for w from Equation 12:

$$w = \frac{kv(1 - v)(v - a) - \frac{dv}{dt}}{v}$$

3. Substituting w into Equation 3

$$\frac{d}{dt} \left[\frac{kv(1 - v)(v - a) - \frac{dv}{dt}}{v} \right] = \varepsilon(v, a, x) \left(kv - \frac{kv(1 - v)(v - a) - \frac{dv}{dt}}{v} \right)$$

For $v \neq 0$, we divide by v and simplify.

$$\frac{d}{dt} \left[k(1 - v)(v - a) - v^{-1} \frac{dv}{dt} \right] = \varepsilon(v, a, x) \left(kv - \left[k(1 - v)(v - a) - v^{-1} \frac{dv}{dt} \right] \right)$$

4. Simplification of the Left Side

$$\frac{d}{dt} [k(1 - v)(v - a) - v^{-1} \frac{dv}{dt}] = k \frac{d(v-a-v^2+a.v)}{dt} - \frac{d[v^{-1} \frac{dv}{dt}]}{dt}$$

We differentiate each term:

$$k \frac{d}{dt} [(1 - v)(v - a)] = k \left(\frac{dv}{dt} - 2v \frac{dv}{dt} + a \frac{dv}{dt} \right) (*)$$

and

$$- \frac{d[v^{-1} \frac{dv}{dt}]}{dt} = - \left(- \frac{dv}{dt} \frac{dv}{dt} v^{-2} + v^{-1} \frac{d^2 v}{dt^2} \right) (**)$$

Putting these together:

$$\frac{d}{dt} [k(1 - v)(v - a) - v^{-1} \frac{dv}{dt}] = k \left[\frac{dv}{dt} - 2v \frac{dv}{dt} + a \frac{dv}{dt} \right] + \left(\frac{dv}{dt} \right)^2 v^{-2} - v^{-1} \frac{d^2 v}{dt^2}$$

This simplifies to:

$$= k \frac{dv}{dt} [1 - 2v + a] + \left(\frac{dv}{dt} \right)^2 v^{-2} - v^{-1} \frac{d^2 v}{dt^2}$$

5. Putting Together the Left and Right Sides

$$F_{AP} \left(v, \frac{dv}{dt}, \frac{d^2 v}{dt^2}, a, k \right) = v^{-2} \left(\frac{dv}{dt} \right)^2 - v^{-1} \frac{d^2 v}{dt^2} - \frac{dv}{dt} [\varepsilon(v) v^{-1} - k(1 - 2v + a)] - k \varepsilon(v) (v^2 - av + a) = 0$$

“

Reviewer #4 (Remarks to the Author):

The authors report a novel methodology for electrophysiological characterization of stem cell derived cardiomyocytes using nanoelectrode arrays, with the support of artificial intelligence tools. The interesting part of methodology is the noninvasive reconstruction of cardiac Intracellular Action Potentials (iAPs) from the external Actio. Potentials (eAP) recordings. The integration of physics-informed deep learning to reconstruct iAPs from eAPs is novel and addresses the limitations of both invasive patch-clamp techniques and lower-fidelity extracellular recordings. While the results are promising, there are significant weaknesses that need to be addressed through revisions of the manuscript and inclusion of additional data.

MAJOR CONCERNS

1. The biological component of this research is largely neglected. It is mentioned that studies were done using “stem cell derived cardiomyocytes”, without specifying what are the stem cells (human or animal; embryonic or iPSC cells), and how many lineages were used. I was unable to find a method section with detailed disclosure of the methodology. Even if broadly studied cell source and a published methodology were used, it is critical to provide extensive characterization of the molecular, ultrastructural and functional (contractility, electrophysiology, calcium handling) of cardiomyocytes and determine the level of their maturity.

Response:

We thank the reviewer for this comment. In this study we used two types of hiPSCs and both have detailed published methodologies. The data used in this study comes from human induced pluripotent stem cell (iPSC)-derived cardiomyocytes. For the microelectrode array (MEA) recordings, we utilized commercially available Human iPSC-derived ventricular cardiomyocytes (Celo.Cardiomyocytes, Celogics, CAT # C50), which are specifically ventricular-like. For the nanoelectrode array (NEA) recordings, we employed human iPSC-derived cardiomyocytes that were heterogeneously composed, with a predominance of ventricular-like cells (57%), as well as atrial-like and nodal-like cells. These heterogeneous cardiomyocytes were recorded at Stanford University. The derivation of these cells followed a protocol approved by the Stanford University Human Subjects Research Institutional Review Board, which ensures ethical standards and informed consent from participants (Burrige et al. 2014). We have now included a more detailed description of the cells and also added references to the published methodologies in our methods section.

“Fabrication of Nano Electrode Arrays

The fabrication process is fully explained in the paper by Jahed et al³⁵. Briefly, we used maskless photolithography followed by deep reactive ion etching to develop vertical SiO₂ nanopillars, which were then coated with Pt metal to achieve conductivity. The metal was etched from the tip of the pillars using a directional dry etch to achieve the nano crown shape.

hiPSC-CMs culture.

The hiPSC-CMs differentiation and culture process were described previously in detail by Jahed et al.³⁵. Briefly, hiPSCs (line SCVI-273) were treated with 6 μM CHIR99021 (Selleck Chemical) in RPMI supplemented with B27 without insulin for 2 days, followed by recovery in RPMI supplemented with B27 without insulin for 1 day, and then by treatment with 5 μM IWR-1 (Selleck Chemical) for 2 days. After recovery in fresh RPMI plus B27 without insulin medium for 2 days, cells were switched to RPMI plus B27 with insulin for 2 days. The hiPSC-CMs were purified with glucose-free RPMI plus B27 with insulin medium for 2–4 days and maintained in RPMI plus B27 with insulin medium for subsequent experiments. These cells were demonstrated to be heterogeneous with ventricular-like cells being the predominant (57%) along with atrial-like and nodal-like cells³⁶. Prior to plating cells on NEA devices, the device was coated with 1 mg/ml poly-L-Lysine at room temperature for 15 min, then treated with 0.5% Glutaraldehyde in PBS at room temperature for 10 min, followed by 1:200 Matrigel in DMEM/F12 at 37 °C for 3 hours before seeding cells. The cultured hiPSC-CM were disassociated from the plate with TryPLE select 10X at 37°C for 5 min after 25-60 days of differentiation. Cells were resuspended in culture medium supplemented with 10% KnockOut Serum Replacement (KSR), then seeded at ~1.2 x10⁵ cells/device. Measurements were taken for over a month from 5 days post cell attachment.

For MEA recordings, commercial hiPSC-ventricular cardiomyocytes (Celo.Cardiomyocytes, Celogics, CAT # C50) were used. The cells were thawed and prepared according to the manufacturer’s protocol. These cardiomyocytes were derived from a proprietary human iPSC line (fibroblast, Caucasian male donor) and exhibited spontaneous beating starting on day 2 post-thaw, with stabilization at 45-60 beats

per minute (bpm) by day 7. Celo.Cardiomyocytes formed synchronous monolayers and expressed ventricular cardiomyocyte-specific markers, such as connexin 43, indicating high interconnectivity. These cells also demonstrated physiologically relevant electrophysiological properties, as evidenced by their response to various ion channel modulators.”

2. Related to the previous comment, it would be also important to document reproducibility of the measurements both within the same cell preparation and batch to batch.

Response:

Thank you for your comment. Indeed, there are several batch-to-batch variabilities, and even heterogeneity within a single culture in the electrophysiological recordings of hiPSC-CMs, further highlighting the need for high-throughput approaches like the one proposed in our manuscript to obtain statistically significant results. The batch-to-batch differences can result from slight variations in differentiation protocols, cell densities, and minor changes in temperature and humidity. Although these factors are important to consider when characterizing hiPSC-CMs for disease modeling and drug screening, they are actually beneficial to our study as they provide a range of biological phenotypes that similarly impact both the eAP and iAP. Our new results included in our revised manuscript demonstrate that our model can accurately predict iAP across various cell batches under different conditions.

3. In Figure 2h, through SHAP, ΔT_d , DR, and IR were identified as the most important features to predict APD (especially for $> APD_{50}$). How do these features correspond to the biological cardiac action potential? Are the minima (labeled as bp3) generally observed in all NEA eAP recordings? If so, is this feature an empirically determined one or one based on known cardiac biology? Was there any feature selection prior to training the XGBoost algorithm?

Response:

Thank you for your comments. Explaining the extracellular action potential (eAP) features through direct biological interpretation can be challenging, as the primary goal of our study was to establish a relationship between eAPs and intracellular action potentials (iAPs). However, based on the performance of the XGBoost algorithm and the SHAP value analysis, we identified that ΔT_d (decay time between bp₂ and bp₃) is a significant predictor for higher action potential durations (APD) such as APD₅₀, APD₇₀, and APD₉₀. This suggests that longer decay times are associated with higher APD values, which aligns with the biological understanding that longer repolarization phases correlate with extended action potential durations.

On the other hand, ΔT_s , which corresponds to the length of the eAP, appears to be a critical factor in predicting lower APD values, such as APD₃₀. Typically, a higher ΔT_s is associated with a lower APD₃₀. However, it also shows a slight positive effect on higher APD values, such as APD₅₀ to APD₉₀. This nuanced behavior reflects the complex relationship between these eAP features and the underlying cardiac electrophysiological processes.

Regarding the observation of bp₃, it was consistently observed in high-quality recordings from both nanoelectrode arrays (NEA) and microelectrode arrays (MEA). However, in cases where recordings were

extremely noisy, bp_3 might not be detectable. The identification and selection of these features were empirical, based on our goal of accurately reconstructing eAPs and understanding their relationship to iAPs. We defined specific breakpoints such as bp_1 , bp_2 , and bp_3 , which correspond to different phases of the eAP, including the pre- and post-spike phases and the end of repolarization. These breakpoints were chosen based on their correlation with APD values, as demonstrated in Figure 2c. There was no additional feature selection process beyond defining these points, as our aim was to use these defined features to reconstruct the eAP and explore its relationship with iAPs. For feature selection, we studied the correlations between ΔT_1 , ΔT_2 , and ΔT_s . The data showed that while the first two are limited to non-distorted spikes, they exhibit a high correlation with ΔV s. Therefore, we decided to use ΔV s as the primary feature instead."

"Section IV: eAP features screening.

Distorted spikes in eAP signals, as shown in Figure 2b, and the strong correlation between ΔT_1 and ΔT_2 with ΔT_s ($r = 0.93$ and $r = 0.87$ respectively), led us to opt ΔT_s the representative of the eap spike temporal features.

a Correlation Heatmap Between ΔT_1 , ΔT_2 , ΔT_s and ΔT_d in data with Undistorted eAP Spikes

b Correlation Heatmap Between eAP features in data including Both Distorted and Undistorted eAP Spikes

Figure-SI-6. **a)** Correlation between temporal-related eAP features (ΔT_1 , ΔT_2 , ΔT_s , and ΔT_d) for the undistorted portion of eAP/iAP data collected to be utilized in the machine learning and deep learning models. **b)** Correlation between measurable eAP features (excluding ΔT_1 and ΔT_2) for all of the eAP/iAP data (with distorted and undistorted eap spikes) utilized in the machine learning and deep learning models.”

4. Are the datasets used for the feature-based model in Figure 2 the same as the ones used in Figure 3? If so, it seems like the feature-based approach is better (APD Error = 0.196s) than the proposed PIA-UNET (APD Error = 0.25s). What is the added benefit of the DL approach? It also seems that the DL approach might have an overfitting issue, with APD error of the test set diverging significantly from the train set.

Response:

Thank you for your comment. Yes, the datasets used in Figure 2 and Figure 3 are from the same source. However, for the XGBoost method, we applied screening methods to only retain signals where the critical points were computed accurately by our algorithm and exhibited measurable ΔT s in their spikes. While XGBoost demonstrated robust performance, it is limited to predicting APD values and cannot reconstruct the entire iAP shape. This makes it suitable for simple studies but insufficient for more comprehensive analysis. Additionally, XGBoost requires intensive data preparation to extract waveform features, whereas the deep learning (DL) method does not. The DL approach can also handle distorted spikes, providing greater versatility.

5. Dofetilide was used to generate a more diverse set of waveform data to train the model. In validating the model, the authors use the same drug to demonstrate the reconstruction of iAP using eAP recordings. An additional drug other than dofetilide would validate the method in a more robust manner. On a similar note, does the model generalize to other classes of cardiac drugs that affect the AP waveform differently (e.g. in the QRS phases/depolarization rather than QT)? Investigating the effects of another set of drugs (such as a Na class IC blocker that perturbs the QRS phase or a class Ib drug that shortens QT) would strengthen the generalizability of this model to capture perturbations to additional parts of the AP waveform.

Response:

Thank you for your insightful comment. We appreciate the suggestion to include additional drugs and drug classes to better validate the generalizability of our model. Based on your feedback, we have expanded our study to incorporate more diverse drugs in both the training and test sets, including drugs that affect different phases of the action potential (AP).

To ensure a broad spectrum of synchronized eAP and iAP pairs for our training dataset, we introduced various ion-channel blockers in a dose-dependent manner to hiPSC-derived cardiomyocytes (hiPSC-CMs), which allowed us to achieve a wide range of action potential shapes. The drugs used in the training dataset included:

6. **Dofetilide:** A Class III antiarrhythmic drug, dofetilide primarily blocks the hERG (IKr)³⁷ potassium channels. By inhibiting this channel, it prolongs the repolarization phase, resulting in an increased action potential duration (APD) and a longer QT interval (Jaiswal and Goldbarg 2014).
7. **Quinidine:** As a Class IA antiarrhythmic, quinidine blocks sodium (Na⁺) channels (INa), effectively depressing the rapid initial depolarization phase of the action potential. This mechanism reduces the sodium current and affects the overall electrical activity of the heart

- (Imaizumi and Giles 1987)
8. **Nifedipine:** A calcium channel blocker in the dihydropyridine class, nifedipine selectively blocks L-type calcium channels (ICa-L)⁴¹. This results in a shortened plateau phase of the action potential, leading to a reduced action potential duration (Zemzemi and Rodriguez 2015)
 9. **Flecainide:** A Class IC antiarrhythmic drug, flecainide blocks both Na⁺ channels (INa) and, to a lesser extent, potassium (IKr) channels^{42,43}. Its overall effect is to prolong the duration of the action potential and the effective refractory period in ventricular fibers, while both are shortened in the Purkinje system, which is likely consistent with its sodium channel blockade (Lavalley et al. 2021)
 10. **Lidocaine:** A Class IB antiarrhythmic, lidocaine primarily blocks Na⁺ channels (INa)⁴⁴. It may shorten the APD, though the effect observed in the experiments varied based on the sample conditions.
 11. **Propranolol:** Known as a beta-blocker⁴⁵ and classified as a Class II antiarrhythmic, propranolol also exhibits sodium (INa) channel blocking properties, further influencing cardiac electrical activity (Kirkorian et al. 1988). It can result in shortening APD 90, by increasing the repolarization slope(Hirose et al. 2020).

In addition to these, we incorporated recordings from an unseen device, the commercial Microelectrode Array (MEA), instead of the NEA, using different commercially available hiPSC-derived cardiomyocytes. This was done to further test the model's generalizability. We collected a total of 2364 synchronized eAP/iAP pairs from ten independent experiments using different cell cultures on NEA, sourced from different wafers and batches. For the validation of our study, we used three distinct test sets:

- **Test 1:** hiPSC-CMs exposed to dofetilide (a drug included in the training set) were tested on a different NEA from a separate batch and wafer to obtain eAP/iAP pairs.
- **Test 2:** Propranolol (not used in the training set) was tested on a different NEA from a separate batch and wafer to obtain eAP/iAP pairs.
- **Test 3:** eAP/iAP pairs were recorded from a different laboratory using commercial hiPSC-CMs on a commercial MEA(“Pharmacology (cardio),” n.d.). For this test, simultaneous patch clamp recordings of the same cells were used to obtain iAPs.

By including a broader range of drugs and an unseen device in our testing, we have aimed to enhance the robustness and generalizability of our model, ensuring it can capture perturbations to various parts of the AP waveform, not just the QT interval.

MINOR COMMENTS AND SUGGESTIONS

12. References to Fig 1A and B are missing in the manuscript.

Response:

Thank you for your comment. We have addressed the issue and updated the manuscript accordingly

13. In the model evaluations shown in Fig 2d-g and Fig 3c-g, it would be more compelling to resplit the dataset from the first 2 sets of experiments to create another test set rather than comparing the train set (which I assume the model has already seen before) vs test set as the authors currently show. The current test set is from an independent 3rd experiment set, which is great, but it would be even more compelling to assess performance differences between data from the first 2 sets of experiments that the model has not yet seen compared to the current test set to assess generalizability across experimental setup and hardware.

Response:

Thank you for your insightful comment. To enhance the evaluation of our model's generalizability, we expanded our dataset and incorporated three distinct test sets corresponding to different scenarios which represent potential real life tests sets:

- **Test 1:** hiPSC-CMs exposed to dofetilide (a drug included in the training set) were tested on a different NEA from a separate batch and wafer to obtain eAP/iAP pairs.
- **Test 2:** Propranolol (a drug not used in the training set) was tested on a different NEA from a separate batch and wafer to obtain eAP/iAP pairs.
- **Test 3:** eAP/iAP pairs were recorded from a different laboratory using commercial hiPSC-CMs on a commercial MEA. For this test, simultaneous patch clamp recordings of the same cells were used to obtain iAPs.

This approach allowed us to validate our model's performance across multiple setups and hardware, providing a comprehensive assessment of its generalizability. We believe that these varied test scenarios effectively demonstrate the model's robustness and adaptability to different experimental conditions.

14. Please include the number of points in the correlation plot in 3C, as the coefficient of determination is dependent on the number of plotted points.

Response:

Thank you for your comment. We have updated the figure caption to include the number of points in the correlation plot: *“c) This shows the model's predicted potential values closely aligning with the actual values, with a coefficient of determination ($r = 0.99$) based on 421, 187, and 149 samples for the three respective test sets. “*

15. Please add a color legend for Fig 4F.

Response:

Thank you for your comment. We have fixed that.

Reviewer #5 (Remarks to the Author):

I co-reviewed this manuscript with one of the reviewers who provided the listed reports. This is part of the Nature Communications initiative to facilitate training in peer review and to provide appropriate recognition for Early Career Researchers who co-review manuscripts. Here are my additional comments:

The paper presents a promising methodology in electrophysiology by combining high-throughput NEA recordings with a physics-informed loss to train the DL model to enable noninvasive reconstruction of cardiac iAPs using eAP recordings. The integration of physics-informed deep learning to reconstruct iAPs from eAPs is novel and addresses the limitations of both invasive patch-clamp techniques and lower-fidelity extracellular recordings. Addressing the identified weaknesses and conducting additional experiments will further strengthen the methodology and broaden its applicability.

1. References to Fig 1A and B are missing in the manuscript.

Response:

Thank you for pointing this out. We have now added references to Figures 1A and 1B in the manuscript.

2. In Figure 2h, through SHAP, ΔT_d , DR, and IR were identified as the most important features to predict APD (especially for $> APD_{50}$). How do these features correspond to the biological cardiac action potential? Are the minima (labeled as bp3) generally observed in all NEA eAP recordings? If so, is this feature an empirically determined one or one based on known cardiac biology? Was there any feature selection prior to training the XGBoost algorithm?

Response:

Thank you for your comments. Explaining the extracellular action potential (eAP) features through direct biological interpretation can be challenging, as the primary goal of our study was to establish a relationship between eAPs and intracellular action potentials (iAPs). However, based on the performance of the XGBoost algorithm and the SHAP value analysis, we identified that ΔT_d (decay time between bp₂ and bp₃) is a significant predictor for higher action potential durations (APD) such as APD₅₀, APD₇₀, and APD₉₀. This suggests that longer decay times are associated with higher APD values, which aligns with the biological understanding that longer repolarization phases correlate with extended action potential durations.

On the other hand, ΔT_s , which corresponds to the length of the eAP, appears to be a critical factor in predicting lower APD values, such as APD₃₀. Typically, a higher ΔT_s is associated with a lower APD₃₀. However, it also shows a slight positive effect on higher APD values, such as APD₅₀ to APD₉₀. This nuanced behavior reflects the complex relationship between these eAP features and the underlying cardiac electrophysiological processes.

Regarding the observation of bp₃, it was consistently observed in high-quality recordings from both nanoelectrode arrays (NEA) and microelectrode arrays (MEA). However, in cases where recordings were extremely noisy, bp₃ might not be detectable. The identification and selection of these features were empirical, based on our goal of accurately reconstructing eAPs and understanding their relationship to

iAPs. We defined specific breakpoints such as bp_1 , bp_2 , and bp_3 , which correspond to different phases of the eAP, including the pre- and post-spike phases and the end of repolarization. These breakpoints were chosen based on their correlation with APD values, as demonstrated in Figure 2c. There was no additional feature selection process beyond defining these points, as our aim was to use these defined features to reconstruct the eAP and explore its relationship with iAPs. For feature selection, we studied the correlations between ΔT_1 , ΔT_2 , and ΔT_s . The data showed that while the first two are limited to non-distorted spikes, they exhibit a high correlation with ΔV s. Therefore, we decided to use ΔV s as the primary feature instead."

"Section IV: eAP features screening.

Distorted spikes in eAP signals, as shown in Figure 2b, and the strong correlation between ΔT_1 and ΔT_2 with ΔT_s ($r = 0.93$ and $r = 0.87$ respectively), led us to opt ΔT_s the representative of the eap spike temporal features.

a Correlation Heatmap Between ΔT_1 , ΔT_2 , ΔT_s and ΔT_d in data with Undistorted eAP Spikes

b Correlation Heatmap Between eAP features in data including Both Distorted and Undistorted eAP Spikes

Figure-SI-6. **a)** Correlation between temporal-related eAP features (ΔT_1 , ΔT_2 , ΔT_s , and ΔT_d) for the undistorted portion of eAP/iAP data collected to be utilized in the machine learning and deep learning models. **b)** Correlation between measurable eAP features (excluding ΔT_1 and ΔT_2) for all of the eAP/iAP data (with distorted and undistorted eap spikes) utilized in the machine learning and deep learning models.”

3. Are the datasets used for the feature-based model in Figure 2 the same as the ones used in Figure 3? If so, it seems like the feature-based approach is better (APD Error = 0.196s) than the proposed PIA-UNET (APD Error = 0.25s). What is the added benefit of the DL approach? It also seems that the DL approach might have an overfitting issue, with APD error of the test set diverging significantly from the train set.

Response:

Thank you for your comment. Yes, the datasets used in Figure 2 and Figure 3 are from the same source. However, for the XGBoost method, we applied screening methods to only retain signals where the critical points were computed accurately by our algorithm and exhibited measurable ΔT s in their spikes. While XGBoost demonstrated robust performance, it is limited to predicting APD values and cannot reconstruct the entire iAP shape. This makes it suitable for simple studies but insufficient for more comprehensive analysis. Additionally, XGBoost requires intensive data preparation to extract waveform features, whereas the deep learning (DL) method does not. The DL approach can also handle distorted spikes, providing greater versatility.

4. In the model evaluations shown in Fig 2d-g and Fig 3c-g, it would be more compelling to resplit the dataset from the first 2 sets of experiments to create another test set rather than comparing the train set (which I assume the model has already seen before) vs test set as the authors currently show. The current test set is from an independent 3rd experiment set, which is great, but it would be even more compelling to assess performance differences between data from the first 2 sets of experiments that the model has not yet seen compared to the current test set to assess generalizability across experimental setup and hardware.

Response:

Thank you for your insightful comment. To enhance the evaluation of our model's generalizability, we expanded our dataset and incorporated three distinct test sets corresponding to different scenarios which represent potential real life tests sets:

- **Test 1:** hiPSC-CMs exposed to dofetilide (a drug included in the training set) were tested on a different NEA from a separate batch and wafer to obtain eAP/iAP pairs.
- **Test 2:** Propranolol (a drug not used in the training set) was tested on a different NEA from a separate batch and wafer to obtain eAP/iAP pairs.
- **Test 3:** eAP/iAP pairs were recorded from a different laboratory using commercial hiPSC-CMs on a commercial MEA. For this test, simultaneous patch clamp recordings of the same cells were used to obtain iAPs.

This approach allowed us to validate our model's performance across multiple setups and hardware, providing a comprehensive assessment of its generalizability. We believe that these varied test scenarios effectively demonstrate the model's robustness and adaptability to different experimental conditions.

5. Please include the number of points in the correlation plot in 3C, as the coefficient of determination is dependent on the number of plotted points.

Response:

Thank you for your comment. We have updated the figure caption to include the number of points in the correlation plot: “*c*) This shows the model's predicted potential values closely aligning with the actual values, with a coefficient of determination ($r = 0.99$) based on 421, 187, and 149 samples for the three respective test sets. “

6. Please add a color legend for Fig 4F.

Response:

Thank you for your comment. We have fixed that.

7. Dofetilide was used to generate a more diverse set of waveform data to train the model. In validating the model, the authors use the same drug to demonstrate the reconstruction of iAP using eAP recordings. An additional drug other than dofetilide or a disease model would validate the method in a more robust manner. On a similar note, does the model generalize to other classes of cardiac drugs that affect the AP waveform differently (e.g. in the QRS phases/depolarization rather than QT)? Investigating the effects of another set of drugs (such as a Na class IC blocker that perturbs the QRS phase or a class Ib drug that shortens QT) would strengthen the generalizability of this model to capture perturbations to additional parts of the AP waveform.

Response:

Thank you for your insightful comment. We appreciate the suggestion to include additional drugs and drug classes to better validate the generalizability of our model. Based on your feedback, we have expanded our study to incorporate more diverse drugs in both the training and test sets, including drugs that affect different phases of the action potential (AP).

To ensure a broad spectrum of synchronized eAP and iAP pairs for our training dataset, we introduced various ion-channel blockers in a dose-dependent manner to hiPSC-derived cardiomyocytes (hiPSC-CMs), which allowed us to achieve a wide range of action potential shapes. The drugs used in the training dataset included:

4. **Dofetilide:** A Class III antiarrhythmic drug, dofetilide primarily blocks the hERG (IKr)³⁷ potassium channels. By inhibiting this channel, it prolongs the repolarization phase, resulting in an increased action potential duration (APD) and a longer QT interval (Jaiswal and Goldberg 2014).
5. **Quinidine:** As a Class IA antiarrhythmic, quinidine blocks sodium (Na⁺) channels (INa), effectively depressing the rapid initial depolarization phase of the action potential. This mechanism reduces the sodium current and affects the overall electrical activity of the heart (Imaizumi and Giles 1987).
6. **Nifedipine:** A calcium channel blocker in the dihydropyridine class, nifedipine selectively blocks L-type calcium channels (ICa-L)⁴¹. This results in a shortened plateau phase of the action potential, leading to a reduced action potential duration (Zemzemi and Rodriguez 2015).
7. **Flecainide:** A Class IC antiarrhythmic drug, flecainide blocks both Na⁺ channels (INa) and, to a lesser extent, potassium (IKr) channels^{42,43}. Its overall effect is to prolong the duration of the action potential and the effective refractory period in ventricular fibers, while both are shortened in the Purkinje system, which is likely consistent with its sodium channel blockade (Lavalle et al. 2021).
8. **Lidocaine:** A Class IB antiarrhythmic, lidocaine primarily blocks Na⁺ channels (INa)⁴⁴. It may shorten the APD, though the effect observed in the experiments varied based on the sample conditions.
9. **Propranolol:** Known as a beta-blocker⁴⁵ and classified as a Class II antiarrhythmic, propranolol also exhibits sodium (INa) channel blocking properties, further influencing cardiac electrical activity (Kirkorian et al. 1988). It can result in shortening APD 90, by increasing the repolarization slope(Hirose et al. 2020).

In addition to these, we incorporated recordings from an unseen device, the commercial Microelectrode Array (MEA), instead of the NEA, using different commercially available hiPSC-derived cardiomyocytes. This was done to further test the model's generalizability. We collected a total of 2364 synchronized eAP/iAP pairs from ten independent experiments using different cell cultures on NEA, sourced from different wafers and batches. For the validation of our study, we used three distinct test sets:

- **Test 1:** hiPSC-CMs exposed to dofetilide (a drug included in the training set) were tested on a different NEA from a separate batch and wafer to obtain eAP/iAP pairs.
- **Test 2:** Propranolol (not used in the training set) was tested on a different NEA from a separate batch and wafer to obtain eAP/iAP pairs.
- **Test 3:** eAP/iAP pairs were recorded from a different laboratory using commercial hiPSC-CMs on a commercial MEA(“Pharmacology (cardio),” n.d.). For this test, simultaneous patch clamp recordings of the same cells were used to obtain iAPs.

By including a broader range of drugs and an unseen device in our testing, we have aimed to enhance the robustness and generalizability of our model, ensuring it can capture perturbations to various parts of the AP waveform, not just the QT interval.

References

- Bean, B. P., C. J. Cohen, and R. W. Tsien. 1983. "Lidocaine Block of Cardiac Sodium Channels." *The Journal of General Physiology* 81 (5): 613–42.
- Ben-Or, Dvir, Michael Kolomenkin, and G. Shabat. 2020. "Generalized Quantile Loss for Deep Neural Networks." *ArXiv abs/2012.14348* (December).
<https://www.academia.edu/download/76340201/2012.14348v1.pdf>.
- Burridge, Paul W., Elena Matsa, Praveen Shukla, Ziliang C. Lin, Jared M. Churko, Antje D. Ebert, Feng Lan, et al. 2014. "Chemically Defined Generation of Human Cardiomyocytes." *Nature Methods* 11 (8): 855–60.
- Clark, Alexander P., Siyu Wei, Darshan Kalola, Trine Krogh-Madsen, and David J. Christini. 2022. "An in Silico-in Vitro Pipeline for Drug Cardiotoxicity Screening Identifies Ionic pro-Arrhythmia Mechanisms." *British Journal of Pharmacology* 179 (20): 4829–43.
- Hadley, R. W., and W. J. Lederer. 1995. "Nifedipine Inhibits Movement of Cardiac Calcium Channels through Late, but Not Early, Gating Transitions." *The American Journal of Physiology* 269 (5 Pt 2): H1784–90.
- Hirose, Sayako, Takeru Makiyama, Dario Melgari, Yuta Yamamoto, Yimin Wuriyanghai, Fumika Yokoi, Suguru Nishiuchi, et al. 2020. "Propranolol Attenuates Late Sodium Current in a Long QT Syndrome Type 3-Human Induced Pluripotent Stem Cell Model." *Frontiers in Cell and Developmental Biology* 8 (August):761.
- Hortigon-Vinagre, M. P., V. Zamora, F. L. Burton, J. Green, G. A. Gintant, and G. L. Smith. 2016. "The Use of Ratiometric Fluorescence Measurements of the Voltage Sensitive Dye Di-4-ANEPPS to Examine Action Potential Characteristics and Drug Effects on Human Induced Pluripotent Stem Cell-Derived Cardiomyocytes." *Toxicological Sciences: An Official Journal of the Society of Toxicology* 154 (2): 320–31.
- Imaizumi, Y., and W. R. Giles. 1987. "Quinidine-Induced Inhibition of Transient Outward Current in Cardiac Muscle." *The American Journal of Physiology* 253 (3 Pt 2): H704–8.
- Jahed, Zeinab, Yang Yang, Ching-Ting Tsai, Ethan P. Foster, Allister F. McGuire, Huaxiao Yang, Aofei Liu, et al. 2022. "Nanocrown Electrodes for Parallel and Robust Intracellular Recording of Cardiomyocytes." *Nature Communications* 13 (1): 2253.
- Jaiswal, Abhishek, and Seth Goldberg. 2014. "Dofetilide Induced Torsade de Pointes: Mechanism, Risk Factors and Management Strategies." *Indian Heart Journal* 66 (6): 640–48.
- Kirkorian, G., P. Touboul, and G. Atallah. 1988. "Electrophysiologic Effects of Propranolol in Intraventricular Conduction Disturbance." *The American Journal of Cardiology* 61 (4): 341–45.
- Lavalle, Carlo, Michele Magnocavallo, Martina Straito, Luca Santini, Giovanni Battista Forleo, Massimo Grimaldi, Roberto Badagliacca, Luigi Lanata, and Renato Pietro Ricci. 2021. "Flecainide How and When: A Practical Guide in Supraventricular Arrhythmias." *Journal of Clinical Medicine Research* 10 (7). <https://doi.org/10.3390/jcm10071456>.
- Liu, Pei, and Evan W. Miller. 2020. "Electrophysiology, Unplugged: Imaging Membrane Potential with Fluorescent Indicators." *Accounts of Chemical Research* 53 (1): 11–19.
- Melgari, Dario, Yihong Zhang, Aziza El Harchi, Christopher E. Dempsey, and Jules C. Hancox. 2015. "Molecular Basis of hERG Potassium Channel Blockade by the Class Ic Antiarrhythmic Flecainide." *Journal of Molecular and Cellular Cardiology* 86 (September):42–53.
- Nenov, N. I., W. J. Crumb Jr, J. D. Pigott, L. H. Harrison Jr, and C. W. Clarkson. 1998.

- “Quinidine Interactions with Human Atrial Potassium Channels: Developmental Aspects.” *Circulation Research* 83 (12): 1224–31.
- O’ Brien, Sian, Andrew P. Holmes, Daniel M. Johnson, S. Nashitha Kabir, Christopher O’ Shea, Molly O’ Reilly, Adelisa Avezzu, et al. 2022. “Increased Atrial Effectiveness of Flecainide Conferred by Altered Biophysical Properties of Sodium Channels.” *Journal of Molecular and Cellular Cardiology* 166 (May):23–35.
- “Pharmacology (cardio).” n.d. Accessed September 2, 2024.
<https://www.multichannelsystems.com/content/pharmacology-cardio>.
- Salata, J. J., and J. A. Wasserstrom. 1988. “Effects of Quinidine on Action Potentials and Ionic Currents in Isolated Canine Ventricular Myocytes.” *Circulation Research* 62 (2): 324–37.
- Saliba, W. I. 2001. “Dofetilide (Tikosyn): A New Drug to Control Atrial Fibrillation.” *Cleveland Clinic Journal of Medicine* 68 (4): 353–63.
- Wang, Dao W., Akshitkumar M. Mistry, Kristopher M. Kahlig, Jennifer A. Kearney, Jizhou Xiang, and Alfred L. George Jr. 2010. “Propranolol Blocks Cardiac and Neuronal Voltage-Gated Sodium Channels.” *Frontiers in Pharmacology* 1 (December):144.
- Zemzemi, Nejib, and Blanca Rodriguez. 2015. “Effects of L-Type Calcium Channel and Human Ether-a-Go-Go Related Gene Blockers on the Electrical Activity of the Human Heart: A Simulation Study.” *Europace: European Pacing, Arrhythmias, and Cardiac Electrophysiology: Journal of the Working Groups on Cardiac Pacing, Arrhythmias, and Cardiac Cellular Electrophysiology of the European Society of Cardiology* 17 (2): 326–33.

Response to Reviewers

We sincerely thank the reviewers for their thoughtful and insightful comments, which have greatly enhanced the quality of our work. We are pleased to note that four reviewers have accepted our revised manuscript, with only minor revisions, all of which we have now fully addressed. In response to the key concerns raised by Reviewer #4, we have included baseline electrical recording data as suggested and expanded the Methods section to provide more comprehensive details about the characteristics of the cells used in our study. Additionally, we have referenced two previous studies from our lab, where we extensively characterized the electrophysiological responses of the cells used in this study. These references underscore the reproducibility of our measurements, despite the inherent variability within experiments and between batches.

Reviewer #1 (Remarks to the Author):

I would like to thank the authors for addressing my questions and the other reviewers questions so thoroughly. My concerns were all sufficiently well addressed, and I support the publication of this manuscript if the concerns of the other reviewers were addressed as well. There are only a few remaining formal items that need to be addressed:

1. please include the information of your response to question 12 in the supplementary information

Response:

Thank you for your comment, we have added that to the supplementary information under section II:

“It should be noted that recording from the same electrode during repetitive stimulation to obtain two consecutive identical iAPs, rather than using neighboring channels for iAPs, presents two main challenges with this approach:

1. **Gradual Change in iAP Shapes:** When cells are exposed to drugs, the iAP shapes can gradually change over time. As a result, two subsequent iAPs may not necessarily be similar, especially if the drug effects are still evolving. This makes it difficult to ensure that the recorded iAPs are truly identical or representative of a steady-state condition.
 2. **Electroporation Recovery Time:** After a cell is electroporated, it typically takes around 30 to 60 minutes for the membrane to reseal and recover fully. If we were to record an eAP) followed by the next iAP after electroporation, the 30-minute waiting period would significantly reduce the throughput of data recording. This slow process would hinder our ability to collect the large number of synchronized eAP/iAP pairs necessary for training deep learning models, which require substantial amounts of data.”
2. please include the information regarding your normalization and the rationale behind it as explained in your response to question 13 in the supplementary information

Response:

Thank you for your comment, we have added that to the methods section under eAP and iAP normalization.

“*eAP and iAP normalization*. The normalization is crucial to bring all data to a comparable scale and to emphasize relative changes in signal features over absolute values. The windows of iAPs

are normalized to a range between 0 and 1 starting from 0.1. For eAPs, a distinct approach is used for normalization. A specific segment of the eAP signal, particularly the values between indices 1150 and 1350, is selected. This segment is critical as it characterizes the noise within the eAP signal, based on the assumption that it accurately represents the noise characteristics of the entire signal. The peak index in each window is typically around index 1000, just before this segment. To quantify the noise, the function calculates the standard deviation (σ) of this selected segment, which serves as a measure of variation or dispersion within the values. Next, the eAP signal undergoes normalization by subtracting the mean of the entire eAP signal from each value. To ensure the normalization starts from 0, the initial value of the eAP signal is subtracted from all subsequent values. The final step scales the signal relative to its noise level, achieved by dividing the mean-adjusted values by 60 times the calculated standard deviation. The normalization methods applied to eAP and iAP arrays given that eAP or iAP are arrays of values ($u_1, u_2, \dots, u_{8000}$), are as follows:

$$iAP_{Normalized}(x_i) = 0.1 + 0.9 \left(\frac{iAP_{raw}(x_i) - iAP_{raw}(x_1)}{\max(iAP_{raw}) - iAP_{raw}(x_1)} \right) \quad (10)$$

$$eAP_{Normalized}(x_i) = \frac{eAP_{raw}(x_i) - \text{Mean}(eAP_{raw})}{60 \times \sigma(N)} - eAP_{raw}(x_1) \quad (11)$$

Reviewer #2 (Remarks to the Author):

The authors have made appropriate revisions to address my concerns. In the revised version, the authors supplemented data from different nanoelectrode arrays (NEA) recordings of eAP/iAP to validate the feasibility of the intracellular action potentials reconstruction algorithm. The results showed that the algorithm could be applied to different NEA systems, verifying the algorithm's generality and enhancing the innovation of the paper. Furthermore, the intracellular action potentials algorithm was validated in eAP/iAP obtained from hiPSC-CMs exposed to different drugs, including multiriot, quinine, nifedipine, flumequine, and lidocaine, demonstrating the algorithm's promising application prospects. Finally, the authors refined the details of the algorithm and writing to meet publication requirements. Therefore, I would recommend the acceptance of the work in Nature Communication.

Reviewer #3 (Remarks to the Author):

The authors have made significant improvements to the manuscript by addressing most of my suggestions, as well as those of the other reviewers. In particular, testing additional drugs has strengthened the evaluation of PIA-UNET's benefits and predictive power. However, I still have a few minor comments that need to be addressed:

1. Regarding my previous comment 9: While the explanation provided is now sufficient, I would recommend the following improvements to the notation for clarity and standardization: It is unconventional to use "x" and "y" for time and voltage, respectively. The more standard symbols would be "t" for time and "V" for voltage. Using the familiar "t" and "V" will make it easier for readers to follow.

Response:

Thank you for your suggestion. We have made the necessary corrections in Figure 2 and throughout the entire manuscript.

2. The derivatives should be expressed explicitly as " dV/dt " and " d^2V/dt^2 " rather than just " d/dt " or " d^2/dt^2 ", which are missing the dependent variable.

Response:

Thank you for your suggestion. We have made the necessary corrections throughout the entire manuscript and supplementary information. However, as an example, in the supplementary section, if the derivative is shown as $\frac{d}{dt}[k(l - v)(v - a) - v^{-1} \frac{dv}{dt}]$, this means that the derivative is applied to the entire expression within the brackets."

3. There seems to be inconsistency in the use of the variables "u" and "v", where sometimes " $1/u$ " and sometimes " $1/v$ " is written. Please ensure uniformity in the variable naming throughout the text.

Response:

Thank you for your suggestion. We have made the necessary corrections throughout the entire manuscript and supplementary information.

4. Regarding my previous comment 15: The explanation of "quantile crossing" is now clear, but the rationale for how separate output layers for the 0.05, 0.5, and 0.95 quantiles prevent this phenomenon remains unclear. Since these output layers are independent, one might expect them to increase the risk of quantile crossing. Could you provide a more detailed explanation or add some discussion to clarify how the separate layers address this issue?

Response:

Thank you for your comment. By integrating three parallel output layers corresponding to the 0.05, 0.5, and 0.95 quantiles into a single model, we effectively reduce the risk of quantile crossing. The shared hidden layers in the Q-PIA-UNET architecture learn common underlying features from the input data, providing a consistent foundation for all quantile predictions. The separate output layers then specialize in predicting their respective quantiles, fine-tuning the shared representations to capture the unique aspects of each quantile level.

Training these quantile predictions simultaneously within the same model allows for joint optimization, where the errors of all quantiles are considered together. This joint learning process implicitly enforces the natural ordering of quantiles, as the model adjusts its weights to minimize discrepancies between quantile levels. As a result, the predictions for the lower quantile (0.05) remain less than or equal to the median (0.5), which in turn remains less than or equal to the upper quantile (0.95), thus preventing quantile crossing.

This approach contrasts with models that predict quantiles independently, where the lack of shared learning and coordination between quantiles increases the risk of crossing. By utilizing a unified model with specialized output layers, we enhance both the accuracy and reliability of the quantile predictions, ensuring they align with the expected statistical order.

Additionally, we have included the following in the manuscript under the section "Quantile PIA-UNET with Confidence Interval (Q-PIA-UNET). *"By using separate output layers corresponding*

to each quantile in one model, the model ensures that the predicted values follow the correct order, improving both the accuracy and reliability of the results. The shared hidden layers in the Q-PIA-UNET architecture learn common underlying features from the input data, providing a consistent foundation for all quantile predictions. The separate output layers then specialize in predicting their respective quantiles, fine-tuning the shared representations to capture the unique aspects of each quantile level. Training these quantile predictions simultaneously within the same model allows for joint optimization, where the errors of all quantiles are considered together. This joint learning process implicitly enforces the natural ordering of quantiles, as the model adjusts its weights to minimize discrepancies between quantile levels. As a result, the predictions for the lower quantile (0.05) remain less than or equal to the median (0.5), which in turn remains less than or equal to the upper quantile (0.95), thus preventing quantile crossing.”

Reviewer #4 (Remarks to the Author):

The authors provided very long and detailed responses to the concerns raised during the review (over 80 pages, not counting references). While some issues have been addressed (for example inclusion of experiments with additional drugs), my main concerns remain. I copy here my major comments followed by the assessment of the authors' responses.

1. **Copied comment: Concern 1:** The biological component of this research is largely neglected. It is mentioned that studies were done using “stem cell derived cardiomyocytes”, without specifying what are the stem cells (human or animal; embryonic or iPSC cells), and how many lineages were used. I was unable to find a method section with detailed disclosure of the methodology. Even if broadly studied cell source and a published methodology were used, it is critical to provide extensive characterization of the molecular, ultrastructural and functional (contractility, electrophysiology, calcium handling) of cardiomyocytes and determine the level of their maturity. **Assessment of the authors' responses:** Characterization of commercially obtained iPSC-CMs is still lacking, and it is really important to document their properties at the time when the APs were measured. Likewise, the “human iPSC-derived cardiomyocytes that were heterogeneously composed, with a predominance of ventricular-like cells (57%), as well as atrial-like and nodal-like cells” are not characterized either. It is also not disclosed how the cells were classified into the ventricular, atrial-like and nodal-like cells in terms of both the methods and the criteria used, and how reproducible is the cell derivation. The authors only state that “the derivation of these cells followed a protocol approved by the Stanford University Human Subjects Research Institutional Review Board, which ensures ethical standards and informed consent from participants (Burrige et al. 2014)”. IRB is certainly needed if the cells were derived from the patients’ samples, but their detailed characterization is also very important. The authors seem to have limited appreciation for the spectrum of the phenotypes and maturity levels for iPSC-derived cardiomyocytes, both of which can markedly change the APs that are being measured. The proposed methodology cannot be agnostic to the biological properties of the cells.

Response:

We thank the reviewer for their valuable comment. We fully agree that the characterization and reproducibility of cell derivation are crucial, which is exactly why we selected the gold standard cell lines used in this study. These cell lines have been extensively tested and characterized by multiple labs, and we consistently achieve reproducible cell adhesion, contraction, and electrical activity within three days of culturing the cells on our devices. The commercial cell lines were chosen as an additional test set due to their low variability and high sensitivity to electrophysiological changes induced by drug treatment, as demonstrated by the vendor. Additionally, data is available on their gene expression patterns, electrophysiological properties, and calcium handling (Burrige et al. 2014)(Burrige, Holmström, and Wu 2015)(“Celo.Cardiomyocytes,” n.d.-a). To address the reviewer’s comments, we have included new sections detailing the characterization of the cell lines used to train our model, based on the ample scientific literature available on these cells. We recognize the importance of making this information easily accessible to readers, so we have expanded our Methods section to summarize the characteristics of these cells and provided more references to the many studies that have characterized them. We have also added a statement acknowledging that, although human iPSC-

CMs closely resemble human adult cardiomyocytes in many aspects, they remain developmentally immature. This is a critical issue in cardiovascular research, and we agree with the reviewer that it should be explicitly stated in any paper using hiPSC-CMs as a model system. We have updated the manuscript with the following content under the section "hiPSC-CMs Differentiation and Characterization:

“hiPSC-CMs differentiation and characterization. The non-commercial hiPSC-CMs used in this study were differentiated in the laboratory of Prof. Joseph C. Wu, at the Stanford Cardiovascular Institute using highly standardized protocols described previously in detail by Burridge et al. (Burridge, Holmström, and Wu 2015) (Burridge et al. 2014). Briefly, hiPSCs (line SCVI-273) were treated with 6 μ M CHIR99021 (Selleck Chemical) in RPMI supplemented with B27 without insulin for 2 days, followed by recovery in RPMI supplemented with B27 without insulin for 1 day, and then by treatment with 5 μ M IWR-1 (Selleck Chemical) for 2 days. After recovery in fresh RPMI plus B27 without insulin medium for 2 days, cells were switched to RPMI plus B27 with insulin for 2 days. The hiPSC-CMs were purified with glucose-free RPMI plus B27 with insulin medium for 2–4 days and maintained in RPMI plus B27 with insulin medium for subsequent experiments. Using patch clamp, these cells were demonstrated to be heterogeneous with ventricular-like cells being the predominant (57%) along with atrial-like and nodal-like cells (Burridge et al. 2014). The chemically defined differentiation method used in the Wu lab has repeatedly shown to provide a reproducible and scalable method for deriving cardiomyocytes from hiPSCs (Burridge, Holmström, and Wu 2015). Furthermore, results from independent labs suggest that hiPSC-CMs derived from iPSC lines using this protocol can be consistently recovered after cryopreservation, and demonstrate comparable and functional sarcoplasmic reticulum calcium handling (“Comparable Calcium Handling of Human iPSC-Derived Cardiomyocytes Generated by Multiple Laboratories” 2015). The commercial hiPSC cardiomyocytes were purchased from Celogics (Celo.Cardiomyocytes, Celogics, CAT # C50) and used as an additional test set due to their low variability and high Sensitivity to electrophysiological changes from drug treatment as shown by the vendors (“Celo.Cardiomyocytes,” n.d.-b). Previous characterization of these cells by patch clamp and fluorescent microscopy demonstrated their Human cardiomyocyte-like electrophysiology, as well as ventricular cardiomyocyte markers and structural characteristics (“Celo.Cardiomyocytes,” n.d.-c).”

2. **Copied comment: Concern 2:** Related to the previous comment, it would be also important to document reproducibility of the measurements both within the same cell preparation and batch to batch.

Assessment of the authors' responses

The authors acknowledge that “there are several batch-to-batch variabilities, and even heterogeneity within a single culture in the electrophysiological recordings of hiPSC-CMs”, but do not report this variability as suggested. Furthermore, they say that this variability is “ further highlighting the need for high-throughput approaches like the one proposed in our manuscript to obtain statistically significant results”. I do not agree that this is a better approach than optimizing and standardizing the cell derivation to minimize random changes in cell properties. Finally, the authors state: “Our new results included in our revised manuscript demonstrate that our model can accurately predict iAP across various cell batches under different conditions” without properly documenting and explaining this statement.

Response:

We thank the reviewer for their valuable comment. We would like to address the reviewer's comments by focusing on three main points:

- 1) We have now included additional baseline recordings for our devices, as suggested by the reviewer, to demonstrate signal variability across experiments, both with and without drug treatments.

Before Drug Addition (Baseline)

After Drug Addition

First Recorded Sample Order Last

Figure-SI-5. Comparison of eAP and iAP baseline recordings across different experiments, as well as a comparison of the drugs used in the study, their dosages, and their impact on normalized iAP shapes. Additionally, we include the wafer (W) and device (D) numbers of the unique Nano Electrode Arrays used in the study. A color gradient from red to blue indicates the timeline of the experiments, with the bluer tones representing data from more recent experiments.

- 2) Regarding the electrophysiological characterization and drug response of the cells within a single culture and across batches, much of this characterization has been detailed in two recent manuscripts from our labs (Jahed et al. 2022; "Cardiotoxicity Drug Screening Based on Whole-Panel Intracellular Recording" 2022): In these papers, where we first introduced the nano-crown technology used in this study, we provided a full characterization of the electrophysiological properties and drug response of the hiPSC-CMs. Please refer to manuscripts and the plots below for examples of this characterization.

Figure 2. Multi-channel intracellular recording showing slight shape variability in action potentials shapes for cells within the same culture (from recent publications by our labs) (Jahed et al. 2022; “Cardiotoxicity Drug Screening Based on Whole-Panel Intracellular Recording” 2022).

Figure 3. A raw data trace of intracellular action potentials (iAPs) with sequential addition of pharmacological compounds for ~84 cells across 3 separate experiments. (Jahed et al. 2022; “Cardiotoxicity Drug Screening Based on Whole-Panel Intracellular Recording” 2022).

- 3) We want to clarify that the variability we report is not due to a lack of reproducibility. As demonstrated in our previous papers, our high-throughput recording methods enable us to observe a range of electrophysiological characteristics in cells within a single culture and across different cultures while maintaining constant experimental conditions. This type of

variability was previously unobservable with lower-throughput methods like patch clamp, which lack the ability to measure multiple cells simultaneously. In this study, these small variations provide valuable data for training our model on the relationships between intracellular and extracellular signals, which is a key objective of our work. However, the natural variations observed were not substantial enough on their own, which is why we introduced drugs to induce more pronounced changes in signal shape. Drug-induced variations are much larger than the minor heterogeneity observed across cells within one culture. Additionally, we did not control for beating frequency because we want our model to capture a range of beating frequencies. Some of the observed shape variations are indeed due to differences in beating frequencies.

3. **Concern 3:** In Figure 2h, through SHAP, ΔT_d , DR, and IR were identified as the most important features to predict APD (especially for $> APD_{50}$). How do these features correspond to the biological cardiac action potential? Are the minima (labeled as bp3) generally observed in all NEA eAP recordings? If so, is this feature an empirically determined one or one based on known cardiac biology? Was there any feature selection prior to training the XGBoost algorithm? This question (also raised by other reviewers) was aimed at clarifying to which extent the culture iPSC-derived cardiomyocytes recapitulate cardiac electrophysiology. Instead of trying to benchmark their recordings against the biological data, they discuss relationships between different parameters extracted from the recordings.

Response:

The reviewer has raised many points. We have broken down the question to answer each point:

- 3.1 In Figure 2h, through SHAP, ΔT_d , DR, and IR were identified as the most important features to predict APD (especially for $> APD_{50}$). How do these features correspond to the biological cardiac action potential?

Response:

Thank you for your insightful comment. In the updated manuscript using new data, we found that ΔT_d is the most important feature in predicting **APD₅₀**, **APD₇₀**, and **APD₉₀**. According to the SHAP plots, higher values of ΔT_d result in an increase in the action potential duration (APD) of the cardiac action potential. For smaller APD values, such as **APD₃₀**, ΔT_s emerged as the most important feature, and an increase in ΔT_s results in a lower APD value. Interestingly, ΔT_d has a negative impact on lower APD values like **APD₃₀**; increasing ΔT_d results in a decrease in the APD value, which is the opposite of its relationship with higher APD values.

Regarding how these features correspond to the biological cardiac action potential, it is important to note that these features cannot be explicitly correlated with specific phases of the iAP that result from the opening or closing of particular ion channels. This is due to the complex relationship between the eAP recorded by our NEA and the iAP within cardiomyocytes.

The relationship between eAP and iAP can be explained using circuit models of cardiac electrophysiology. In these models, the cell membrane is represented as a capacitor in parallel with resistive elements that symbolize ionic conductances. The capacitor represents the membrane's ability to store and release charge, capturing the dynamic changes in membrane potential. The presence of capacitive elements introduces temporal dynamics that influence the relationship

between intracellular and extracellular potentials. Specifically, capacitors cause the extracellular potentials to depend not only on the instantaneous transmembrane currents but also on the history of voltage changes due to the accumulation and release of charge over time. This effect introduces phase shifts and time delays in the eAP relative to the iAP.

Further, these capacitive properties disrupt direct, instantaneous relationships between changes in the iAP and eAP. Dependencies are introduced that are not strictly based on immediate past values but also on the accumulated charge over time. Consequently, it becomes difficult to establish a direct, phase-specific correspondence between features of the eAP and specific phases of the iAP associated with ion channel activities.

3.2 Are the minima (labeled as bp3) generally observed in all NEA eAP recordings?

Response:

Yes, if the eAP recordings are not extremely noisy, bp3 can be captured. This feature is observed in all recordings (except for the extremely noisy ones) from both NEA and MEA data, and for both commercial and non-commercial hiPSC-CMs used in the study. The existence of this point is also supported by other studies not conducted by our group, such as (Millard et al. 2018) in Figure 1 on hiPSC-CMs, (Dipalo et al. 2017) in Figure 4 on HL-1 cardiac-derived cells, and (Hayes et al. 2019) in Figure 1 on neonatal rat ventricular myocytes (NRVMs). These studies employed different types of MEAs for their recordings. Interestingly, (Hayes et al. 2019) referred to eAPs that do not follow the same trend or shape as "abnormal."

3.3 If so, is this feature an empirically determined one or one based on known cardiac biology?

Response:

These features were selected based on three primary factors: (1) alignment with methodologies used in previous studies, (2) the critical role of dV/dt (the rate of voltage change) in circuit models of cardiac electrophysiology, and (3) the ability to comprehensively reconstruct the extracellular action potential (eAP) waveform from the selected features.

These features are selected based on three factors:

1) **Alignment with Previous Methodologies:** We adhered to the methods used in prior studies to ensure consistency and comparability of results. Previous studies have demonstrated the importance of specific features in analyzing cardiac action potentials. For example, (Hayes et al. 2019) used ΔV_1 for their eAP characterization and discovered, based on their *in silico* models, that ΔV_1 correlates highly with conduction velocity. However, this correlation was not significant in actual experiments. Another study investigating the relationship between eAP and iAP has selected and analyzed features such as bp_1 , bp_2 , ΔV_1 , and ΔV_2 (Halbach et al. 2003) in their research on mouse cardiac myocyte cultures. Furthermore, another study employed a feature equivalent to the ΔT_d used in our research (Clements and Thomas 2014) on human embryonic stem cell-derived cardiomyocytes.

2) **Significance of dV/dt in Circuit Models:** The derivative of voltage with respect to time (dV/dt) is crucial in cardiac circuit models for accurately capturing the dynamics of action potentials. We

ensured that the selected features capture points where either the derivative (dV/dt) or the potential values (V) exhibit significant changes.

3) Reconstruction Capability of eAP: The selected features must enable the comprehensive reconstruction of the eAP from the chosen parameters. Building on these foundations, we expanded the feature set to fully capture the waveform shape of the eAP, ensuring a more comprehensive analysis of its characteristics.

3.4 Was there any feature selection prior to training the XGBoost algorithm? This question (also raised by other reviewers) was aimed at clarifying to which extent the culture iPSC-derived cardiomyocytes recapitulate cardiac electrophysiology.

Response:

Yes, feature selection was conducted prior to training the XGBoost algorithm to ensure that only the most relevant and highly correlated features were utilized in reconstructing iAP from eAP waveforms. Please consider that the primary objective of this study was to develop methods for the accurate reconstruction of iAP using artificial intelligence (AI). We analyzed the correlations between all extracted features to identify pairs or groups of features that were highly correlated with each other. For features exhibiting high mutual correlation, we removed the redundant ones to eliminate multicollinearity and reduce model complexity. Among the highly correlated features, we retained the feature that was less prone to distortion. For instance, we selected ΔT_s over ΔT_1 and ΔT_2 because ΔT_s demonstrated a high correlation value with them and were not affected by distortions in the spike region of the eAP waveform.

a Correlation Heatmap Between ΔT_1 , ΔT_2 , ΔT_s , and ΔT_d in data with Undistorted eAP Spikes

b Correlation Heatmap Between eAP features in data including Both Distorted and Undistorted eAP Spikes

Figure-SI-6. **a)** Correlation between temporal-related eAP features (ΔT_1 , ΔT_2 , ΔT_s , and ΔT_d) for the undistorted portion of eAP/iAP data collected to be utilized in the machine learning and deep learning models. **b)** Correlation between measurable eAP features (excluding ΔT_1 and ΔT_2) for all of the eAP/iAP data (with distorted and undistorted eap spikes) utilized in the machine learning and deep learning models.

3.5 Instead of trying to benchmark their recordings against the biological data, they discuss relationships between different parameters extracted from the recordings.

Response:

The inherent complexity of eAP features makes it challenging to correlate them explicitly (Halbach et al. 2003) and in an autoregressive manner with iAP phases. In our study, we collected data from hiPSC-CMs exposed to different drugs to assess our model's performance. Importantly, we also evaluated the model on commercially sourced hiPSC-CMs that were not included in the training data and on samples exposed to previously unseen drugs. These assessments demonstrate our model's capability to accurately reconstruct iAP from eAP waveforms and their features. This robust performance across diverse hiPSC-CM sources and novel pharmacological conditions underscores the effectiveness of our AI-based approach in capturing the essential electrophysiological characteristics necessary for accurate iAP reconstruction. Achieving this was the primary objective of our study.

4. **Concern 4:** Are the datasets used for the feature-based model in Figure 2 the same as the ones used in Figure 3? If so, it seems like the feature-based approach is better (APD Error = 0.196s) than the proposed PIA-UNET (APD Error = 0.25s). What is the added benefit of the DL approach? It also seems that the DL approach might have an overfitting issue, with APD error of the test set diverging significantly from the train set. This is another important question for any AI/ML algorithm: clear separation of the training datasets from the experimental datasets. The authors do not clearly respond to this question and instead say: “Yes, the datasets used in Figure 2 and Figure 3 are from the same source. However, for the XGBoost method, we applied screening methods to only retain signals where the critical points were computed accurately by our algorithm and exhibited measurable ΔT s in their spikes. While XGBoost demonstrated robust performance, it is limited to predicting APD values and cannot reconstruct the entire iAP shape. This makes it suitable for simple studies but insufficient for more comprehensive analysis. Additionally, XGBoost requires intensive data preparation to extract waveform features, whereas the deep learning (DL) method does not. The DL approach can also handle distorted spikes, providing greater versatility”.

Response:

Yes, the same dataset was utilized for both the XGBoost models and the PIA-UNET as presented in Figures 2 and 3. However, it is important to note that our analysis was conducted on an expanded and more diverse dataset to better evaluate the performance of both models under various conditions.

For the XGBoost model, the mean absolute APD (Action Potential Duration) errors for the test sets were as follows:

- **Test 1:** 0.020 ± 0.007 seconds
- **Test 2:** 0.040 ± 0.006 seconds
- **Test 3:** 0.047 ± 0.038 seconds

In contrast, the PIA-UNET model exhibited the following mean absolute APD errors:

- **Test 1:** 0.030 ± 0.024 seconds
- **Test 2:** 0.027 ± 0.023 seconds

- **Test 3:** 0.020 ± 0.019 seconds

These results indicate that **PIA-UNET outperforms XGBoost in more challenging scenarios (Test 2 and Test 3)**, such as when dealing with unseen drugs or new devices (MEA). Specifically, while XGBoost showed lower errors in simpler test sets (e.g., Test 1), its performance degraded significantly in more complex scenarios (e.g., Test 3). Conversely, PIA-UNET maintained consistent and lower error rates across all test sets, demonstrating superior robustness and generalizability.

Here is a summary of the added benefits of the deep learning (DL) approach:

- 1. Comprehensive Reconstruction:**
 - a. XGBoost:** Limited to predicting APD values, which provides a singular aspect of the action potential.
 - b. PIA-UNET:** Capable of reconstructing the entire iAP shape from the eAP waveforms, offering a more holistic understanding of cardiac electrophysiology.
- 2. Minimal Data Preparation:**
 - a. XGBoost:** Requires extensive feature engineering and precise feature detection on eAPs. This process is particularly challenging with distorted or highly noisy eAP signals.
 - b. PIA-UNET:** Operates directly on raw eAP data without the need for feature extraction, simplifying the workflow and reducing potential sources of error.
- 3. Handling Noisy and Distorted Data:**
 - a. XGBoost:** Struggles with distorted or noisy eAPs as accurate feature detection becomes unreliable.
 - b. PIA-UNET:** Exhibits greater versatility by effectively managing distorted spikes and maintaining performance, as evidenced by its consistent APD error rates across diverse test sets.
- 4. Reduced Risk of Overfitting:**
 - a.** Although there was an initial concern regarding overfitting in the DL approach, our extensive testing on diverse and unseen datasets (including different hiPSC-CM sources and novel drug exposures) demonstrated that **PIA-UNET generalizes well**. The APD errors on the test sets did not show significant divergence from the training set, indicating that overfitting is not a prominent issue with our model.

We also added the following to the manuscript: *“PIA-UNET outperforms the feature-based XGBoost in action potential (AP) analysis. While XGBoost is limited to predicting APD values and requires extensive feature engineering, PIA-UNET reconstructs entire iAP shapes directly from raw eAP data, providing a more comprehensive understanding of cardiac electrophysiology. PIA-UNET also handles noisy or distorted data more effectively and demonstrates better generalization across diverse test sets, minimizing the risk of overfitting.”*

- 5. Concern 5:** Dofetilide was used to generate a more diverse set of waveform data to train the model. In validating the model, the authors use the same drug to demonstrate the reconstruction of iAP using eAP recordings. An additional drug other than dofetilide would validate the method in a more

robust manner. On a similar note, does the model generalize to other classes of cardiac drugs that affect the AP waveform differently (e.g. in the QRS phases/depolarization rather than QT)? Investigating the effects of another set of drugs (such as a Na class IC blocker that perturbs the QRS phase or a class Ib drug that shortens QT) would strengthen the generalizability of this model to capture perturbations to additional parts of the AP waveform.

Assessment of the authors' responses: In response to this comment, the authors extended their studies to additional drugs, which is an improvement.

Response:

We thank the reviewer for accepting our new drug and cell experiments in response to this concern.

Reviewer 54 (Remarks to the Author):

References

- Burridge, Paul W., Alexandra Holmström, and Joseph C. Wu. 2015. "Chemically Defined Culture and Cardiomyocyte Differentiation of Human Pluripotent Stem Cells." *Current Protocols in Human Genetics / Editorial Board, Jonathan L. Haines ... [et Al.]* 87 (October):21.3.1.
- Burridge, Paul W., Elena Matsa, Praveen Shukla, Ziliang C. Lin, Jared M. Churko, Antje D. Ebert, Feng Lan, et al. 2014. "Chemically Defined Generation of Human Cardiomyocytes." *Nature Methods* 11 (8): 855–60.
- Cardiotoxicity Drug Screening Based on Whole-Panel Intracellular Recording." 2022. *Biosensors and Bioelectronics* 216 (November):114617.
- Celo.Cardiomyocytes." n.d.-a. Celogics. Accessed October 22, 2024. <https://www.celogics.com/celocardiomyocytes>.
- . n.d.-b. Celogics. Accessed October 22, 2024. <https://www.celogics.com/celocardiomyocytes>.
- . n.d.-c. Celogics. Accessed October 22, 2024. <https://www.celogics.com/celocardiomyocytes>.
- Clements, Mike, and Nick Thomas. 2014. "High-Throughput Multi-Parameter Profiling of Electrophysiological Drug Effects in Human Embryonic Stem Cell Derived Cardiomyocytes Using Multi-Electrode Arrays." *Toxicological Sciences: An Official Journal of the Society of Toxicology* 140 (2): 445–61.
- Comparable Calcium Handling of Human iPSC-Derived Cardiomyocytes Generated by Multiple Laboratories." 2015. *Journal of Molecular and Cellular Cardiology* 85 (August):79–88.
- Dipalo, Michele, Hayder Amin, Laura Lovato, Fabio Moia, Valeria Caprettini, Gabriele C. Messina, Francesco Tantussi, Luca Berdondini, and Francesco De Angelis. 2017. "Intracellular and Extracellular Recording of Spontaneous Action Potentials in Mammalian Neurons and Cardiac Cells with 3D Plasmonic Nanoelectrodes." *Nano Letters* 17 (6): 3932–39.
- Halbach, Marcel, Ulrich Egert, Jürgen Hescheler, and Kathrin Banach. 2003. "Estimation of Action Potential Changes from Field Potential Recordings in Multicellular Mouse Cardiac Myocyte Cultures." *Cellular Physiology and Biochemistry: International Journal of Experimental Cellular Physiology, Biochemistry, and Pharmacology* 13 (5): 271–84.
- Hayes, Heather B., Anthony M. Nicolini, Colin A. Arrowood, Stacie A. Chvatal, David W. Wolfson, Hee

- Cheol Cho, Denise D. Sullivan, et al. 2019. "Novel Method for Action Potential Measurements from Intact Cardiac Monolayers with Multiwell Microelectrode Array Technology." *Scientific Reports* 9 (1): 11893.
- Jahed, Zeinab, Yang Yang, Ching-Ting Tsai, Ethan P. Foster, Allister F. McGuire, Huaxiao Yang, Aofei Liu, et al. 2022. "Nanocrown Electrodes for Parallel and Robust Intracellular Recording of Cardiomyocytes." *Nature Communications* 13 (1): 2253.
- Millard, Daniel, Qianyu Dang, Hong Shi, Xiaou Zhang, Chris Strock, Udo Kraushaar, Haoyu Zeng, et al. 2018. "Cross-Site Reliability of Human Induced Pluripotent Stem Cell-Derived Cardiomyocyte Based Safety Assays Using Microelectrode Arrays: Results from a Blinded CiPA Pilot Study." *Toxicological Sciences: An Official Journal of the Society of Toxicology* 164 (2): 550–62.